# Structure and activation of the RING E3 ubiquitin ligase TRIM72 on the membrane

Si Hoon Park [1,6], Juhyun Han[1], Byung-Cheon Jeong[1,7], Ju Han Song[1,8], Se Hwan Jang[2], Hyeongseop Jeong [3], Bong Heon Kim[1], Young-Gyu Ko [1], Zee-Yong Park[2], Kyung Eun Lee[4], Jaekyung Hyun [5] & Hyun Kyu Song [1]✉

Defects in plasma membrane repair can lead to muscle and heart diseases in humans. Tripartite motif-containing protein (TRIM)72 (mitsugumin 53; MG53) has been determined to rapidly nucleate vesicles at the site of membrane damage, but the underlying molecular mechanisms remain poorly understood. Here we present the structure of *Mus musculus* TRIM72, a complete model of a TRIM E3 ubiquitin ligase. We demonstrated that the interaction between TRIM72 and phosphatidylserine-enriched membranes is necessary for its oligomeric assembly and ubiquitination activity. Using cryogenic electron tomography and subtomogram averaging, we elucidated a higher-order model of TRIM72 assembly on the phospholipid bilayer. Combining structural and biochemical techniques, we developed a working molecular model of TRIM72, providing insights into the regulation of RING-type E3 ligases through the cooperation of multiple domains in higher-order assemblies. Our findings establish a fundamental basis for the study of TRIM E3 ligases and have therapeutic implications for diseases associated with membrane repair.

Cells require the repair of plasma membrane injuries to protect the cytoplasm from external environments. In muscle tissue, plasma membrane injuries frequently occur in response to mechanical and metabolic stresses[1]. Fortunately, membrane-repair machinery can rapidly reseal the damaged plasma membrane[2,3]. Defects in membrane repair can cause diseases, including muscular dystrophy and neurodegenerative diseases[4–6]. Since the initial discovery of membrane resealing[7], several membrane-repair models have been proposed: patching, constriction, exocytosis–endocytosis-mediated and endosomal sorting complex required for transport (ESCRT)-mediated models[8,9]. However, despite the expanding biological and biochemical understanding gained from these models in recent years[10–15], the lack of structural evidence makes it difficult to elucidate the molecular mechanisms of membrane resealing.

One key factor is that extracellular $Ca^{2+}$ influx triggers membrane repair at damaged sites rich in phosphatidylserine (PS)[2,16] through a $Ca^{2+}$-dependent ESCRT system and $Ca^{2+}$-induced exocytosis of lysosomes[7,17]. However, despite the importance of $Ca^{2+}$ influx, there is also a unique $Ca^{2+}$-independent membrane-repair process known to be mediated by tripartite motif-containing protein (TRIM)72 (ref. 18). TRIM72, also known as MG53, which is highly expressed in muscle[18,19], is a key initiator of the plasma membrane-repair machinery that facilitates vesicle transport to the injury site[18]. Additionally, TRIM72 exerts protective effects in wound healing following ischemia–reperfusion injury[20] and acute kidney and lung injury[21,22]. It has been shown to have therapeutic potential for various injuries in both muscle and non-muscle tissues[23]. In addition to participating in the membrane-repair process, TRIM72 has functions related to the innate immune response[24],

[1]Department of Life Sciences, Korea University, Seoul, South Korea. [2]School of Life Sciences, Gwangju Institute of Science and Technology, Gwangju, South Korea. [3]Center for Electron Microscopy Research, Korea Basic Science Institute, Cheongju-si, South Korea. [4]Advanced Analysis Center, Korea Institute of Science and Technology, Seoul, South Korea. [5]School of Pharmacy, Sungkyunkwan University, Suwon, South Korea. [6]Present address: Friedrich Miescher Institute for Biomedical Research, Basel, Switzerland. [7]Present address: CSL Seqirus, Waltham, MA, USA. [8]Present address: Department of Pharmacology and Dental Therapeutics, School of Dentistry, Chonnam National University, Gwangju, South Korea. ✉e-mail: hksong@korea.ac.kr

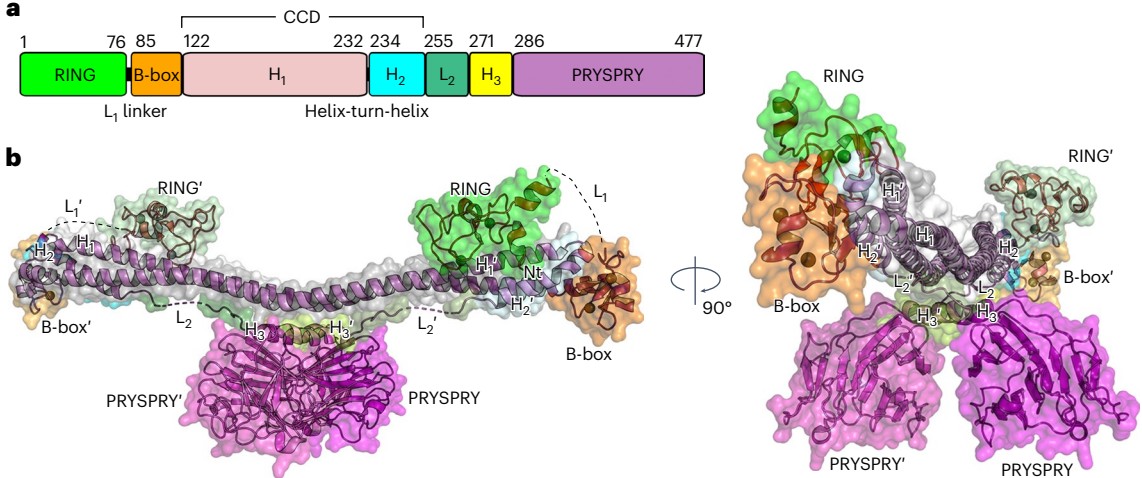

**Fig. 1 | Crystal structure of TRIM72. a,** Domain organization of TRIM72. Each domain and small motif is shown as follows: RING (green); $L_1$ linker, B-box (orange); $H_1$ helix (pink); helix-turn-helix, $H_2$ helix (cyan); $L_2$ linker (mint); $H_3$ helix (yellow); and PRYSPRY (magenta). The residues are numbered based on the mouse TRIM72 sequence. **b,** Overall structure of mouse TRIM72 represented by a surface model with a ribbon diagram. The color of each domain corresponds to the domain architecture in **a**. The second protomer is labeled with a prime symbol (') and is shown in a less-vibrant color. The structural models were derived from mouse TRIM72 ΔRING (2.75 Å), except for the RING domain (green), which came from TRIM72 FL (4.6 Å).

insulin resistance[25,26] and muscle differentiation[19]. However, due to the lack of structural and biochemical studies, its molecular mechanism remains largely unknown.

TRIM72 is a member of the TRIM superfamily[27], a major subfamily of Really Interesting New Gene (RING)-type E3 ubiquitin (Ub) ligases[28]. Ubiquitination is a key protein quality-control system in which Ub is attached to substrates in a reaction catalyzed by an enzymatic cascade involving E1, E2 and E3 enzymes, leading to degradation of the substrate by the 26S proteasome. In addition, Ub ligases are known to trigger signal transduction, membrane trafficking and DNA damage repair processes[29]. They are classified into three types based on their mechanism of Ub transfer: RING, Homologous to the E6AP C-Terminus (HECT) and RING-between-RING (RBR) ligases[30]. RING-type ligases are the most abundant, and TRIM RING ligases contain over 70 members among the 600 Ub ligases in humans[31,32]. TRIMs are characterized by multimodular domains, including the conserved RING–B-box–coiled-coil (RBCC) and variable C-terminal domains. Although their domain architecture is similar, their functions and related diseases, such as antiviral defense, cancers and inherited disorders, are extremely broad[32–34]. However, despite biological evidence, in-depth structural and biochemical research on TRIMs is quite limited due to their flexibility and complexity[35–52].

Here, we have elucidated the structures of TRIM72 and higher-order assemblies of TRIM72 complexed with a phospholipid membrane using integrative structural techniques. These structures enable us to understand how TRIM72 recognizes negatively charged phospholipid domains independent of $Ca^{2+}$ during membrane repair. We have shown that the binding of TRIM72 to the phospholipid membrane induces its self-assembly, which enhances its ubiquitination activity via intermolecular RING domain dimerization. In conclusion, we propose an activation mechanism of higher-order TRIM72 assembly for Ub transfer on the phospholipid membrane.

## Results

### Crystal structure of dimeric TRIM72

TRIM72 is a member of the class IV TRIM proteins, which is the largest subtype among the 11 groups of TRIM proteins. Its domain organization includes an RBCC domain followed by a PRY motif associated with a SPRY domain identified in splA kinase and ryanodine receptors (PRYSPRY) domain (Fig. 1a)[53]. TRIM72 is a tight dimer, as determined

by size exclusion chromatography (SEC) coupled with multiangle light scattering (SEC–MALS; Extended Data Fig. 1a,b), and its overall shape was determined to be elongated in solution by SEC coupled with small-angle X-ray scattering (SEC–SAXS; Extended Data Fig. 1c–f). To obtain a high-resolution model, numerous constructs were generated, purified and crystallized; however, most of them diffracted poorly. After introducing point mutations and deletions, we were finally able to resolve molecular models of mouse TRIM72, including RING-deleted (ΔRING, 2.75 Å; the highest resolution is in parentheses), full-length (FL, 4.6 Å) and wild-type (WT, 7.1 Å) TRIM72. Details of the constructs and crystal structures used are given in Supplementary Table 1 and Table 1.

The overall structure of the TRIM72 dimer has an elongated shape (Fig. 1b and Supplementary Fig. 1). The coiled-coil domains (CCDs) form the main scaffold of an antiparallel dimer with a length of approximately 17 nm (Supplementary Fig. 2a–e). In the middle of the CCDs, the structural core consists of the hendecad repeated region of the $H_1$:$H_1$' helices (prime indicates the second protomer of the dimer) and the $H_3$:$H_3$' helices followed by a pair of PRYSPRY domains (Supplementary Fig. 2f). Between the $H_1$:$H_1$' helices and PRYSPRY domains, the $H_3$:$H_3$' helices are sandwiched to interact with both domains. One side of the $H_3$:$H_3$' helices is wrapped in a four-helical bundle with the hendecad $H_1$:$H_1$' helices, and the other side contacts the β-sheet of PRYSPRY through hydrophobic interfaces, hydrogen bonding and ionic interactions (Supplementary Fig. 2g,h). Owing to these interdomain interactions, two PRYSPRY domains are located on the opposite side of the CCDs across the $H_3$:$H_3$' helices (Fig. 1b).

At the N terminus, a catalytic RING exists in a bent conformation near the other protomer of the CCD (Supplementary Fig. 3a). Next to the RING domain, the B-box domains are located at each end of the CCDs and contact the $H_1$ and $H_2$' helices of the CCDs in a hydrophobic manner (Supplementary Fig. 3b). The overall structure of TRIM72 obtains a complete model of a member of the TRIM family, which enabled us to investigate the structure–function relationship in detail.

### TRIM72 binds to the membrane via its PRYSPRY pair

PRYSPRY comprises two antiparallel β-sheets: one sheet ($\beta_3$, $\beta_5$–$\beta_8$ and $\beta_{12}$) forms a concave surface, and the other sheet ($\beta_1$, $\beta_2$, $\beta_4$, $\beta_9$–$\beta_{11}$ and $\beta_{13}$) forms a convex surface (Fig. 2a)[37]. The TRIM72 dimer contains

**Table 1 | X-ray crystallography data collection and structure-refinement statistics**

| | ΔRING (PDB 7XV2) | ΔRING (PDB 7XZ0) | FL/C242S (PDB 7XZ1) | FL (PDB 7XYZ) | FL (PDB 7XZ2) | WT (PDB 7XYY) |
|---|---|---|---|---|---|---|
| **Data collection** | | | | | | |
| Space group | $C2$ | $C2$ | $I4_122$ | $P3_1$ | $P2_1$ | $C2$ |
| Cell dimensions | | | | | | |
| $a, b, c$ (Å) | 73.3, 110.2, 96.0 | 239.7, 73.0, 83.8 | 235.9, 235.9, 158.8 | 174.4, 174.4, 225.0 | 77.0, 327.0, 121.0 | 367.0, 98.1, 103.2 |
| $\alpha, \beta, \gamma$ (°) | 90.0, 100.8, 90.0 | 90.0, 97.6, 90.0 | 90.0, 90.0, 90.0 | 90.0, 90.0, 120.0 | 90.0, 90.9, 90.0 | 90.0, 101.3, 90.0 |
| Resolution (Å) | 30.0–2.75 (2.80–2.75) | 50.0–3.28 (3.34–3.28) | 50.0–5.20 (5.29–5.20) | 50.0–4.60 (4.68–4.60) | 50.0–3.50 (3.56–3.50) | 50.0–7.10 (7.22–7.10) |
| $R_{merge}$ | 0.084 (2.165) | 0.116 (1.943) | 0.135 (3.094) | 0.252 (0.513) | 0.216 (0.984) | 0.191 (1.143) |
| $I/\sigma(I)$ | 27.3 (1.3) | 21.5 (1.2) | 49.8 (1.3) | 6.8 (1.9) | 9.3 (1.8) | 10.1 (1.2) |
| Completeness (%) | 98.0 (98.9) | 98.8 (98.9) | 99.9 (100.0) | 97.7 (87.6) | 99.9 (99.9) | 99.4 (98.4) |
| Redundancy | 4.1 (4.0) | 3.9 (3.9) | 28.8 (25.9) | 4.8 (3.0) | 3.8 (3.5) | 3.4 (3.2) |
| **Refinement** | | | | | | |
| Resolution (Å) | 27.4–2.75 (2.79–2.75) | 37.7–3.28 (3.36–3.28) | 38.9–5.20 (5.53–5.20) | 47.3–4.62 (4.70–4.62) | 38.2–3.50 (3.59–3.50) | 49.0–7.10 (7.81–7.10) |
| No. reflections | 18,994 | 21,675 | 8,803 | 38,003 | 74,203 | 5,474 |
| $R_{work}$ / $R_{free}$ | 20.38/25.84 | 25.41/30.78 | 26.59/31.52 | 31.21/34.84 | 24.90/26.89 | 26.94/31.37 |
| No. atoms | | | | | | |
| Protein | 3,022 | 6,108 | 6,044 | 14,220 | 12,152 | 6,032 |
| Ligand/ion | 2 | 4 | 4 | 16 | 10 | 4 |
| $B$ factors | | | | | | |
| Protein | 115.0 | 139.0 | 444.0 | 332.7 | 149.0 | 295.7 |
| Ligand/ion | 148.2 | 218.1 | 492.4 | 571.2 | 308.7 | 325.3 |
| R.m.s. deviations | | | | | | |
| Bond lengths (Å) | 0.009 | 0.003 | 0.002 | 0.005 | 0.002 | 0.003 |
| Bond angles (°) | 1.113 | 0.650 | 0.691 | 1.028 | 0.497 | 0.772 |
| Ramachandran plot | | | | | | |
| Favored (%) | 94.44 | 94.01 | 94.97 | 85.14 | 95.08 | 89.02 |
| Allowed (%) | 5.56 | 5.99 | 6.03 | 14.8 | 4.92 | 10.98 |
| Outliers (%) | 0 | 0 | 0 | 0.06 | 0 | 0 |

One crystal was used for each structure. Values in parentheses are for the highest-resolution shell.

two PRYSPRY domains that exhibit unique structural characteristics. First, several arginines (R356, R368, R369, R371 and R386) and lysines (K317, K330, K389, K398, K460 and K462) are situated on the concave surface, consisting of variable loops (VLs), and contribute to its positive charge (Fig. 2a and Extended Data Fig. 2). Second, each PRYSPRY domain displays a concave surface that is positively charged toward the same side, facing opposite the CCDs (Fig. 2b). Finally, the orientation of the PRYSPRY pair is strictly maintained by its interaction with the $H_3$:$H_3'$ helices (Fig. 2b and Supplementary Fig. 2g,h). Through extensive protein–lipid overlay analysis, it was revealed that TRIM72 specifically recognizes negatively charged phospholipids such as PS and phosphatidylinositol but not phosphatidylcholine, phosphatidylethanolamine, oxidized phospholipids or sphingolipids (Extended Data Fig. 3). Flow cytometry analysis showed that TRIM72 also interacted with PS-containing liposomes (PS liposomes) with a much higher affinity for 30 mol% PS than for 10 mol% PS (Fig. 2c). This suggests that clustered and negatively charged phospholipid domains are required for TRIM72–PS binding.

To investigate the interaction between PRYSPRY domains and negatively charged phospholipid domains, we conducted liposome coflotation assays using the TRIM72 mutants $K^2D$ (K460D, K462D) and $R^3E$ (R368E, R369E, R371E), which introduce charge reversal in the PRYSPRY domain. In contrast to WT, the charge-reversal mutants

did not exhibit any interaction with PS liposomes (Fig. 2d). We also examined the orientations of the PRYSPRY domains with respect to the membrane by generating a TRIM72 mutant, $\Delta H_3$ (deletion of residues 272–281), that disrupted the orientation of the PRYSPRY domains. As expected, we observed a loss of interaction between PS liposomes and the deletion mutant, confirming the importance of the orientation of PRYSPRY domains in recognizing the negatively charged phospholipid membrane (Fig. 2d). Next, we measured the binding affinity between TRIM72 and PS liposomes. The binding constant $K_D$ was calculated to be 9.9 nM, which is similar to the affinity between annexin V and PS (Extended Data Fig. 4a,b)[54]. We observed that dimeric glutathione S-transferase (GST)-fused $H_3$–PRYSPRY interacted with PS liposomes with $_{a}K_D$ of 76.7 nM, which was 7.7-fold lower than that of WT (Extended Data Fig. 4a,g). However, monomeric maltose-binding protein (MBP)-fused $H_3$–PRYSPRY did not bind to PS liposomes, similar to the defective mutants RBCC, $K^2D$, $R^3E$ and $\Delta H_3$ (Extended Data Fig. 4). Thus, we interpret this result to mean that avidity is critical for PS recognition by TRIM72.

The results were also confirmed in vivo upon fractionation of both the membrane and cytosol in C2C12 myoblast cells after transfection to express human TRIM72. In contrast to TRIM72 WT, the mutants ($K^2D$, $R^3E$ and $\Delta H_3$) exhibited significant defects in membrane localization (Fig. 2e and Supplementary Fig. 4). Therefore, under

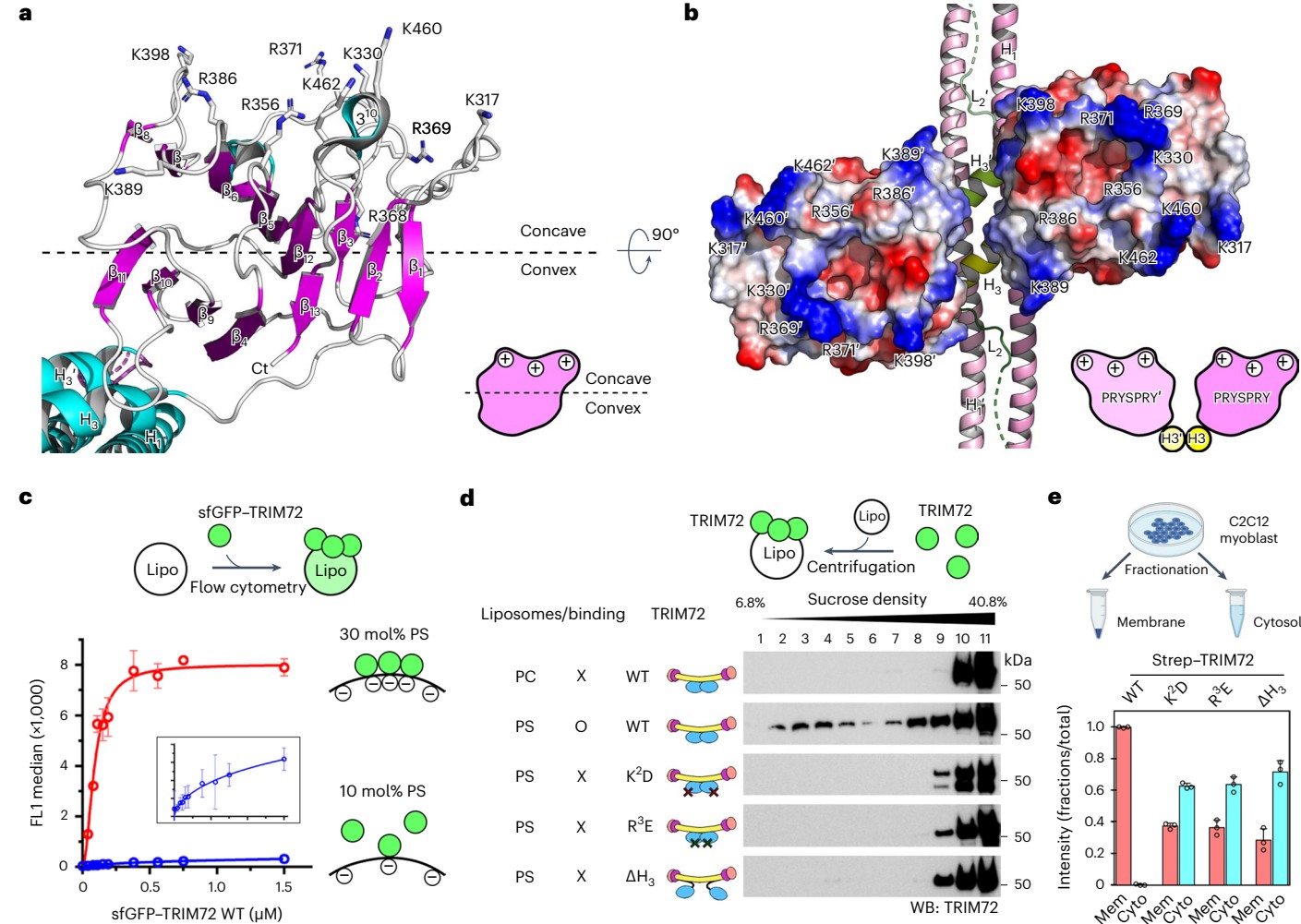

**Fig. 2 | Membrane binding of TRIM72 requires a pair of PRYSPRY domains.**
**a**, Ribbon structure of the PRYSPRY domain of TRIM72. Positively charged lysine and arginine residues are shown in a stick model and labeled. Ct indicates carboxyl terminus. **b**, Electrostatic potential surfaces of PRYSPRY domains. The positively charged surface of each PRYSPRY domain is maintained by the interaction between $H_3$:$H_3'$ helices (yellow). The second protomer is labeled with a prime symbol ('). **c**, Saturation curves for TRIM72 binding to PS liposomes (lipo) with PS concentrations of 30 mol% (red) and 10 mol% (blue). **d**, Liposome coflotation assay with TRIM72 WT and mutants using sucrose density gradients. The fractions collected in tubes are indicated from top to bottom. TRIM72 was detected with an anti-TRIM72 antibody. PC, phosphatidylcholine-containing liposomes; PS, PS liposomes (30 mol% PS). WB, western blot. X and O indicate non-binding and binding, respectively. **e**, Subcellular fractionation of TRIM72 in C2C12 myoblasts. Membrane (Mem) and cytosol (Cyto) fractions are indicated. The experimental scheme (top) was created with BioRender.com. Western blotting results are shown in Supplementary Fig. 4. Independent experiments were performed in triplicate. Data are presented as mean values with error bars representing s.d. Details of the mutants used in the experiments are provided in Supplementary Table 1.

physiological conditions, TRIM72 binds to the lipid membrane using its positively charged PRYSPRY domains. In conclusion, our results demonstrate the importance of the orientation, avidity and charge distribution of the PRYSPRY domains in recognizing negatively charged phospholipid domains.

**TRIM72 forms a higher-order assembly on the membrane**
TRIM72 is known to oligomerize in an oxidation-dependent manner and is critical for vesicle nucleation to facilitate rapid membrane repair[18]. Under oxidizing conditions by the addition of $H_2O_2$, oligomerized TRIM72 was observed only in PS liposomes (Supplementary Fig. 5a). This result indicates that TRIM72 self-oligomerizes on the negatively charged phospholipid membrane. To further characterize the multimerization of TRIM72, we treated TRIM72 with various sulfhydryl- and amine-reactive cross-linkers in the presence or absence of PS liposomes (Supplementary Fig. 5b). Higher-order TRIM72 oligomers were observed under conditions in which they were bound to PS liposomes. Time-course cross-linking analysis showed that TRIM72 WT

gradually oligomerized over time, but the mutants ($K^2D$, $R^3E$ and $\Delta H_3$) produced fewer higher-order oligomers because they were unable to bind negatively charged PS liposomes (Extended Data Fig. 5). These findings suggest that TRIM72 binds to the negatively charged phospholipid membrane and then oligomerizes into higher-order assemblies.

Interestingly, human TRIM72 purified from mammalian cells cross-linked much faster than mouse TRIM72 from bacterial cells, likely due to its prior binding to endogenous membranes in mammalian cells (Extended Data Fig. 6a). In fact, during affinity chromatography (AC) and SEC, mammalian-expressed human TRIM72 was copurified with microvesicles that were approximately 100 nm in diameter (Extended Data Fig. 6b,c). These microvesicles were identified as originating from extracellular vesicles or junctional membranes and were enriched with TRIM72 and $Na^+$–$K^+$ ATPase (Supplementary Data 1). When microvesicles were visualized by transmission electron microscopy (TEM) after negative staining, long stripe patterns were observed on the microvesicles (Extended Data Fig. 6c). To confirm whether the patterns originated from oligomerized TRIM72, we

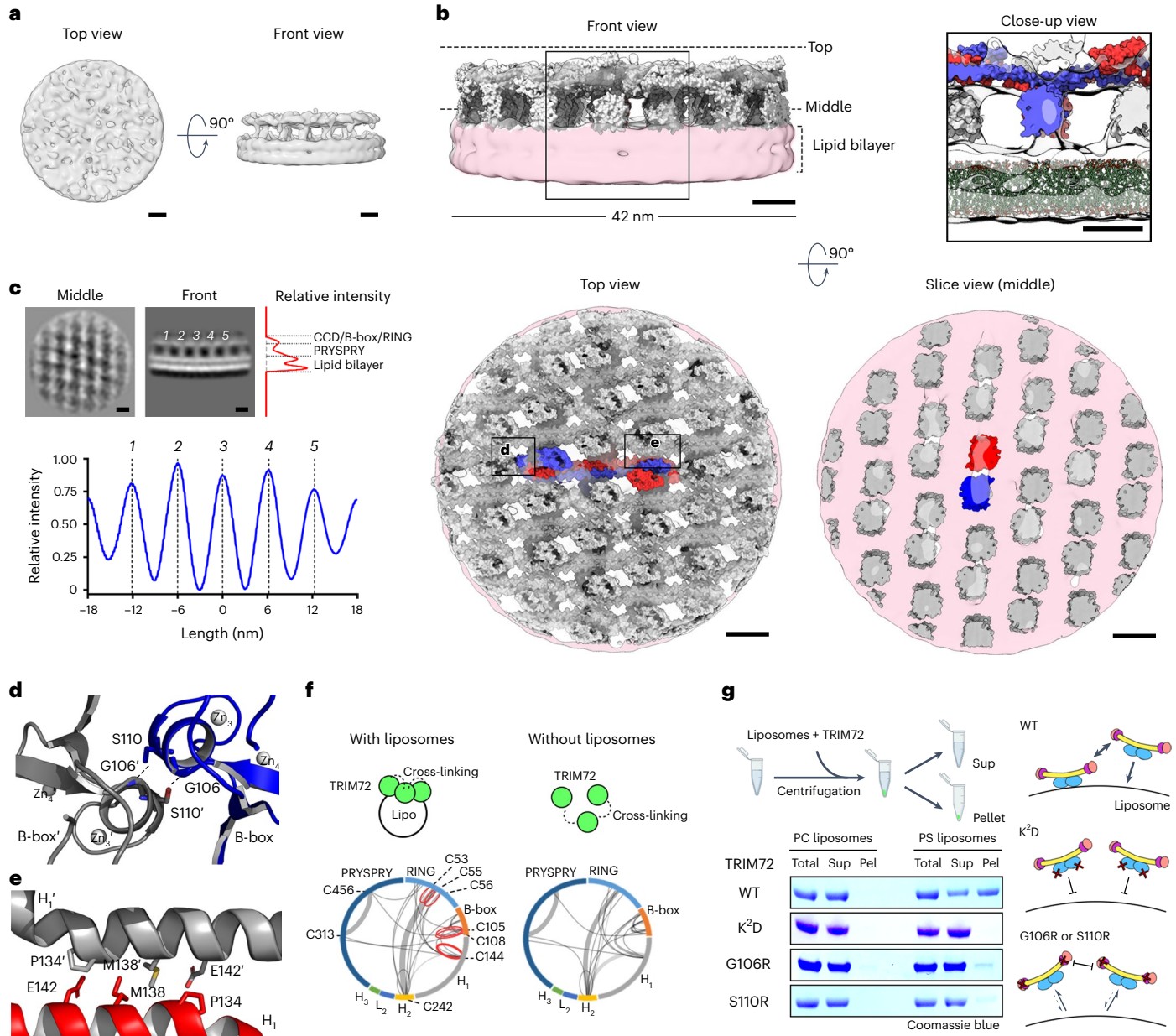

**Fig. 3 | Higher-order TRIM72 assembly on the lipid bilayer. a**, Cryo-ET maps of the reconstituted TRIM72 proteoliposome. Top view (left) and front view (right) are shown with subtomogram averaging reconstruction density contoured at 1 σ. **b**, Higher-order TRIM72-assembly model on the lipid bilayer. Top left, front view (perpendicular to the lipid surface) of the TRIM72-assembly model with a subtomogram averaging map contoured at 1 σ. Top right, the black box in the front view is enlarged to a close-up view. The top view (parallel to the lipid surface, bottom left) and a sliced view (bottom right) of the TRIM72 assembly. The black boxes in the top view are represented as ribbon diagrams in **d**,**e**. The highlighted TRIM72 protomers in the dimer are colored red and blue, respectively. The lipid bilayer models are shown in stick representation in green (top right). Subtomogram averaging reconstruction maps of the TRIM72 assembly and lipid bilayer are colored in white and pink, respectively. **c**, Assembly pattern of TRIM72 as observed in subtomogram averaging reconstruction.

Averages of *z* slices correspond to the middle and front view in **b**. The relative intensities according to the *y* axis (right; red) and the *x* axis (bottom; blue) were calculated from the averaged *z* slices of the front view. Numbers in italics (*1–5*) indicate each column-like density corresponding to the PRYSPRY domain. Note that the regular intervals between the PRYSPRY densities are 6 nm (bottom). Black scale bars, 5 nm in **a–c**. **d**, Interface between B-boxes. Interacting residues and zinc ions are shown in stick and sphere models, respectively. **e**, Interface between H₁ helices in the CCDs. **f**, Circle plots of cross-linked cysteine residues of TRIM72 with or without PS liposomes. Frequently identified cross-linked pairs are shown as thick lines. Homotypic cysteine-bridged pairs are shown in red and were identified in the presence of PS liposomes. **g**, Liposome co-sedimentation assay of TRIM72 WT and mutants. The experimental scheme is shown at the top; the image was created with BioRender.com. PC liposomes, phosphatidylcholine-containing liposomes; pel, pellet; sup, supernatant.

reconstituted TRIM72 proteoliposomes in vitro and characterized them. TRIM72 proteoliposomes were successfully isolated by SEC after incubation with TRIM72 and small PS liposomes approximately 100 nm in diameter and showed an SEC profile similar to that of TRIM72-bound

microvesicles from mammalian cells (Extended Data Fig. 6b). In the negative-staining TEM images, long stripe patterns were also observed on the reconstituted proteoliposomes, which were similar to those found on microvesicles (Extended Data Fig. 6d).

To investigate how TRIM72 oligomerizes on the phospholipid membrane, we analyzed TRIM72 proteoliposomes using cryogenic electron tomography (cryo-ET) (Extended Data Fig. 6e and Supplementary Videos 1 and 2). We observed higher-order TRIM72 assembly on both surfaces of the lipid bilayer, with the inner surface being better resolved than the outer surface. The TRIM72 assembly exhibited vertical column-like densities standing on the lipid membrane at regular intervals and a unique bridge-like structure connecting these column densities horizontally.

## Molecular model of the higher-order TRIM72 assembly

To obtain a detailed model of TRIM72 assembly on the phospholipid membrane, we averaged subtomograms containing TRIM72-repeating units from tomograms of TRIM72 proteoliposomes. This allowed us to generate a density map of the higher-order TRIM72 assembly complexed with the lipid bilayer at an overall resolution of 25 Å (Fig. 3a,b, Extended Data Fig. 7 and Table 2). The TRIM72 assembly appeared in a regular pattern with 6-nm spacing on the lipid bilayer, composed of vertical column densities and a planar density spanning the region between them (Fig. 3c and Extended Data Fig. 8c). Double column densities connected the planar density above the lipid bilayer, forming the characteristic bridge-like structure of the TRIM72 assembly. This structure was observed in detail from the view perpendicular to the lipid surface (Fig. 3b,c).

Using X-ray crystallography and subtomogram averaging, we investigated the molecular architecture of the higher-order TRIM72 assembly complexed with the phospholipid bilayer. We were able to fit approximately 18 dimers of TRIM72 as a rigid body on the phospholipid bilayer with an area of 1,400 nm$^2$ approximately with the guidance of crystal packing (Fig. 3b). The PRYSPRY domains are located on the double column-like density, while other domains, such as CCDs, B-boxes and RINGs, occupy the planar density above the lipid bilayer. The assembly allows for the positively charged surfaces of the PRYSPRY domains to be in close proximity to the surface of the phospholipid bilayer (Fig. 3b).

There are two interactions between each dimer in the TRIM assembly. The major interaction is between B-boxes, which mediate dimer-of-dimer formation through Gly106 and Ser110′ interacting with each other by hydrogen bonding (Fig. 3b,d and Supplementary Fig. 6a,b). The interaction was confirmed by cross-linking mass spectrometry (CLMS), in which cross-linked residues between the B-boxes (Cys105–(cross-linker)–Cys105′ and Cys105–(cross-linker)–Cys108′) were detected only in the presence of PS liposomes (Fig. 3f and Supplementary Fig. 7a). Mutations at the B-box interface (TRIM72$^{G106R}$ and TRIM72$^{S110R}$) resulted in defective membrane binding and assembly (Fig. 3g).

CCD-mediated interactions also play a crucial role in ensuring that the PRYSPRY domains are aligned in the same orientation (Fig. 3b,e). These CCD interactions are organized by a hydrophobic interface mediated by Met138, which is exposed on the outer side of the H$_1$ helix in the CCD. Mutations at the Met138 site (TRIM72$^{M138A}$) resulted in a severe defect in higher-order TRIM72 assembly on PS liposomes, despite maintaining membrane binding (Extended Data Fig. 8a,b). However, the TRIM72$^{M138R}$ variant had a higher binding affinity to PS liposomes and clearer patterns of TRIM72 assembly than TRIM72 WT (Extended Data Fig. 8c and Supplementary Videos 3 and 4). Subtomogram averaging of TRIM72$^{M138R}$ revealed that the density map of the TRIM72$^{M138R}$ assembly was similar to that of TRIM72 WT (Extended Data Fig. 8d–g). In addition, cross-linked residues (Cys144–(cross-linker)–Cys144′) were detected near the interface between CCDs, supporting the observation of proximity between CCDs (Fig. 3f and Supplementary Fig. 7a).

The distance between cross-linked cysteines identified by CLMS was used to validate the model of higher-order TRIM72 assembly. The TRIM72-assembly model showed minimized distances between cross-linked cysteines, whereas the dimer or dimer-of-dimer models

## Table 2 | Cryo-ET data collection

| | Mouse TRIM72$^{WT}$–proteoliposome (EMD-33569) | Mouse TRIM72$^{M138R}$–proteoliposome (EMD-33582) |
|---|---|---|
| **Data collection and processing** | | |
| Microscope | FEI Titan Krios G2 | FEI Titan Krios G2 |
| Voltage (kV) | 300 | 300 |
| Detector | Falcon 3EC | Falcon 3EC |
| Total electron dose (e–/Å$^2$) | ~52 | ~52 |
| Electron dose per tilt image (e–/Å$^2$) | 1.4 | 1.4 |
| Number of frames per tilt image | 37 | 37 |
| Defocus range (μm) | −4.0 to −6.0 | −4.0 to −6.0 |
| Pixel size (Å) | 2.300 | 2.300 |
| Tilt range | −54°/54° | −54°/54° |
| Number of tomograms | 1 | 2 |
| Map resolution (Å) | ~25 | ~26 |
| FSC threshold | 0.143 | 0.143 |
| Number of proteoliposomes | 1 | 2 |
| Number of initial subtomograms | 18,715 | 9,570 |
| Number of final subtomograms | 12,324 | 4,033 |
| Symmetry | C1 | C1 |
| Map sharpening B factor (Å$^2$) | −64.1 | −55.3 |

showed outliers above 50 Å (Supplementary Fig. 7b and Supplementary Data 2). In conclusion, by integrating structural and biochemical analyses, we determined the molecular model of the higher-order TRIM72 assembly complexed with the negatively charged phospholipid bilayer. The TRIM72 assembly aligns the PRYSPRY domains on the surface of the phospholipid membrane. Furthermore, intermolecular domain interactions cooperatively affect TRIM72 assembly to maintain tighter binding with the negatively charged phospholipid membrane.

## Ubiquitination activity of TRIM72 is enhanced on the membrane

Screening of the E2 conjugating enzyme revealed that TRIM72 exhibits ubiquitination activity with ubiquitin-conjugating enzyme E2 D (UBE2D) enzymes (Supplementary Fig. 8a). However, TRIM72 expressed in bacterial cells, even with duplicated RING domains (2×RING), did not show detectable ubiquitination activity (Supplementary Fig. 8b), in contrast to TRIM72 expressed in mammalian cells (Supplementary Fig. 8a). We also checked the phosphorylation status of these TRIM72 proteins but found no difference between them (Supplementary Fig. 8c). Hence, we investigated whether the ubiquitination activity of TRIM72 could be enhanced by phospholipid membrane binding. We reconstituted TRIM72 proteoliposomes and evaluated their ubiquitination activity. Indeed, ubiquitination activity increased only in reconstituted TRIM72 proteoliposomes, not in TRIM72 alone or liposomes alone (Fig. 4a). This indicates that the ubiquitination activity of TRIM72 is suppressed in solution. To investigate how ubiquitination activity is suppressed, we focused on the residue next to the last zinc-coordinating cysteine, called the 'linchpin'. The linchpin is critical to Ub transfer because it forms hydrogen bonds with both the main-chain carbonyl oxygen atoms of E2 enzymes (Gln92 in UBE2Ds and Lys94 in UBE2N) and Ub (Arg72)[55,56]. While the canonical linchpin is an arginine residue, the corresponding residue in TRIM72 is a suboptimal glutamine residue

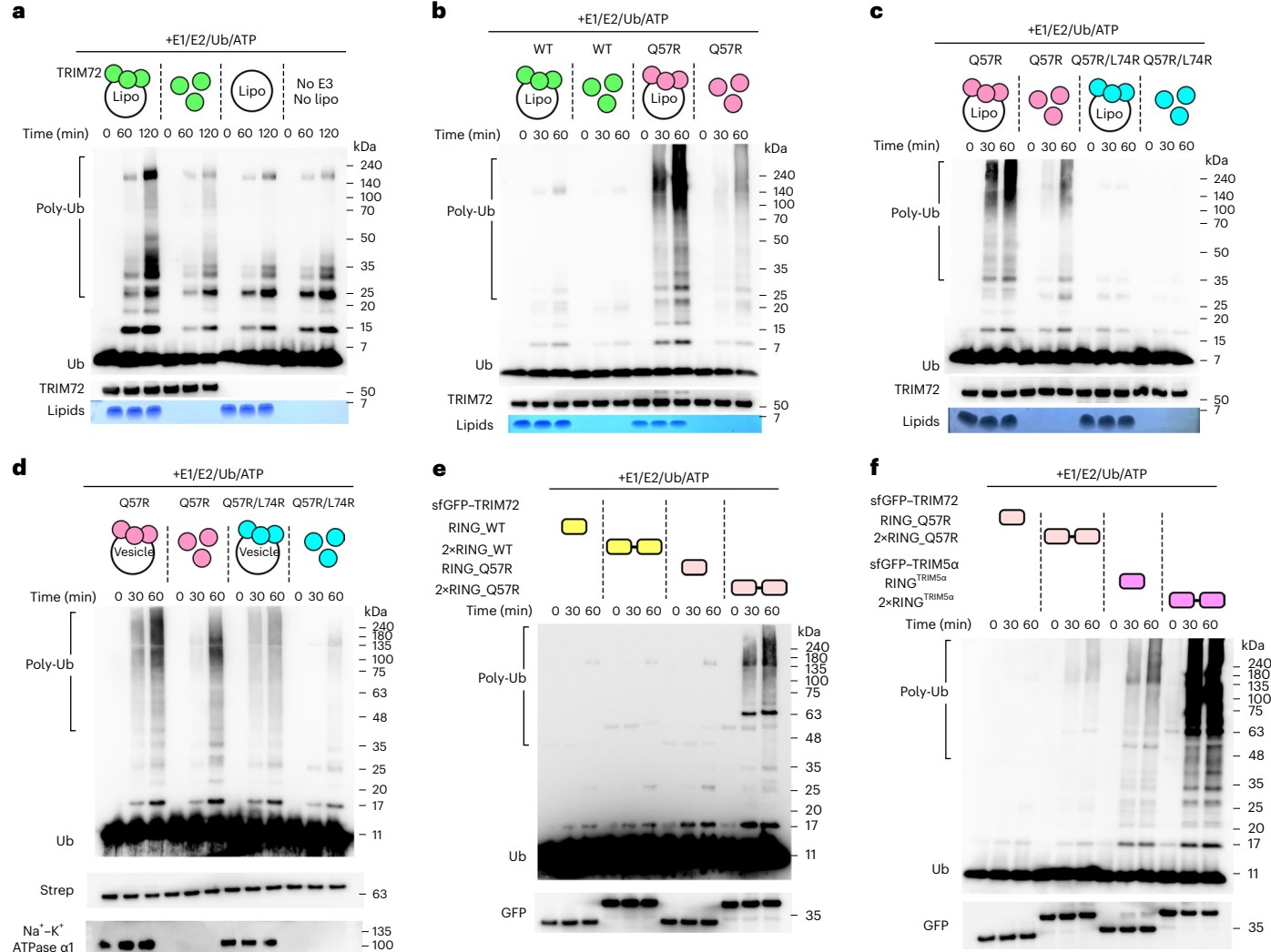

**Fig. 4 | Ubiquitination activity of TRIM72. a**, Ubiquitination activity of TRIM72 in the presence or absence of PS liposomes. **b**, Increased ubiquitination activity of TRIM72$^{Q57R}$ in the presence or absence of PS liposomes. **c**, Ubiquitination activity of TRIM72$^{Q57R}$ and TRIM72$^{Q57R/L74R}$ in the presence or absence of PS liposomes. **d**, Ubiquitination activity of endogenous microvesicle-bound or free-solution states of TRIM72$^{Q57R}$ and TRIM72$^{Q57R/L74R}$. **e**, Ubiquitination activity of RING and 2×RING constructs with TRIM72 WT and TRIM72$^{Q57R}$. **f**, Ubiquitination activity of RING and 2×RING constructs with TRIM72$^{Q57R}$ and TRIM5α. TRIM72 proteoliposomes were reconstituted and further separated by SEC (**a–c**). TRIM72 microvesicles were purified from HEK293T cells and isolated by SEC (**d**). Poly-Ub is represented as a polyubiquitination ladder. Bands close to 140 kDa were assumed to be ubiquitin-like modifier activating enzyme 1 (UBA1)-Ub. Ub was detected with an anti-Ub antibody. Strep-tagged TRIM72 or intact TRIM72 was detected using an anti-Strep-tag antibody or an anti-TRIM72 antibody, respectively (**a–d**). The superfolder variant of green fluorescent protein (sfGFP)-fused RING and 2×RING constructs were detected with an anti-GFP antibody (**e,f**). Phospholipids were stained with Sudan Black B (**a–c**). Microvesicles were detected with an anti-Na⁺–K⁺ ATPase α1 antibody (**d**). Independent experiments were performed in triplicate. TRIM72 WT, green; TRIM72$^{Q57R}$, pink; TRIM72$^{Q57R/L74R}$, cyan; TRIM72 RING_WT, yellow; TRIM72 RING_Q57R, light pink; TRIM5α RING, magenta.

(Gln57). We mutated the glutamine linchpin to arginine (Q57R) and then tested the ubiquitination activity. The TRIM72$^{Q57R}$ mutant exhibited much higher ubiquitination activity than the TRIM72-WT protein (Fig. 4b). The membrane-bound TRIM72$^{Q57R}$ mutant also showed much higher ubiquitination activity than the soluble TRIM72$^{Q57R}$ mutant, although soluble TRIM72$^{Q57R}$ exhibited partially restored ubiquitination activity. These results clearly demonstrate that the ubiquitination activity of TRIM72 is suppressed in the soluble state due to the suboptimal linchpin. Similar to reconstituted proteoliposomes, the mammalian-expressed TRIM72$^{Q57R}$ mutant possessed highly enhanced ubiquitination activity in both fractions containing vesicle-bound and vesicle-free soluble forms (Supplementary Fig. 8d).

### Ub transfer is catalyzed by transient RING dimerization

In our crystal structures of TRIM72, the key residue Gln57 in the RING domain was observed near Arg207′ in the CCD through

the bent RING conformation (Supplementary Fig. 3a). However, when we superimposed the structure with the TRIM25 RING–UBE2D2–Ub ternary complex[43], we found that the bent RING conformation was not accessible to E2–Ub intermediates due to steric hindrance with the CCDs. To further explore the conformation of the RING domain, we used the SAXS refinement through flexibility (SREFLEX) approach (Supplementary Fig. 9a–c; statistics of the SAXS analysis are summarized in Supplementary Table 2)[57]. Based on the experimental SAXS profiles, the calculated models showed that the RING domain is highly dynamic and does not prefer the bent conformation (Supplementary Fig. 9a). We also observed that the electron density map of the RING domain appears in only one of the four crystal structures of TRIM72 FL, suggesting that the RING domain is intrinsically flexible. Additionally, an AlphaFold[58] prediction suggested that TRIM72 has the extended conformation of the RING domain, similar to one of the models calculated by SREFLEX. Not only

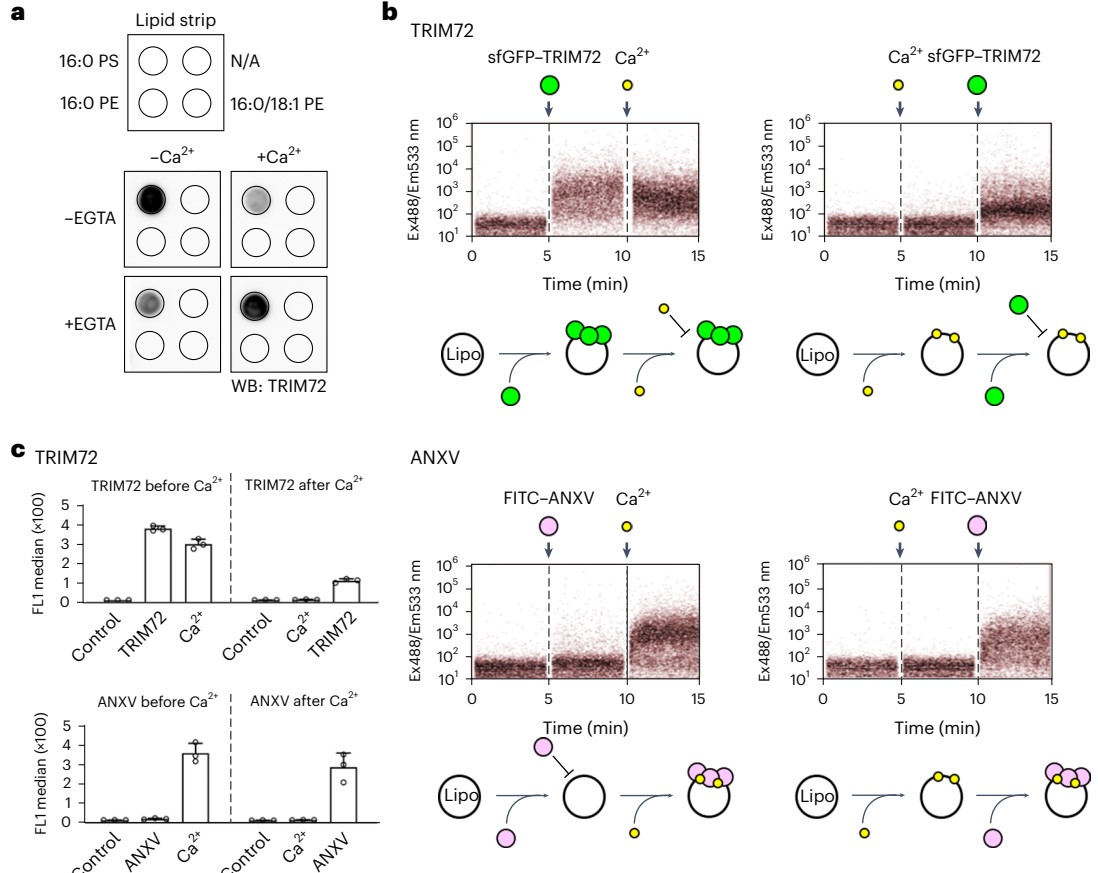

**Fig. 5 | Effect of Ca²⁺ on membrane binding of TRIM72. a**, Protein–lipid overlay analysis showing the interference. TRIM72 proteins were incubated with lipid strips in the presence or absence of Ca²⁺ and/or EGTA and then probed with an anti-TRIM72 antibody. PE, phosphatidylethanolamine. N/A indicates phospholipid is not applied. **b**, Flow cytometry analysis comparing the effects of Ca²⁺ on TRIM72 and annexin V (ANXV). PS liposomes were preloaded with Ca²⁺ or loaded after Ca²⁺ exposure and then incubated with TRIM72 or annexin

V. Binding was detected using fluorescently labeled sfGFP–TRIM72 and fluorescein isothiocyanate (FITC)–annexin V. Ex488/Em533 nm, excitation at 488 nm/emission at 533 nm. **c**, Quantification of the data in **b**. Experiments were performed in triplicate. Data are presented as mean values ± s.d. The PS concentration was 30 mol% in PS liposomes. Ca²⁺ ion, yellow; TRIM72, green; annexin V, pink.

structural evidence but also biochemical data indicated that there was no difference in ubiquitination activities between the TRIM72^Q57R and TRIM72^Q57R/R207E mutants (Supplementary Fig. 9d). Therefore, we conclude that the RING domain is highly flexible in both the TRIM72 dimer and oligomer on the phospholipid membrane.

Next, we investigated whether dimerization of the RING domain is necessary for Ub transfer. In contrast to the TRIM72 RING domains in the dimer model, which are approximately 20 nm apart, the RING domains are located close to each other in the higher-order assembly model on the membrane (Fig. 3b). This suggests that TRIM72 RING domains transiently dimerize between dimeric TRIM72 and another dimeric TRIM72 in the oligomeric assembly. To determine whether dimerization of the RING domains is essential for Ub transfer, we introduced a mutation to disrupt the dimeric interface on the RING domain. Structural comparison with the TRIM25 RING dimer complexed with UBE2D2–Ub[43] predicted that Leu74 of TRIM72 (corresponding to Val72 in TRIM25) is critical to the dimerization of the RING domain. Thus, we replaced Leu74 of TRIM72 (L74R) with a bulky and hydrophilic arginine residue to block the dimeric interface. Because the WT activity was weak, we compared the ubiquitination activity of TRIM72^L74R with that of the TRIM72^Q57R-mutant background. In contrast to the highly active TRIM72^Q57R mutant, TRIM72^Q57R/L74R completely lacked activity in both membrane-bound and soluble forms (Fig. 4c). Furthermore, we observed similar results not only for the reconstituted TRIM72

proteoliposomes but also for mammalian-expressed TRIM72 bound to endogenous microvesicles (Fig. 4d), suggesting that TRIM72 RING domains dimerize and activate in the higher-order assembly on the membrane. Finally, we dissected the dimerization effect of the RING domain without other domain interactions of TRIM72 by carrying out the ubiquitination assay using TRIM72 RING and 2×RING based on the TRIM72^Q57R background. In contrast to the low ubiquitination activities of TRIM72 RING and 2×RING, as previously shown, TRIM72 2×RING_Q57R showed significantly increased activity compared to that of monomeric TRIM72 RING_Q57R (Fig. 4e). This finding is similar to the dimeric activation mode of TRIM5α RING, although the activity of TRIM72 2×RING_Q57R was still lower than that of TRIM5α RING (Fig. 4f). These results suggest that RING dimerization is necessary to activate the mechanism of TRIM E3 Ub ligase. Therefore, we conclude that the intermolecular RING domains can transiently dimerize in the higher-order TRIM72 assembly on the phospholipid membrane for Ub transfer.

**TRIM72 assembles on the membrane to overcome Ca²⁺ influx**

In addition to its ubiquitination activity, we investigated the effect of Ca²⁺ on the interaction between TRIM72 and PS. Protein–lipid overlay analysis revealed that Ca²⁺ inhibited the interaction between TRIM72 and PS (Fig. 5a and Extended Data Fig. 3). However, the interaction was completely restored by ethylene-bis(oxyethylenenitrilo)tetraacetic

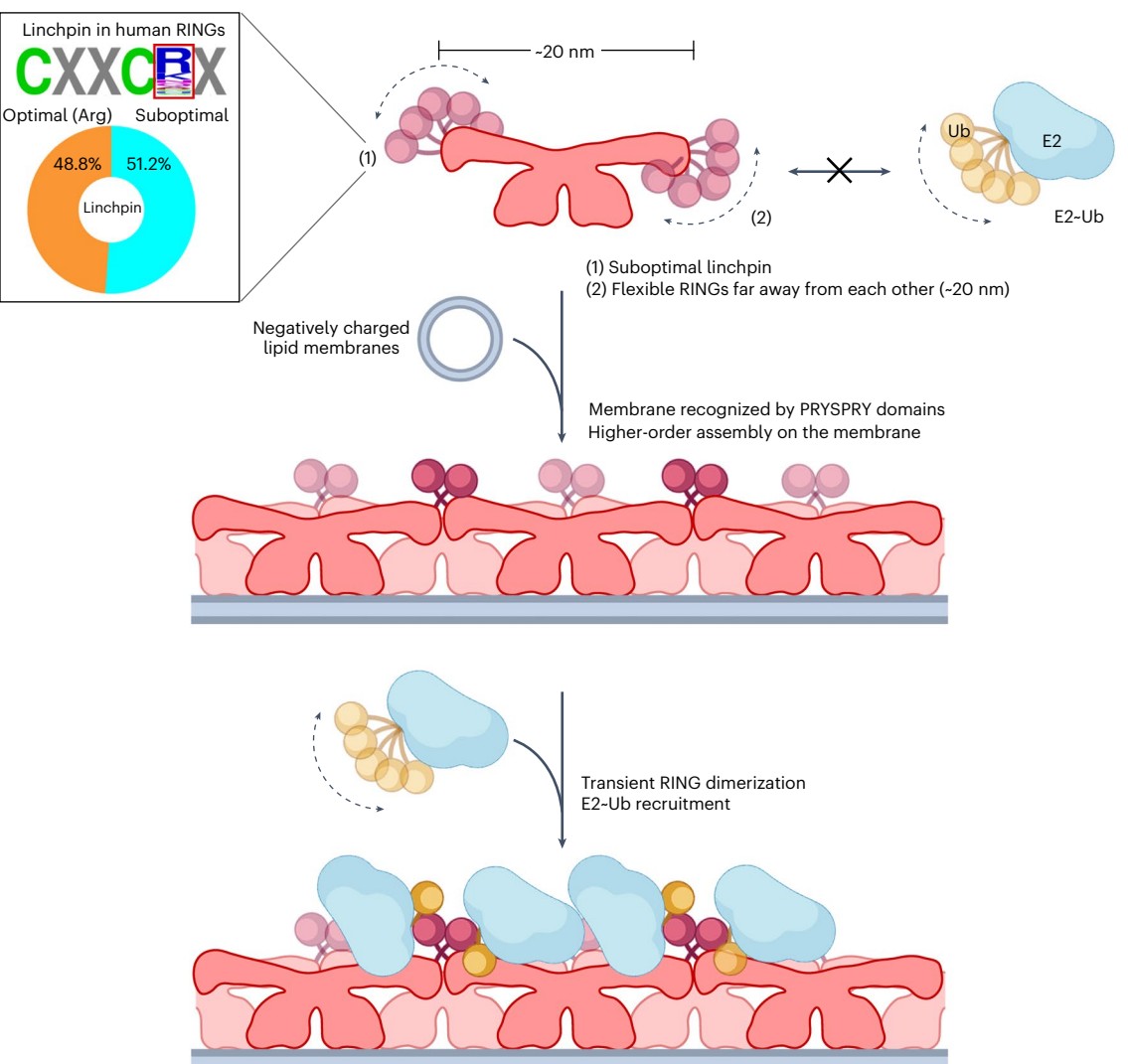

**Fig. 6 | Proposed working model of TRIM72 activation on the membrane.** In the solution state (top), TRIM72 (rose gold) cannot recruit E2–Ub (cyan, yellow) conjugates for Ub (yellow) transfer. The flexibility of the RING domain (pink) at both ends of the dimeric TRIM72 and dynamic Ub-conjugating E2 enzymes is indicated by curved dashed lines with arrows. Although B-boxes and CCDs at both ends are also moved in the perpendicular direction, this is not depicted in this model for clarity. The amino acid frequency of the linchpin is shown in a black-lined box. The linchpin, positioned at the residue following the final Zn²⁺-coordinated cysteine, was analyzed in 303 sequences of human RING domains (Supplementary Data 3). Note that ~50% of RING domains have a suboptimal linchpin, including TRIM72 (glutamine linchpin). The higher-order TRIM72

assembly on the negatively charged membrane (middle) and the active-state model with E2–Ub conjugates (bottom) are shown. When dimeric TRIM72 is targeted to PS-containing vesicles or PS-enriched plasma membranes, it primarily binds to the membrane via the PRYSPRY domain. Next, higher-order TRIM72 assembly is mediated by cooperative interactions among B-box–B-box and CCD–CCD. In this assembly, dynamic RING domains reduce the motion, and intermolecular RING dimer formation occurs. The next layered TRIM72 oligomer in the assembly is shown in a less-vibrant color. Finally, the functional ternary complex is stable enough for efficient Ub transfer. The model was created with BioRender.com.

acid (EGTA), a Ca²⁺-chelating agent (Fig. 5a), indicating that Ca²⁺ directly inhibits the interaction between TRIM72 and PS. Our structural and biochemical evidence suggests that the positively charged PRYSPRY domains of TRIM72 and Ca²⁺ compete with the negatively charged head group of PS via ionic interactions. This finding partially conflicts with a previous report that TRIM72 facilitates vesicle delivery to the damaged site in a Ca²⁺-independent manner[18]. We carefully assessed the inhibitory effect of Ca²⁺ by adding each factor one by one. Real-time flow cytometry analysis demonstrated that TRIM72 could not bind to PS liposomes in the presence of Ca²⁺ (Fig. 5b,c). Surprisingly, preloaded TRIM72 barely dissociated from PS liposomes upon addition of Ca²⁺. By contrast, in annexin, a negatively charged loop engages in a Ca²⁺-mediated interaction with the head group of PS[59]. This indicates that oligomerization acts as a secondary mode of interaction,

strengthening the interaction between TRIM72 and PS-rich domains on the phospholipid membrane. In conclusion, higher-order TRIM72 assembly on the membrane surface, rather than the dimeric form in solution, can overcome Ca²⁺ influx and is recruited, similar to repair vesicles, to damaged sites of the plasma membrane for wound healing.

## Discussion

In this study, we demonstrate the assembly of the RING-type E3 ligase TRIM72 into functional oligomers on phospholipid membranes (Fig. 6). Our findings showed that the ubiquitination activity of TRIM72 is suboptimal in solution, and the active conformation is induced by higher-order assembly on the membranes. This event is mediated by the interaction between the negatively charged phospholipid membranes supplied by PS in cells and the positively charged surfaces of

PRYSPRY domains oriented in the same direction. As a result, dimeric TRIM72 aligns on the membrane and further self-oligomerizes into a higher-order assembly. The conformation of the higher-order TRIM72 assembly allows RING domains to transiently homodimerize. Finally, E2-Ub intermediates (the tilde denotes a covalent thioester bond) can be recruited into dimerized RINGs of the higher-order TRIM72 assembly on the membrane for efficient Ub transfer.

There are two reasons why the TRIM72 dimer is unsuitable for Ub transfer with UBE2D-Ub intermediates in solution. First, the RING domain possesses a suboptimal linchpin, glutamine in TRIM72 (Gln57). In contrast to the optimal linchpin, which is conserved as arginine in half of RING-type Ub ligases[55] and mutated in human cancers (Fig. 6 and Supplementary Data 3)[60,61], the suboptimal linchpin is insufficient as a hydrogen bonding donor to stimulate flexible UBE2D-Ub or UBE2N-Ub intermediates into the closed conformation for robust Ub transfer[56,62]. We showed that changing the linchpin to the optimal arginine significantly enhanced the ubiquitination activity of TRIM72 both on the membrane and in solution. This finding indicates that the RING domain of TRIM72, due to the suboptimal linchpin, is insufficient to bind a closed conformation of the UBE2D-Ub intermediate. To obtain appropriate binding affinity with E2-Ub, it is essential to dimerize RINGs in known TRIM proteins[43,49,63,64]. However, the second reason is that RINGs are separated by antiparallel CCDs and laid in monomeric conformations[43,49,63,64]. Considering the structural prediction obtained with ColabFold[65] and experimental evidence supporting the RING dimer, most of the 60 TRIMs (except TRIM15, TRIM40, TRIM45 and class VI TRIMs) have the potential to possess dimerized RINGs in a similar manner (Supplementary Data 4). Therefore, in contrast to the relatively diverse linchpin mechanism, RING dimerization is likely a more general mechanism for the active conformation in the TRIM family[43,63,64,66].

We highlight the active conformation induced by RING dimerization in higher-order TRIM72 assembly on the membrane. The TRIM72 assembly is cooperatively derived from the intermolecular B-box interaction and the CCD interaction, leading to close positioning of the RING domains to each other and further dimerization despite very weak affinity in solution. Compared to the higher-order assembly of primate TRIM5α, which binds and restricts human immunodeficiency virus capsids[67,68], B-box-mediated interactions are found in the assembly process of both TRIMs. Furthermore, the proximity required to induce RING dimerization is achieved through the interaction between the B-boxes in both TRIMs. Thus, B-box-mediated assembly for functional activation in TRIMs might be a common process. On the other hand, in contrast to the hexagonal and pentagonal assembly of TRIM5α, the bridge-like pattern of TRIM72 assembly is mediated by the interaction between the solvent-exposed hydrophobic surfaces of CCDs, indicating that the structural features of TRIM assembly might vary depending on the target molecules and their interactions. In addition to TRIM72 and TRIM5α, a subset of class IV TRIMs, including TRIM65, potentially oligomerize on flexible double-stranded RNA (dsRNA) filaments through avidity-driven binding between their PRYSPRY domains and the RNA helicase domains of cognate retinoic acid-inducible gene I (RIG-I)-like receptors[50]. Recent cryo-electron microscopy (EM) analysis of TRIM72 and the crystal structure of TRIM72 ΔRING also revealed that the CCD ends are dynamic[69,70]. These results suggest that the flexibility of long TRIM CCDs is suitable for recognizing curved cellular structures, such as phospholipid membranes and viral capsids, and filamentous structures, such as dsRNA and DNA. The current findings provide a conceptual basis for the study of TRIM proteins. Compared to other TRIM PRYSPRY domains, which have been well studied for their interactions with their binding molecules[36,50,51,68], these PRYSPRY domains use the concave surface[52] to recognize their interacting molecules (Extended Data Fig. 2a–f). The crucial residues for binding are located on VL1, VL3–β₆–VL4 and VL6 (Extended Data Fig. 2g), which suggests that substrate specificity is determined by sequence diversity around the VL regions. In contrast to other PRYSPRY domains, TRIM72 PRYSPRY domains show more distributed positively charged residues on the VL regions, which contributes to their unique basic properties for membrane binding (Fig. 2a,b and Extended Data Fig. 2a).

TRIM72's interaction with the membrane involves spatially controlled ubiquitination activity in muscle differentiation and membrane-repair processes. Our findings suggest that higher-order TRIM72 assembly is essential for its membrane-repair function, as previously reported in muscle injury[18]. We have elucidated the molecular basis of TRIM72 oligomerization, including the preference for $Ca^{2+}$-independent PS binding, key residues for oligomerization, the architecture of the higher-order TRIM72 assembly and activation of E3 ligase activity. In the context of membrane repair, the ESCRT machinery plays a vital role, which is closely related to Ub signaling[3,8,9,14]. Based on our observations, we speculate that self-ubiquitination or the ubiquitination of target substrates may contribute to the membrane-repair process[3]. However, further investigation is needed to understand the links between ubiquitination and membrane-repair mechanisms. Although some features are unique to TRIM family proteins or only a subset of them, the extensive information gained from the current structural and biochemical study of TRIM72 offers broad insights into RING-type E3 Ub ligases.

## Online content

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

## Methods

### Cloning, protein expression and purification

FL mouse *Trim72* and human *TRIM72* genes were amplified by PCR and cloned into bacterial (modified pMAL and pRSF) and mammalian (modified pHM6) expression vectors, respectively. The domain-deletion and point mutant constructs were cloned by PCR-based mutagenesis. All constructs (Supplementary Table 1) were verified by DNA sequencing. WT TRIM72 and mutants with an N-terminal His$_6$–MBP tag were expressed in *Escherichia coli* BL21(DE3) cells and purified. For flow cytometry analysis, mouse TRIM72 tagged with N-terminal His$_6$–sfGFP[71] was expressed and purified. Purification of GST–H$_3$–PRYSPRY was carried out as previously described[37]. Cells were induced at an OD$_{600}$ of 0.7 with 500 μM isopropyl β-D-1-thiogalactopyranoside and grown with 200 μM ZnCl$_2$ (depending on the presence of DNA encoding the RING and B2-box domains) at 18 °C for 18 h. Proteins were purified by AC followed by ion exchange (IEX) and SEC with equilibrated SEC buffer (50 mM Tris-HCl, pH 8.0, 300 mM NaCl and 1 mM Tris(2-carboxyethyl) phosphine (TCEP)). The N-terminal MBP tag was cleaved by on-column tobacco etch virus protease digestion. The cleaved MBP tag and tobacco etch virus protease were removed by AC and IEX. The quality of TRIM72 WT and the mutant proteins was verified by circular dichroism and thermal shift assays. Human TRIM72 WT and mutants with an N-terminal 2× Strep tag (MASAWSHPQFEKGGGSAWSHPQFEKGS) were expressed in HEK293T mammalian cells after transfection with polyethyleneimine. After 48 h, the cells were washed with Dulbecco's phosphate-buffered saline (DPBS) and resuspended in lysis buffer (50 mM Tris-HCl, pH 8.0, 150 mM NaCl and 1 mM TCEP). After lysis with an ultrasonic probe and centrifugation (27,216g, 4 °C for 1 h), the supernatants were loaded onto a StrepTrap HP column (Cytiva, 28907548). The proteins were eluted with Strep elution buffer (50 mM Tris-HCl, pH 8.0, 150 mM NaCl, 1 mM TCEP and 2.5 mM D-desthiobiotin) and then subjected to SEC using Superose 6 Increase 10/300 GL columns (Cytiva, 29091596). Each fraction containing TRIM72 was subjected to a ubiquitination assay and EM. Rhesus TRIM5α RING and tandem repeats of RING constructs were purified using His$_6$–sfGFP tags followed by IEX. Plasmids encoding UBA1 (human UBA1, Addgene, 34965)[72] and UBE2D3 (human UBE2D3, Addgene, 15784)[73] were obtained from Addgene (http://addgene.org). E1 and E2 enzymes were purified with general bacterial expression protocols. Ub was prepared as described in previous studies[74].

### Crystallization and data collection

Mouse TRIM72 proteins (5–10 mg ml$^{-1}$) were crystallized by the vapor-diffusion method with a 1:1 mixture of crystallization reagents and proteins. Although many crystals were obtained from the initial crystallization screen at 22 °C, most of these crystals were brittle and poorly diffracted to a resolution of only 7 Å. To improve crystal quality and change crystal packing, we designed numerous constructs by modifying the residues suspected to be exposed on the surface. Among them, distinct crystal forms were obtained by crystallizing a construct containing five point mutations and omitting the N-terminal six amino acids and the C-terminal seven amino acids (Supplementary Table 1). Indeed, the primitive monoclinic crystals showed greatly improved diffraction patterns at resolutions of up to 3.5 Å. After further optimization by limited proteolysis with elastase, mouse TRIM72 ΔRING crystals grown at 4 °C diffracted up to 2.75 Å. The crystallization conditions were as follows: 0.8 M NaCl and 8% (vol/vol) ethanol (WT, 7.1 Å; constructs, highest resolution); 100 mM MES-NaOH, pH 6.5, 200 mM MgCl$_2$, 0.5% (wt/vol) polyvinylpyrrolidone K 15 and 30% (wt/vol) pentaerythritol propoxylate (5/4 PO/OH) with microseeding (FL, 4.6 Å); 100 mM HEPES-NaOH, pH 7.5, 230 mM MgCl$_2$, 100 mM CsCl and 34.16% (wt/vol) pentaerythritol ethoxylate (3/4 EO/OH) with Al's oil (FL, 3.5 Å); 100 mM HEPES-NaOH, pH 7.5, 250 mM MgCl$_2$ and 37% (wt/vol) 5/4 PO/OH (FL/C242S, 5.2 Å); 100 mM Tris-HCl, pH 8.5 and 30% (vol/vol) PEG 400 with microseeding (ΔRING, 2.75 Å); and 100 mM Tris-HCl, pH 8.5 and 8% (vol/vol) PEG 8000 (ΔRING, 3.3 Å). Crystals were flash frozen

in liquid nitrogen under cryoprotective conditions. Datasets were collected at the following synchrotron facilities: the Pohang Accelerator Laboratory in South Korea and the Photon Factory and Super Photon Ring-8 GeV (SPring-8) in Japan. Data processing, integration and scaling were carried out with DENZO and SCALEPACK from the HKL-2000 suite[75]. Detailed statistics are summarized in Table 1.

### Structure determination and refinement

The phases of mouse TRIM72 ΔRING were solved by molecular replacement using phenix.phaser[76] with human TRIM72 PRYSPRY (PDB 3KB5) as a search model[37]. The phases were improved by combining single-wavelength anomalous dispersion from two zinc ions in the B-box domain. An unassigned model of the B2-box domain and the CCD was manually built with Coot[77] and rebuilt with phenix.autobuild between refinement cycles[78]. The phases of the other crystals were solved by molecular replacement with a monomer of ΔRING as a search model. Detailed refinement statistics are summarized in Table 1. Among the FL mouse TRIM72 structures, an electron density map corresponding to the RING domain was found only in the trigonal crystals of FL and diffracted up to a resolution of 4.6 Å. In the other crystal systems, we could not find sufficient electron density for the RING domain; accordingly, we believe that the RING domain does not participate in crystal contacts in these crystal systems and that it is also very dynamic. The electron density map of the RING domain was found at the outer surface of another protomer's CCD and was separated by two bulky difference density maps. Two zinc ions were placed at the center of each separated difference map. Interestingly, the distance between the zinc ions was approximately 14 Å, the interzinc distance in the classical C3HC4-type RING domain[79]. Due to the low resolution, we initially built the RING domain coordinates by homology modeling using SWISS-MODEL[80] and then fitted the model into residual difference maps and refined it with phenix.refine[81]. The quality of the final model was validated by MolProbity[82].

### Structural and multiple-sequence analyses

Structural analysis was performed using the PISA server in the CCP4 suite[83]. Structure-based sequence alignments were conducted with PROMALS3D[84]. Sequences were aligned with Clustal Omega[85] and further analyzed with BioEdit software[86]. The logo graph was generated with WebLogo (https://weblogo.berkeley.edu/logo.cgi). Structural figures were prepared with PyMOL (Schrödinger) and UCSF ChimeraX[87].

### Small-angle X-ray scattering

Purified mouse TRIM72-WT and ΔRING proteins were prepared in SAXS buffer (25 mM Tris-HCl, pH 8.0, 300 mM NaCl, 1 mM TCEP and 1 mM dithiothreitol (DTT)). SAXS data were collected at synchrotron beamlines 4C at the Pohang Accelerator Laboratory in South Korea and 10C at the Photon Factory in Japan. For SEC–SAXS experiments, proteins were loaded onto Superdex 200 Increase 10/300 GL resin (Cytiva, 28990944) equilibrated with SAXS buffer, after which datasets were collected from peak fractions and normalized buffer fractions for background subtraction. Data processing was performed using the ATSAS software package[88]. For flexible refinement of the TRIM72-WT model, two RING domains of dimeric TRIM72 FL were defined as individual rigid bodies and fitted to experimental curves for mouse TRIM72 WT using SREFLEX software in ATSAS online[57]. The final models were superimposed on 25 refined models. Details of data collection and parameters are summarized in Supplementary Table 2.

### Size exclusion chromatography with multiangle laser light scattering

SEC–MALS was performed using an ÄKTA FPLC system (Cytiva) coupled with a Wyatt miniDAWN TREOS detector (Wyatt Technology). Protein was loaded on a Superdex 200 Increase 10/300 GL column equilibrated with MALS buffer (50 mM Tris-HCl, pH 8.0, 300 mM NaCl and 2 mM TCEP).

Ovalbumin (Sigma-Aldrich, A7641) was used as an isotropic scatterer for calibration and normalization of the detectors. Light scattering was measured and analyzed using ASTRA V software (Wyatt Technology).

## In vitro ubiquitination assay

An in vitro E2 screening assay was performed using 50 nM UBA1, 500 nM of each E2 enzyme in a set (Boston Biochem, K-980B) and 10 μM Ub with 2 μM mouse TRIM72 constructs in ubiquitination buffer (50 mM Tris-HCl, pH 8.0, 150 mM NaCl and 1 mM TCEP) at 37 °C for 1 h. In vitro ubiquitination reactions were performed with 50 nM UBA1, 2 μM UBE2D3 and 3 μM Ub with 1 μM mouse TRIM72, rhesus TRIM5α or 0.2 μM human TRIM72. In the ubiquitination assay with TRIM72 mutants based on the Q57R background, 0.5 μM UBE2D3 was used instead of 2 μM UBE2D3 to assess the mutation effects of TRIM72. All reactions were initiated by the addition of 5 mM ATP and $MgCl_2$. The reaction was stopped by the addition of sodium dodecyl sulfate (SDS) sample buffer containing β-mercaptoethanol (β-ME). The samples were boiled at 95 °C for 10 min and then resolved by SDS–PAGE for further analysis. The polyubiquitinated chains were detected by western blot analysis using the Ub-specific antibody P4D1 (Santa Cruz, sc-8017) or VU-1 (LifeSensors, VU101), while TRIM72 WT and mutants were detected using a rabbit polyclonal anti-TRIM72 (residues 90–277) antibody[19], and sfGFP-fused TRIM constructs were detected using a mouse monoclonal anti-GFP antibody (Santa Cruz, sc-9996). Phospholipids were stained with Sudan Black B (Sigma-Aldrich, 199664). TRIM72-bound microvesicles were detected with a mouse monoclonal anti-$Na^+$–$K^+$ ATPase α1 antibody (Santa Cruz, sc-21712).

## Protein–lipid overlay assay

The concept of the protein–lipid overlay assay has been described previously[89]. The following lipid strips were used to screen the lipid specificity of TRIM72: PIP Strips (Thermo Fisher Scientific, P23750), Inositol Snoopers (Sigma-Aldrich, 330500), Oxidized Phospholipid Snoopers (Sigma-Aldrich, 330501) and Sphingolipid Snoopers (Sigma-Aldrich, 330503). The lipid strips were blocked with 3% (wt/vol) bovine serum albumin (BSA; A0100, GenDEPOT) in TBS-T (10 mM Tris-HCl, pH 7.4, 140 mM NaCl and 0.1% (vol/vol) Tween-20) at room temperature (22 °C) for 1 h. After blocking, the lipid strips were washed five times with TBS-T and further incubated with 1 μg ml$^{-1}$ mouse TRIM WT, 3% (wt/vol) BSA and TBS-T at 4 °C overnight. To determine the effects of $Ca^{2+}$, 2.5 mM $CaCl_2$ and 10 mM EGTA (Sigma-Aldrich, E3889) were used as $Ca^{2+}$-chelating agents. Lipid binding specificity was detected by using a rabbit polyclonal anti-TRIM72 antibody[19].

## Liposome preparation

Liposomes were freshly prepared by vortexing and extrusion methods. The compounds 1,2-dioleoyl-*sn*-glycero-3-phosphocholine (DOPC; Sigma-Aldrich, 850375C), 1,2-dioleoyl-*sn*-glycero-3-phospho-L-serine (DOPS; Sigma-Aldrich, 840035C), 1,2-dipalmitoyl-*sn*-glycero-3-p hospho-L-serine (DPPS; Sigma-Aldrich, 840037P) and cholesterol (Sigma-Aldrich, C8667) were dissolved in chloroform or a chloroform–methanol (2:1) solution. One micromole of lipids with the indicated compositions was evaporated in a glass vial under a fume hood above the phase-transition temperature. The dry lipid film was fully hydrated by vortexing to generate large liposomes. To prepare small liposomes, the hydrated lipids were sonicated in a bath for 1 h and extruded 35 times through a 0.1-μm polycarbonate membrane (Sigma-Aldrich, 610005) using an Avanti Mini-Extruder (Sigma-Aldrich, 610000). The quality and size of liposomes were assessed by dynamic light scattering using a DynaPro Titan dynamic light scattering instrument (Wyatt Technology), after which EM was conducted.

## Liposome coflotation assay

Small liposomes were prepared using sonication and extrusion with a composition of 100 mol% DOPC or 30 mol% DPPS and 70 mol% DOPC.

Small liposomes were incubated with 7 μM mouse TRIM72 WT and mutant proteins for 30 min at room temperature. The liposome–protein mixture was transferred to a centrifuge tube, after which sucrose solutions were overlaid on the mixture in the following order: 300 μl of 68% (wt/vol) sucrose, 2.5 ml of 17% (wt/vol) sucrose and 1 ml of 6.8% (wt/vol) sucrose in DPBS. The discontinuous sucrose gradient was centrifuged at 272,800g and 4 °C for 3 h. Each fraction from the top to the bottom of the gradient was resolved by SDS–PAGE and analyzed by western blotting using a rabbit polyclonal anti-TRIM72 antibody (Abcam, ab154238).

## Liposome co-sedimentation assay

Large liposomes with 80 mol% DOPC and 20 mol% cholesterol or 80 mol% PS and 20 mol% cholesterol were prepared by vortexing. Large liposomes were incubated with 50 μg of mouse TRIM72 WT or each mutant at room temperature for 30 min. The liposome–protein mixture was centrifuged at 21,130g and 4 °C for 10 min and then divided into supernatant and pellet fractions. Each fractionated sample was resolved by SDS–PAGE, and signals were detected by Coomassie staining.

## Flow cytometry

Fluorescence-labeled sfGFP–TRIM72 WT and FITC-labeled annexin V (FITC–annexin V; BioLegend, 640945) were prepared for flow cytometry analysis. Fluorescence-labeled and nascent liposomes were prepared by sonication in FACS buffer (50 mM Tris-HCl, pH 8.0, 150 mM NaCl, 1 mM TCEP and 0.5% (wt/vol) BSA). Liposomes containing 99.9 mol% DOPC or 30 mol% DPPS and 69.9 mol% DOPC were labeled using 0.1 mol% 1,2-dioleoyl-*sn*-glycero-3-phosphoethanolamine-*N*-(lissamine rhodamine B sulfonyl) (Sigma-Aldrich, 810150P). Rhodamine-labeled liposomes were analyzed using an Accuri C6 flow cytometer (BD Biosciences) with a 488-nm excitation laser and 533/30-nm (FL1; wavelength/filter) and 585/40-nm (FL2) emission detectors. The detection threshold was set to a forward scatter height of 2,000 and an FL2 height of 100. Spectral overlaps between sfGFP (or FITC) and rhodamine were compensated by sfGFP–TRIM72-WT-bound rhodamine-free 30 mol% PS liposomes and protein-free rhodamine-labeled 30 mol% PS liposomes. For analysis of $Ca^{2+}$ effects, real-time flow cytometry was conducted with the following three sequential steps: (1) measurement of liposomes only, (2) measurement of liposomes to which protein (sfGFP–TRIM72 WT or FITC–annexin V) had been added and (3) measurement of liposomes and proteins to which $CaCl_2$ (final concentration of 2.5 mM) had been added. Each step was conducted over a 5-min period, and the data were evaluated by using BD Accuri C6 Plus software (BD Biosciences).

## Surface plasmon resonance

Surface plasmon resonance data were collected using a Biacore T200 instrument (Cytiva) with the Series S Sensor Chip L1 (Cytiva, 29104993). For immobilization, small liposomes were generated by sonication and extrusion. We calculated the sensorgrams upon subtraction of the signal for the immobilized small liposomes with 100 mol% DOPC (reference flow cell) from that for the immobilized small liposomes with 30 mol% DPPS and 70 mol% DOPC (second flow cell). To enhance immobilization efficiency, a 1:1 mixture of liposomes with 30 mol% PS and 2.5 M NaCl solution was injected into the flow cell. After immobilization, 0.2 mg ml$^{-1}$ fatty acid-free BSA (Sigma-Aldrich, A7030) was used to block both liposome samples to prevent nonspecific binding. After normalization, serially diluted proteins were injected. Regeneration of the sensor chip was carried out by adding 50 mM NaOH solution between measurements. The binding kinetics were calculated using BIAevaluation (Cytiva) software.

## Membrane-fractionation assay

C2C12 myoblasts were cultured in DMEM medium supplemented with 10% (vol/vol) FBS (Cytiva, SH30071.03IH25-40), penicillin (100 IU ml$^{-1}$)

and streptomycin (100 µg ml⁻¹). Plasmids encoding 2× Strep–human TRIM72 WT and mutants were transfected into cells with the Neon transfection system (Thermo Fisher Scientific, MPL5000). After 48 h, cells were collected and separated into membrane and cytosolic fractions using a Mem-PER Plus Membrane fractionation kit (Thermo Fisher Scientific, 89842). Each fractionated sample (10 µg) was resolved by SDS–PAGE and analyzed by western blotting using the following antibodies: anti-Strep (IBA, 2-1509-001), anti-caveolin-1 (CST, 3238), anti-GAPDH (Santa Cruz, sc-47724) and anti-β-actin (Santa Cruz, sc-47778).

## Cross-linking assay

TRIM72-WT proteins were incubated with small liposomes composed of 80 mol% DOPS and 20 mol% cholesterol at room temperature for 3 h. After centrifugal filtration using Spin-X (Sigma-Aldrich, CLS8162), TRIM72 proteoliposomes were purified by SEC using Superose 6 Increase 10/300 GL resin in cross-linking buffer. The purified TRIM72 proteoliposomes were then incubated at 4 °C for 90 min with each of the following cross-linkers at 150 µM: bis(maleimido)ethane (BMOE; Thermo Fisher Scientific, 22323), 1,4-bismaleimidobutane (Thermo Fisher Scientific, 22331), 1,8-bismaleimido-diethyleneglycol (Thermo Fisher Scientific, 22336), 1,11-bismaleimido-triethyleneglycol (Thermo Fisher Scientific, 22337), bis(sulfosuccinimidyl)glutarate (Proteo-Chem, C1126) and bis(sulfosuccinimidyl)suberate (Thermo Fisher Scientific, 21580). The reactions were stopped by the addition of 25 mM Tris-HCl, pH 7.5, and 25 mM DTT and by boiling at 95 °C for 5 min with SDS sample buffer containing β-ME. The cross-linked products were analyzed by mass spectrometry (MS). For the time-dependent cross-linking reaction, mouse TRIM72 WT and mutants were preincubated with small liposomes containing 100 mol% DOPC or 30 mol% DPPS and 70 mol% DOPC in DPBS at room temperature for 30 min. The reactions were initiated by the addition of 20 µM BMOE to the protein–liposome mixtures at room temperature. After the cross-linking reaction was initiated by the addition of BMOE, it was stopped at each time point by the addition of 25 mM DTT and by boiling at 95 °C for 5 min with SDS sample buffer containing β-ME. Cross-linked proteins were resolved by SDS–PAGE, and signals were detected by Coomassie staining.

## Transmission electron microscopy examination of negative-stained samples

Human TRIM72 microvesicles were purified from HEK293T cells. Vesicle-bound and free-solution TRIM72 was copurified by AC using StrepTrap HP columns and fully separated by SEC (see details in the above Cloning, protein expression and purification section). To obtain mouse TRIM72 proteoliposomes, small liposomes with a composition of 30 mol% DPPS and 70 mol% DOPC were generated by sonication and extraction using 100-nm filters. The mixtures of small liposomes and purified mouse TRIM72-WT proteins were agitated gently at room temperature for 3 h to reconstitute TRIM72 proteoliposomes. The mouse TRIM72 proteoliposomes and free mouse TRIM72 proteins were further separated by SEC. The use of detergents was avoided to maintain the structure of microvesicles or liposomes. For negative EM, each peak fraction containing TRIM72 microvesicles or TRIM72 proteoliposomes was loaded onto glow-discharged carbon-coated grids (Electron Microscopy Sciences, CF200-Cu) and then rinsed and stained with 1% (wt/vol) uranyl acetate. Negative EM images were recorded on a charge-coupled device camera (2k by 2k, Gatan) using a Tecnai F20 microscope (FEI) operated at 200 kV.

## Cryogenic electron tomography

For cryo-ET experiments, mouse TRIM72 proteoliposomes were reconstituted by incubation with mouse TRIM72 WT and small liposomes containing 80 mol% DOPS and 20 mol% cholesterol. The reconstituted TRIM72 proteoliposomes were further purified by SEC using

cryo-ET buffer (50 mM Tris-HCl, pH 8.0, 150 mM NaCl and 1 mM TCEP). The proteoliposomes were loaded onto glow-discharged holey grids (Quantifoil R1.2/1.3 300 Mesh, Copper; EMS, Q350CR1.3) and then vitrified using the Vitrobot Mark IV system (Thermo Fisher Scientific) at 4 °C and 90% relative humidity. Tilt series images were collected at the Korea Basic Science Institute using a Titan Krios G2 (Thermo Fisher Scientific) transmission electron microscope operated at 300 kV using a Falcon 3EC direct electron detector (Thermo Fisher Scientific) with Tomography 4.0 (Thermo Fisher Scientific) software, a tilt range from −60° to 60° and an angular increment of 2°. Images were recorded with defocus between −4.0 and −6.0 µm at a nominal magnification of ×47,000, which corresponds to a pixel size of 1.4 Å. The total dose per tomogram was approximately 200 e−/Å². Tilt series alignment, contrast transfer function (CTF) correction and tomogram reconstruction were performed using the IMOD package[90]. Tomograms were reconstructed using weighted back projection (WBP) with a simultaneous iterative reconstruction technique (SIRT)-like filter and binned four times to a pixel size of 5.6 Å.

For subtomogram averaging of the mouse TRIM72-WT and TRIM72^M138R-mutant proteoliposomes, we prepared large liposomes containing 80 mol% DOPS and 20 mol% cholesterol by vortexing. The large liposomes were incubated with mouse TRIM72 WT at 4 °C for 90 min. The mixtures of large liposomes and TRIM72 WT were fractionated by sucrose gradient centrifugation at 312,200$g$ and 4 °C for 1 h. The cofloated fractions containing the TRIM72-WT proteoliposomes were collected and concentrated up to 0.1 mg ml⁻¹. Due to spectral and colorimetric interference by phospholipid molecules, the protein concentration of the TRIM72 proteoliposomes was calculated from the comparison with the band intensities of quantified TRIM72 proteins. TRIM72^M138R proteoliposomes were prepared by the liposome co-sedimentation protocol. Large liposomes were prepared by sonication in a bath for 1 h. The incubated mixtures with sonicated liposomes and TRIM72^M138R proteins were centrifuged at 21,130$g$ and 4 °C for 10 min. After the supernatant was removed, the pellet was washed three times and resuspended in cryo-ET buffer. TRIM72-WT and TRIM72^M138R proteoliposomes were loaded onto glow-discharged holey grids (Quantifoil R1.2/1.3 300 Mesh, Copper; EMS, Q350CR1.3) and then vitrified using the Vitrobot Mark IV system at 4 °C and 90% relative humidity. To enhance the alignment of the tilt series images, AURION Gold Tracer (Aurion, 210.133) was used as a fiducial marker. Gold–BSA nanoparticles with a size of 10 nm were concentrated by collecting a pellet portion of the solution after centrifugation and then mixed with samples at a 1:5 (vol/vol) ratio just before the vitrification procedures. Tilt series images were collected using a Titan Krios G2 microscope operated at 300 kV equipped with a Falcon 3EC direct electron detector with an angular range from −54° to 54° and an angular increment of 3°. Images were recorded using defocus between −4.0 and −6.0 µm at a nominal magnification of ×29,000, which corresponded to a pixel size of 2.3 Å. The total dose per tomogram was approximately 52 e−/Å². The tilt series were aligned using fiducial markers. Tomograms were reconstructed using SIRT and WBP, with 2× pixel binning and a pixel size of 4.6 Å. SIRT reconstruction was used for annotating subtomogram positions, whereas WBP reconstruction was used for subtomogram averaging.

## Subtomogram averaging and model fitting

The particle coordinates were extracted from reconstructed tomograms of TRIM72-WT and TRIM72^M138R proteoliposomes using the Dynamo package[91]. While assigning particle coordinates, the vesicle or ellipsoidal vesicle model was applied, depending on the shapes of the proteoliposomes. In total, we extracted 18,715 and 6,379 particle coordinates from the reconstructed tomograms of TRIM72-WT and TRIM72^M138R proteoliposomes, respectively. The micrographs in the tilt series were subjected to beam-induced motion correction using MotionCor2 (ref. [92]) followed by CTF estimation using CTFFIND4

(ref. [93]). Subtomogram coordinates, tomogram alignment information and the estimated CTF were imported for subtomogram averaging using RELION-4.0 (ref. [94]). An initial model was obtained from four binned pseudo-subtomograms with a box size of 96 pixels and a cropped box size of 48 pixels. The three-dimensional particles were aligned by carrying out a series of sequential steps of classification with alignment, refinement and further classification without alignment. Iterative refinements were performed while the micrographs were progressively unbinned to the original pixel size. After removing duplicates within 30 Å, CTF refinement and frame alignment were performed on the refined model with a box size of 512 pixels and a cropped box size of 192 pixels. The final resolutions of the reconstructions of TRIM72-WT and TRIM72[M138R] proteoliposomes were estimated to be 25 and 26 Å, respectively, based on an FSC of 0.143. The subtomogram averaging process is illustrated in Extended Data Fig. 8a. The statistics of cryo-ET are summarized in Table [2]. The relative intensity of the projection of subtomogram averaging reconstruction was analyzed with Fiji[95]. The initial model of higher-order TRIM72 assembly on the lipid bilayer was derived from the crystal packing of mouse TRIM72 ΔRING. We docked the initial model into the cryo-ET maps by rigid-body fitting using UCSF ChimeraX[87]. The fitted model was further optimized by 'phenix. dock_in_map' (ref. [96]). The model of the lipid bilayer was calculated by CHARMM-GUI[97].

### Cross-linking mass spectrometry

Each band corresponding to a cross-linked protein was subjected to in-gel digestion with trypsin for LC–MS/MS analysis. LC–MS/MS analysis was performed using a Dionex UltiMate 3000 nano HPLC system coupled online to a Q Exactive mass spectrometer (Thermo Fisher Scientific). Chromatographic separation was performed with a 150-mm × 75-μm Acclaim PepMap C18 reversed-phase analytical column (Dionex) using gradient elution (solvent A, 0.1% formic acid in water; solvent B, 0.1% formic acid in acetonitrile; 0–10 min, 2% B; 40 min, 30% B; 44–49 min, 90% B; 50–60 min, 2% B). MS data were acquired with the top ten data-dependent MS/MS scan methods. Raw data were converted to MGF format using MSConvert and used to search a database containing the mouse TRIM72 proteins with pLink 2 (ref. [98]). Peptide spectrum matches were filtered at a false discovery rate of 1%. MS/MS spectra assigned to cross-linked peptides were further confirmed by manual validation. The distances between cross-linked residues in our structural models were calculated with Xwalk[99].

### Mass spectrometry

The fractions containing TRIM72 microvesicles copurified from mammalian cells were solubilized in lysis buffer containing 50 mM Tris-HCl, pH 8.0 and 4% (wt/vol) SDS with protease inhibitor cocktail. SDS lysates were digested using filter-aided sample preparation (FASP) as described previously[100]. LC–MS/MS analysis was performed as described above in Cross-linking mass spectrometry. MS-derived data were used to search human entries in the UniProt database through the Proteome Discoverer platform (version 2.2.0.388, Thermo Fisher Scientific). The search parameters for the SEQUEST algorithm were as follows: carbamidomethylation of cysteine as a fixed modification, oxidation of methionine as a variable modification, a precursor mass tolerance of 10 ppm and a fragment mass tolerance of 0.02 Da. Peptide spectrum matches were validated using a percolator based on $q$ values at a false discovery rate of 1%. Gene ontology analysis was performed using DAVID software to evaluate biological functions[101]. A $P$ value < 0.05 was considered to indicate a statistically significant result.

### Reporting summary

Further information on research design is available in the Nature Portfolio Reporting Summary linked to this article.

## Data availability

The atomic coordinates and structure factors have been deposited in the Protein Data Bank under accession codes 7XV2, 7XYY, 7XYZ, 7XZ0, 7XZ1 and 7XZ2 and in the Small Angle Scattering Biological Data Bank under accession codes SASDK86 and SASDK96. The subtomogram averages have been deposited in the Electron Microscopy Data Bank under accession codes EMD-31150, EMD-31151, EMD-33569 and EMD-33582. CLMS and gene ontology analysis have been deposited in the Proteomics Identifications Database under the accession codes PXD024946 and PXD024978. All other data and materials are available upon request from the corresponding author. Source data are provided with this paper.

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

## Acknowledgements

We thank the staff at beamlines 4C, 5C and 11C at the Pohang Accelerator Laboratory in Korea and beamlines BL-17A and BL-10C at the Photon Factory in Japan for help with X-ray data collection. This work was performed in part under the International Collaborative Research Program of the Institute for Protein Research, Osaka University (ICR-21-05). Diffraction data were collected at Osaka University beamline BL44XU at SPring-8 (Harima, Japan) (proposal nos. 2021A6673 and 2021B6673). E. Yamashita at Osaka University provided helpful discussion during crystallographic data analysis. We thank M. H. Nam at the Korea Basic Science Institute, Seoul Center, for her help with Biacore experiments and N. Thomä at the Friedrich Miescher Institute for Biomedical Research for his comments on the paper. This study was supported by National Research Foundation of Korea grants from the Korean government (grant nos. 2020R1A2C3008285, 2021M3A9I4030068 and 2022M3A9G8082638).

## Author contributions

S.H.P., J. Han and B.-C.J. performed biochemical experiments. S.H.P., J. Han, B.-C.J. and B.H.K. purified recombinant proteins. S.H.P. prepared crystals, solved structures and conducted SAXS experiments. S.H.P. and K.E.L. performed TEM studies. S.H.P., J. Han and J.H.S. performed cell biology experiments. S.H.P., S.H.J. and Z.-Y.P. performed mass spectrometric analysis. S.H.P., H.J. and J. Hyun performed cryo-ET experiments. S.H.P., K.E.L., Y.-G.K., J. Hyun and H.K.S. analyzed data. S.H.P. and H.K.S. designed experiments and wrote the paper.

## Competing interests

The authors declare no competing interests.

## Additional information

**Extended data** is available for this paper at https://doi.org/10.1038/s41594-023-01111-7.

**Correspondence and requests for materials** should be addressed to Hyun Kyu Song.

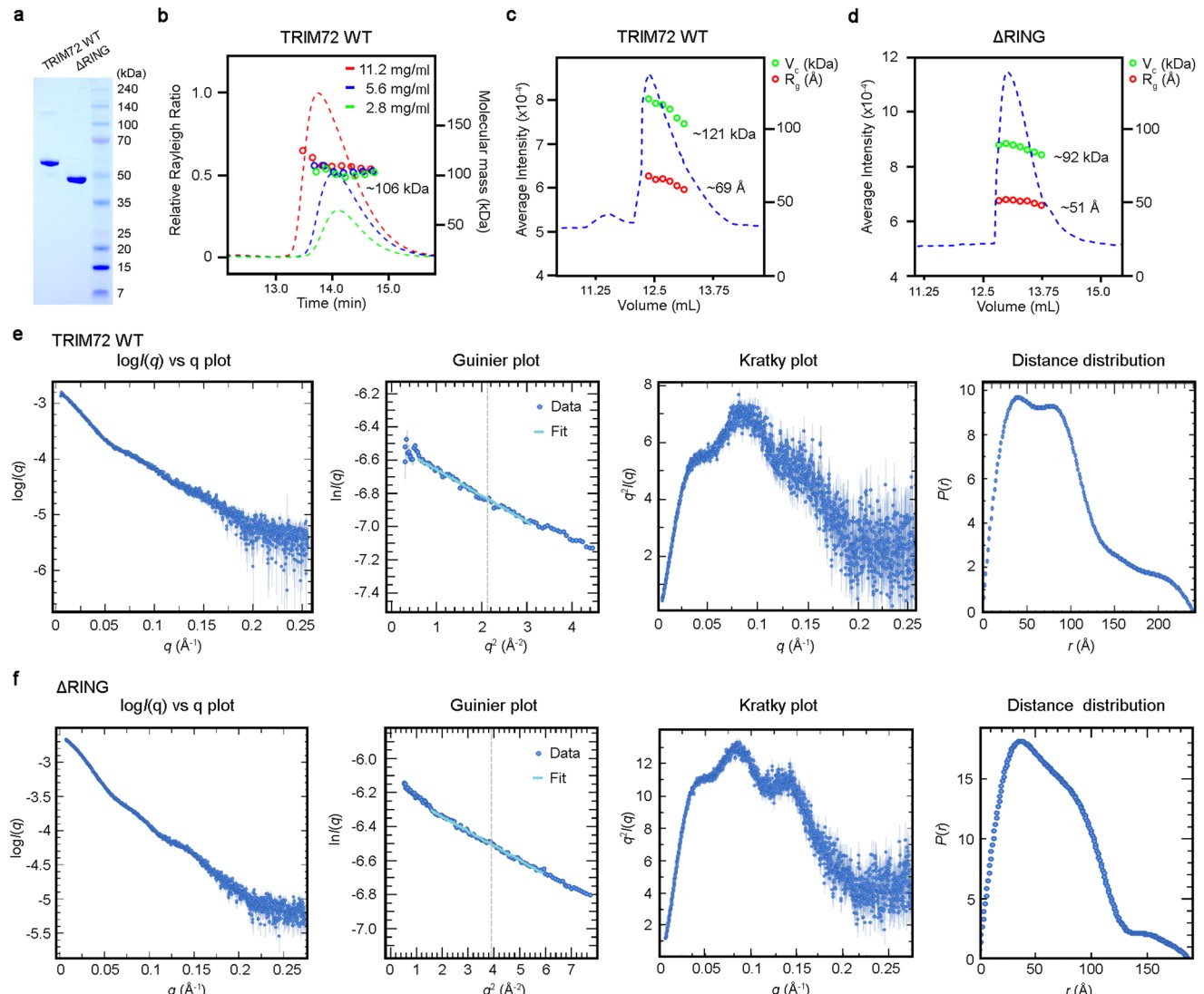

**Extended Data Fig. 1 | SEC-MALS and SEC-SAXS analysis of TRIM72.**
**a**, SDS–PAGE analysis of TRIM72 WT and ΔRING. All proteins used in this study were purified to equivalent purity. **b**, SEC-MALS profiles of TRIM72 WT at various concentrations. The relative Rayleigh ratios (left axis) and molecular masses (right axis) are shown as dashed lines and open circles of the same color for each concentration, respectively (11.2 mg/ml, red; 5.6 mg/ml, blue; 2.8 mg/ml, green). **c-d**, SEC-SAXS profiles of TRIM72 WT (c) and ΔRING (**d**). The average intensity (left axis) is shown as a dashed line. The values (right axis) with volume of correction ($V_c$)[102] and radius of gyration ($R_g$) are indicated by open circles in green and red, respectively. **e–f**, Scattering intensity, Guinier and Kratky plots, and paired distance distribution function P(r) of TRIM72 WT (e) and ΔRING (**f**). All SAXS analyses were performed using the average of the peak fractions of SEC-SAXS for each protein.

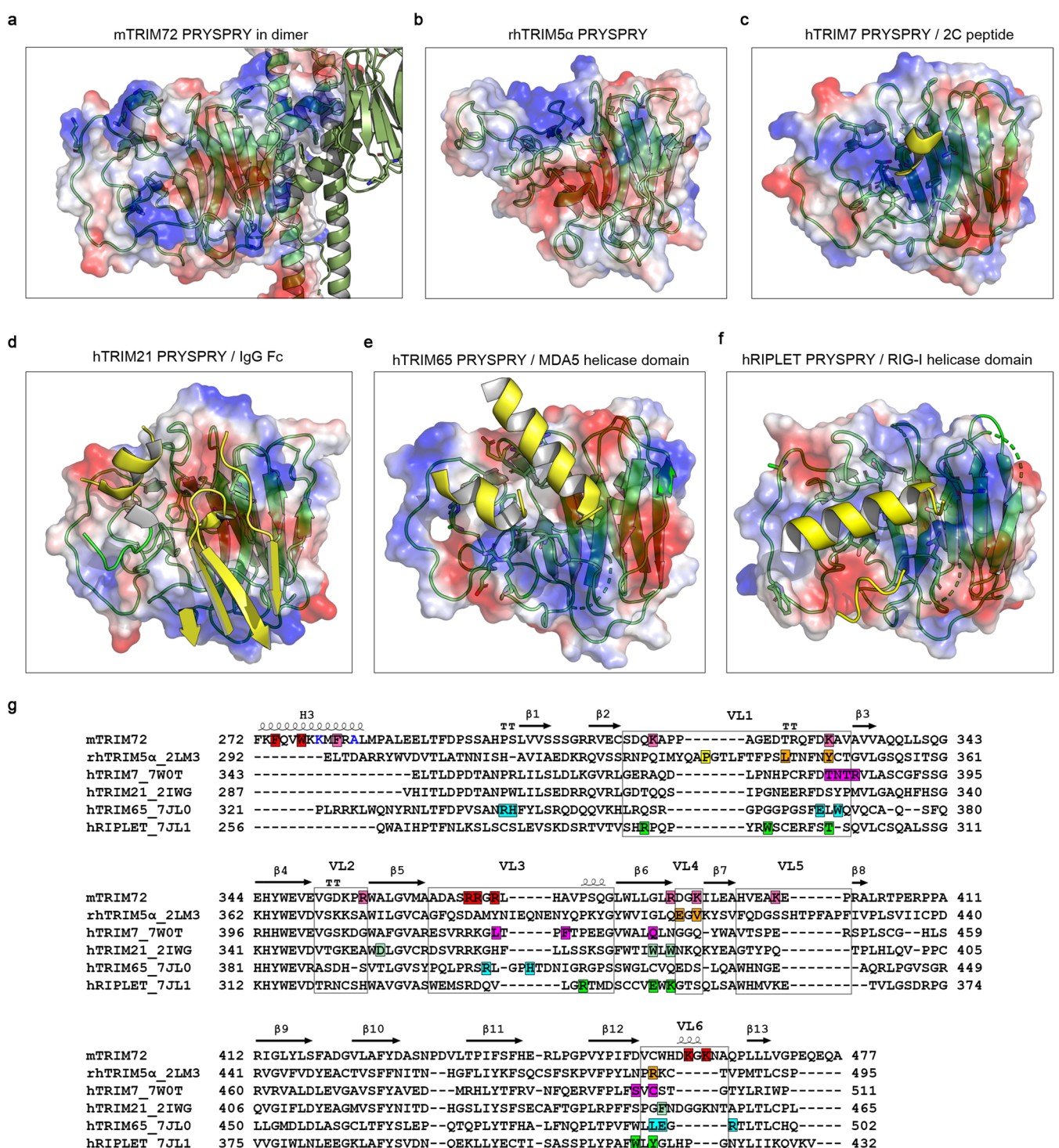

**Extended Data Fig. 2 | Comparison of the ligand binding sites among PRYSPRY domains. a-f**, Structural comparison of PRYSPRY domains. Mouse TRIM72 PRYSPRY in dimer (**a**), rhesus TRIM5α PRYSPRY (**b**), human TRIM7 PRYSPRY complexed with 2 C peptides of human enterovirus (**c**), human TRIM21 PRYSPRY complexed with human IgG Fc (**d**), human TRIM65 PRYSPRY complexed with human MDA5 helicase domain (**e**), and human RIPLET PRYSPRY complexed with the human RIG-I helicase domain (**f**). The PRYSPRY domains are shown in a ribbon diagram and electrostatic surface model with transparency. The binding molecules within 10 Å distances of PRYSPRY are shown in a ribbon diagram colored yellow. The PRYSPRY domain of mouse TRIM72 was derived from the crystal structure of TRIM72 ΔRING (2.75 Å). **g**, Structure-based sequence alignments of PRYSPRY domains corresponding to **a-f**. Mutated residues of mouse TRIM72 in this study are shown in red boxes. Potentially essential residues of TRIM72 in membrane binding are indicated as pink boxes. The proposed

residues of rhesus TRIM5α PRYSPRY for recognizing HIV capsids as indicated by NMR and HIV-1 restriction assays are shown in orange boxes. Pro334 in rhesus TRIM5α (Arg332 in humans), which is a critical residue for HIV restriction, is highlighted with a yellow box. The important residues of human TRIM7 PRYSPRY for recognizing C2 peptides from human enterovirus are shown in magenta boxes. Four hotspot residues of human TRIM21 PRYSPRY in contact with human IgG Fc are shown in mint boxes. The critical residues in human TRIM65 PRYSPRY for IFNβ mRNA levels upon poly (I:C) stimulation are shown in cyan boxes. The important residues in human RIPLET PRYSPRY for IFNβ mRNA levels upon stimulation with dsRNA are shown in green boxes. All boxed residues are represented as stick models in **a-f**. The variable loop (VL) regions 1-6 are indicated as black-lined boxes. Note that the critical residues are crowded in VL1, VL3, VL4, and VL6. The mutated residues in the H₃ helix for crystallization of mouse TRIM72 (K279H/A283H) are shown in blue.

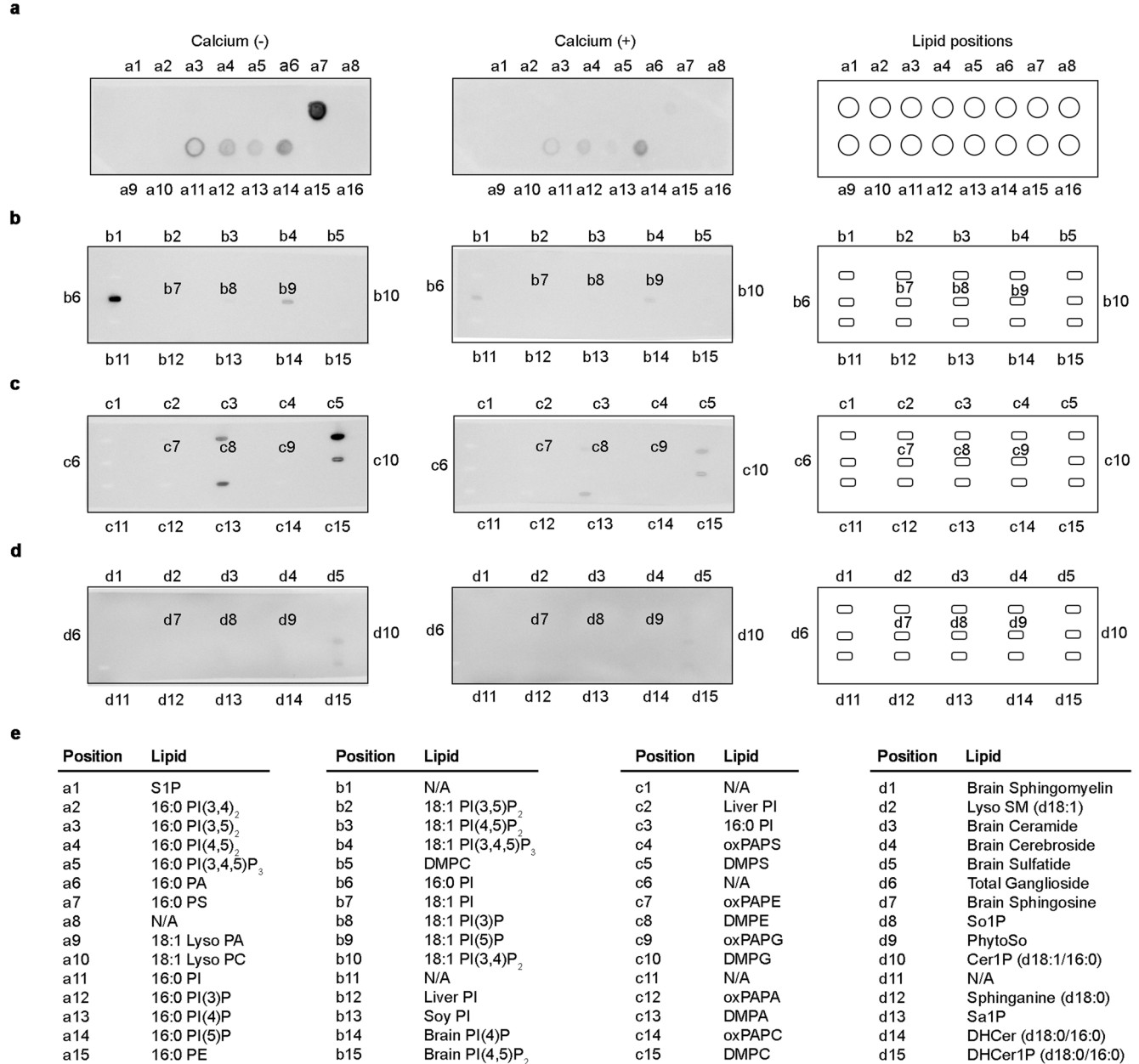

**Extended Data Fig. 3 | Protein–lipid overlay analysis. a-d,** Protein–lipid overlay assay using PIP Strips™ **(a)**, Inositol Snoopers® **(b)**, Oxidized Phospholipid Snoopers® **(c)** and Sphingolipid Snoopers® **(d)** in the absence (left) or presence (middle) of $Ca^{2+}$. The spotted lipid positions are indicated in the right panel. **e,** Lists of the examined lipids. TRIM72 WT was incubated and detected with a rabbit polyclonal anti-TRIM72 antibody. The detailed experimental procedures are described in the Methods.

| Position | Lipid |
|---|---|
| a1 | S1P |
| a2 | 16:0 PI(3,4)$_2$ |
| a3 | 16:0 PI(3,5)$_2$ |
| a4 | 16:0 PI(4,5)$_2$ |
| a5 | 16:0 PI(3,4,5)P$_3$ |
| a6 | 16:0 PA |
| a7 | 16:0 PS |
| a8 | N/A |
| a9 | 18:1 Lyso PA |
| a10 | 18:1 Lyso PC |
| a11 | 16:0 PI |
| a12 | 16:0 PI(3)P |
| a13 | 16:0 PI(4)P |
| a14 | 16:0 PI(5)P |
| a15 | 16:0 PE |
| a16 | 16:0/18:1 PC |

| Position | Lipid |
|---|---|
| b1 | N/A |
| b2 | 18:1 PI(3,5)P$_2$ |
| b3 | 18:1 PI(4,5)P$_2$ |
| b4 | 18:1 PI(3,4,5)P$_3$ |
| b5 | DMPC |
| b6 | 16:0 PI |
| b7 | 18:1 PI |
| b8 | 18:1 PI(3)P |
| b9 | 18:1 PI(5)P |
| b10 | 18:1 PI(3,4)P$_2$ |
| b11 | N/A |
| b12 | Liver PI |
| b13 | Soy PI |
| b14 | Brain PI(4)P |
| b15 | Brain PI(4,5)P$_2$ |

| Position | Lipid |
|---|---|
| c1 | N/A |
| c2 | Liver PI |
| c3 | 16:0 PI |
| c4 | oxPAPS |
| c5 | DMPS |
| c6 | N/A |
| c7 | oxPAPE |
| c8 | DMPE |
| c9 | oxPAPG |
| c10 | DMPG |
| c11 | N/A |
| c12 | oxPAPA |
| c13 | DMPA |
| c14 | oxPAPC |
| c15 | DMPC |

| Position | Lipid |
|---|---|
| d1 | Brain Sphingomyelin |
| d2 | Lyso SM (d18:1) |
| d3 | Brain Ceramide |
| d4 | Brain Cerebroside |
| d5 | Brain Sulfatide |
| d6 | Total Ganglioside |
| d7 | Brain Sphingosine |
| d8 | So1P |
| d9 | PhytoSo |
| d10 | Cer1P (d18:1/16:0) |
| d11 | N/A |
| d12 | Sphinganine (d18:0) |
| d13 | Sa1P |
| d14 | DHCer (d18:0/16:0) |
| d15 | DHCer1P (d18:0/16:0) |

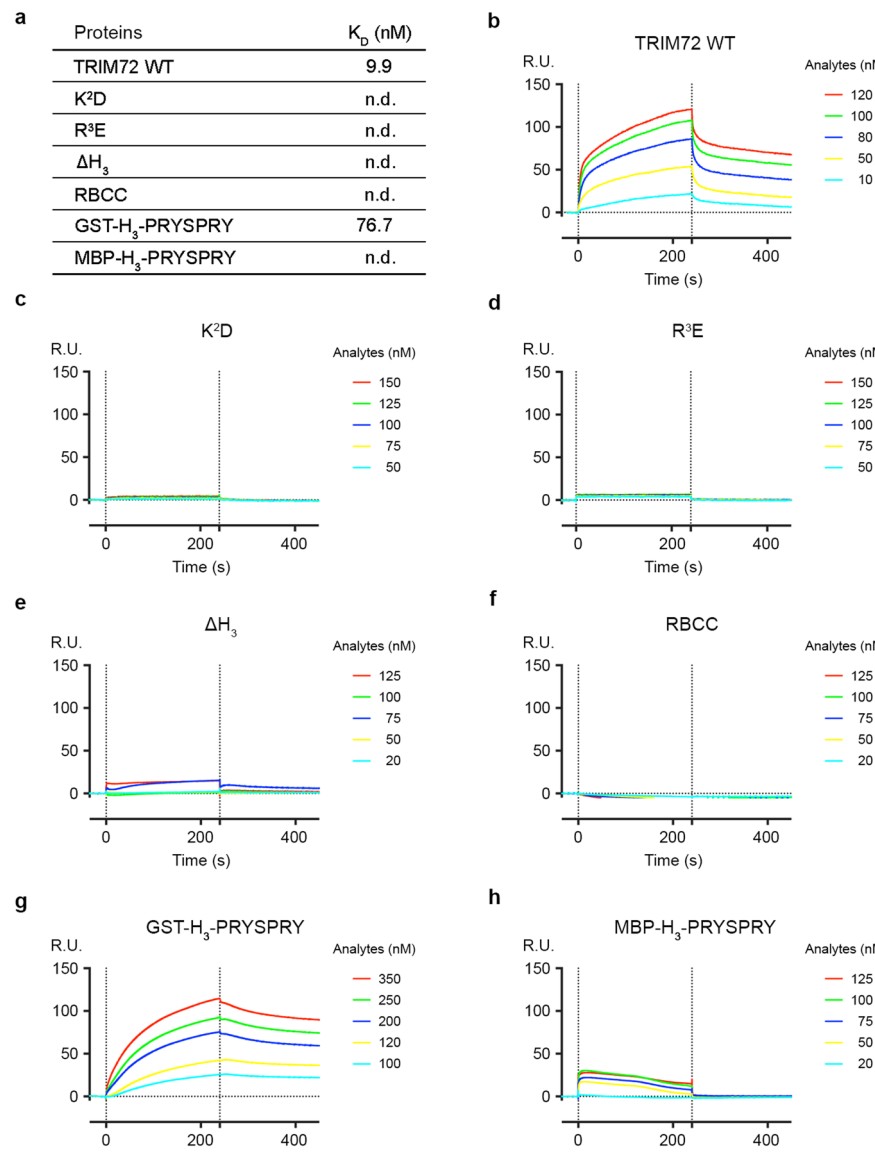

**Extended Data Fig. 4 | SPR analysis of TRIM72 WT and variants. a**, Binding affinity $K_D$ values of TRIM72 WT and variants to PS-liposomes. n.d., not determined. **b-h**, Sensorgrams with different concentrations of TRIM72 WT **(b)**, $K^2D$ **(c)**, $R^3E$ **(d)**, $\Delta H_3$ **(e)**, RBCC **(f)**, GST-$H_3$-PRYSPRY **(g)**, and MBP-$H_3$-PRYSPRY **(h)** are shown. The first and second vertical dashed lines indicate the start and end of the injection, respectively. The PS-liposomes contained 30 mol% PS. The detailed experimental procedures are described in the Methods.

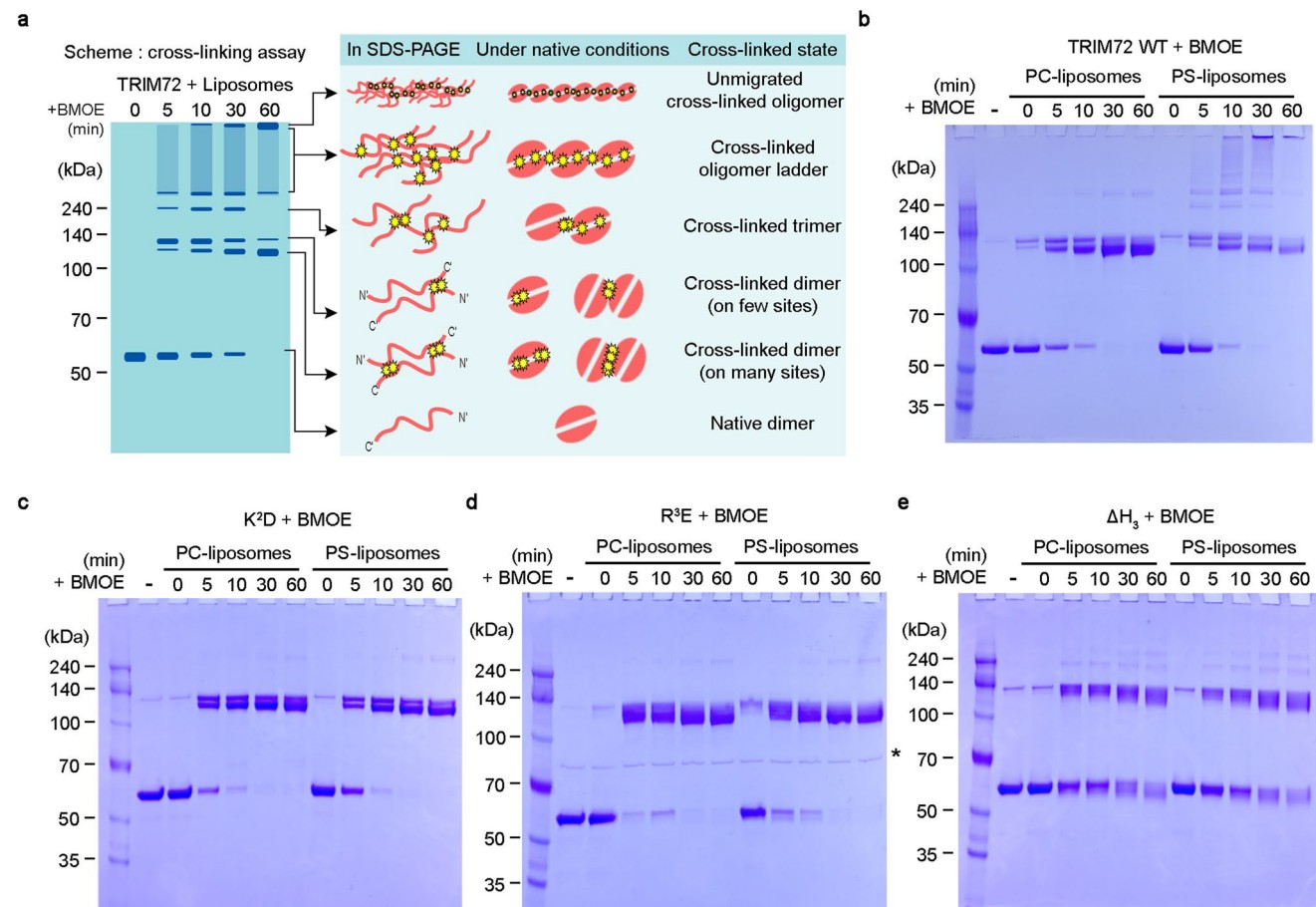

**Extended Data Fig. 5 | Crosslinking assays to identify the TRIM72 oligomer. a**, Overall scheme of the time-dependent crosslinking assay. **b-e**, Time-dependent crosslinking analysis of mouse TRIM72 WT (**b**) and variants (**c-e**). Note that each variant of K$^2$D (**c**), R$^3$E (**d**), and ΔH$_3$ (**e**) forms almost no high-molecular-weight crosslinked oligomers. All PS-liposomes contained 30 mol% PS.

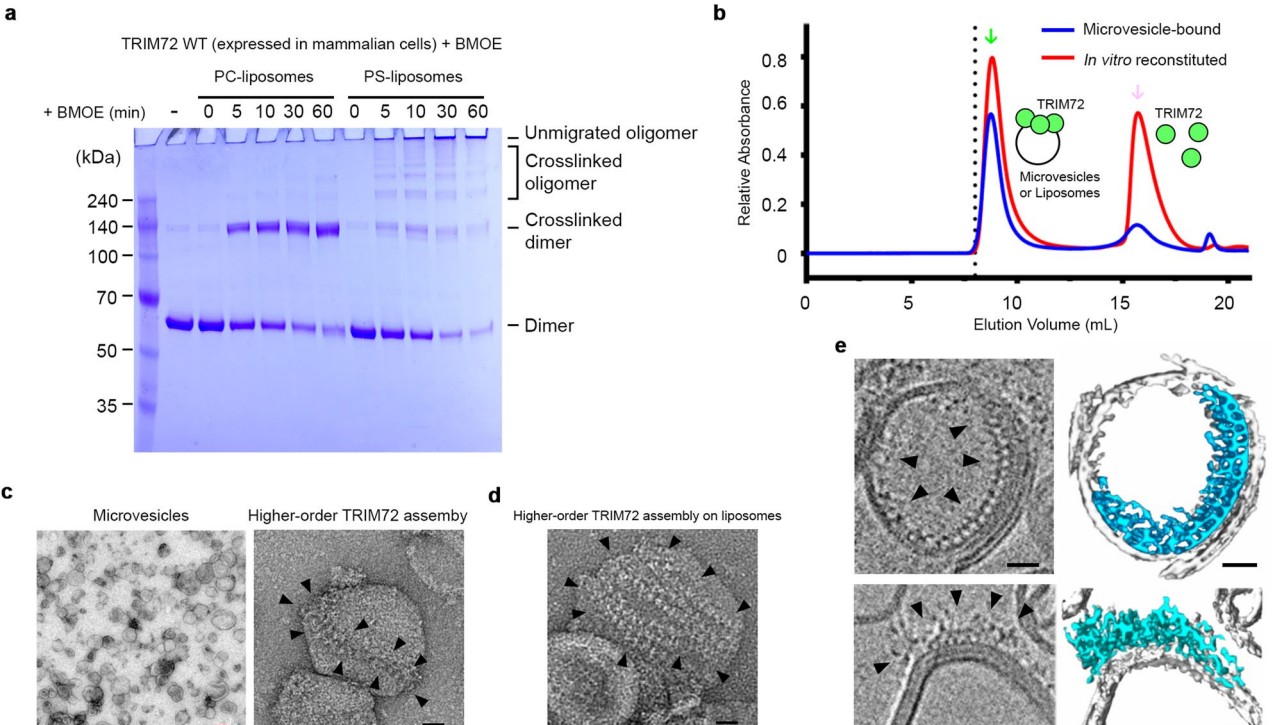

**Extended Data Fig. 6 | Higher-order TRIM72 assembly on both microvesicles and liposomes. a**, Time-dependent crosslinking analysis of TRIM72 WT purified from mammalian cells. **b**, SEC profiles of TRIM72 expressed in mammalian cells (blue line) and *in vitro*-reconstituted TRIM72 proteoliposomes (red line). The fractions containing TRIM72 bound to microvesicles or liposomes and TRIM72 in solution are indicated by arrows colored green and pink, respectively. **c**, Negative-stained TEM images of TRIM72 bound to microvesicles at low (left) and high magnifications (right). **d**, TEM images of the negatively stained TRIM72 proteoliposomes. (**c, d**) Higher-order TRIM72 assemblies are indicated by black arrowheads. The red and black scale bars indicate 100 and 20 nm, respectively. **e**, Cryo-electron tomograms (left) visualized by 3D volume rendering (right) showing the higher-order TRIM72 assembly on the inside convex (up) and outside concave (down) sides of small liposomes. The TRIM72 assembly and liposomal membranes are colored cyan and white, respectively. The black scale bars indicate 20 nm in **e**.

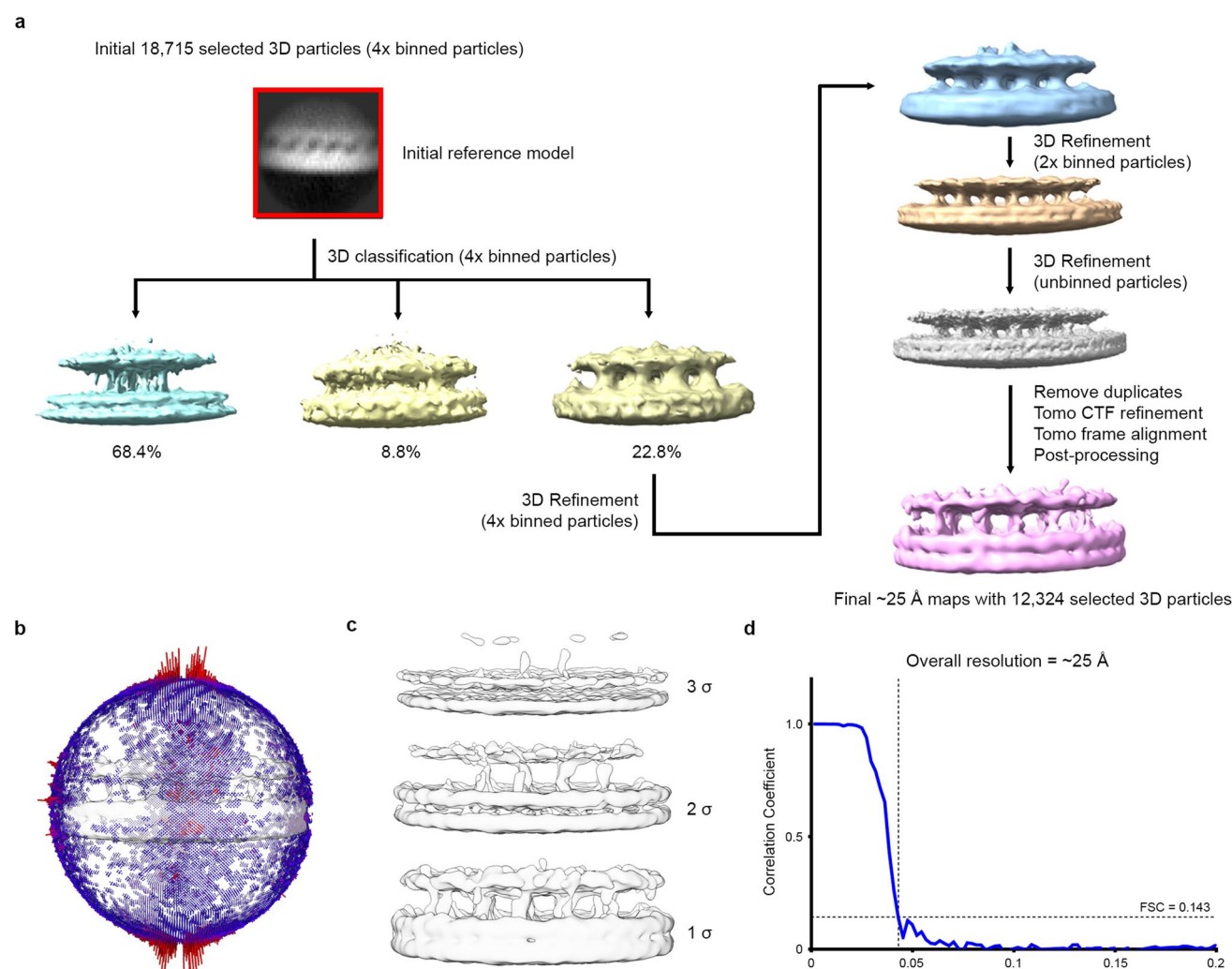

**a**

Initial 18,715 selected 3D particles (4x binned particles)

Initial reference model

3D classification (4x binned particles)

68.4%    8.8%    22.8%

3D Refinement
(4x binned particles)

3D Refinement
(2x binned particles)

3D Refinement
(unbinned particles)

Remove duplicates
Tomo CTF refinement
Tomo frame alignment
Post-processing

Final ~25 Å maps with 12,324 selected 3D particles

**b**    **c**

3 σ

2 σ

1 σ

**d**

Overall resolution = ~25 Å

FSC = 0.143

Correlation Coefficient

Resolution (Å⁻¹)

**Extended Data Fig. 7 | Subtomogram averaging reconstruction of the higher-order TRIM72 WT assembly. a**, Data processing of subtomogram averaging of the TRIM72 assembly. **b**, Angular distribution of the final refined map. **c**, Contour analysis of TRIM72 assembly. Note that the higher-order TRIM72 assembly is clearly distinguished from the phospholipid bilayer. The black scale bars indicate 100 nm (**b, c**). **d**, Fourier shell correlation (FSC) curve of the final refined subtomogram averaging map of the TRIM72 WT assembly. FSC curve calculated from two independently refined half-maps indicating the overall estimated resolution at 25 Å based on a 0.143 FSC criterion. The cryoET statistics of the TRIM72 WT assembly are summarized in Table 2.

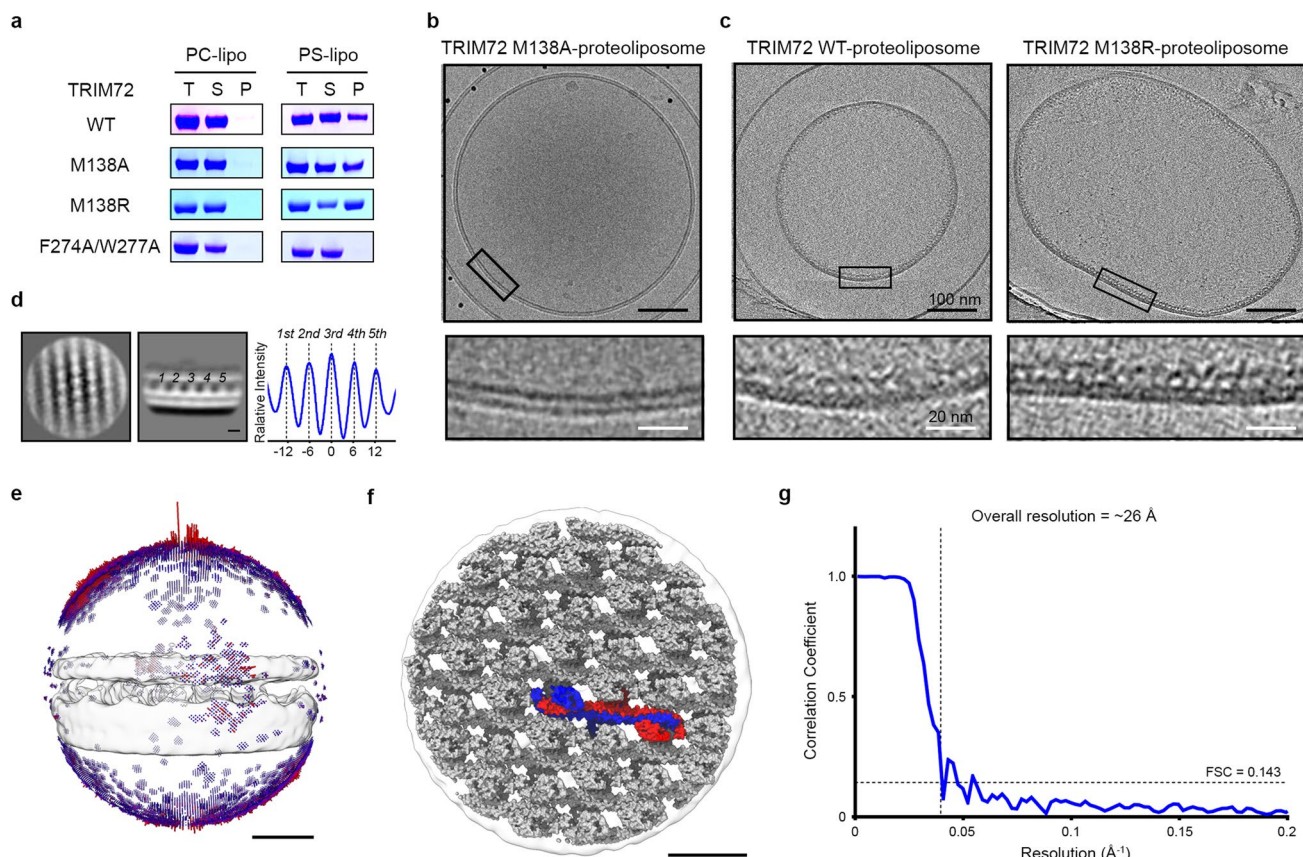

**Extended Data Fig. 8 | CryoET and subtomogram averaging reconstruction of the higher-order TRIM72 M138R assembly. a**, Liposome cosedimentation analysis. **b**, Micrograph of the TRIM72 M138A proteoliposome. **c**, Tomogram slices of TRIM72 WT (left) and M138R proteoliposomes (right). The black boxes in the top panels are enlarged in the bottom panels. Note that the TRIM72 assemblies are shown in both TRIM72 WT and M138R but not in M138A. The black and white scale bars indicate 100 nm and 20 nm, respectively. (**b,c**). **d**, Subtomogram averaging reconstruction of TRIM72 M138R. The averages of Z-slices correspond to the middle (left) and front view (middle). The relative intensity according to the x-axis (left) was calculated from the averaged Z-slices

of the front view. The number in italics (*1-5*) indicates each column-like density. The regular intervals between the peak density are 6 nm, in agreement with the intervals observed for the TRIM72 WT assembly. The black scale bar indicates 5 nm in **d**. **e**, Angular distribution of the final refined map. **f**, higher-order TRIM72 M138R assembly fitted into the subtomogram averaging map. The black scale bars indicate 100 nm (**e,f**). **g**, FSC curve of the final refined subtomogram averaging map of the TRIM72 M138R assembly. FSC curve calculated from two independently refined half-maps indicating overall resolution at 26 Å based on the 0.143 FSC criterion. The cryoET statistics are summarized in Table 2.

# Reporting Summary

Nature Research wishes to improve the reproducibility of the work that we publish. This form provides structure for consistency and transparency in reporting. For further information on Nature Research policies, see our Editorial Policies and the Editorial Policy Checklist.

## Statistics

For all statistical analyses, confirm that the following items are present in the figure legend, table legend, main text, or Methods section.

| n/a | Confirmed | |
|---|---|---|
| ☐ | ☒ | The exact sample size ($n$) for each experimental group/condition, given as a discrete number and unit of measurement |
| ☐ | ☒ | A statement on whether measurements were taken from distinct samples or whether the same sample was measured repeatedly |
| ☐ | ☒ | The statistical test(s) used AND whether they are one- or two-sided
*Only common tests should be described solely by name; describe more complex techniques in the Methods section.* |
| ☒ | ☐ | A description of all covariates tested |
| ☒ | ☐ | A description of any assumptions or corrections, such as tests of normality and adjustment for multiple comparisons |
| ☐ | ☒ | A full description of the statistical parameters including central tendency (e.g. means) or other basic estimates (e.g. regression coefficient) AND variation (e.g. standard deviation) or associated estimates of uncertainty (e.g. confidence intervals) |
| ☐ | ☒ | For null hypothesis testing, the test statistic (e.g. $F$, $t$, $r$) with confidence intervals, effect sizes, degrees of freedom and $P$ value noted
*Give P values as exact values whenever suitable.* |
| ☒ | ☐ | For Bayesian analysis, information on the choice of priors and Markov chain Monte Carlo settings |
| ☒ | ☐ | For hierarchical and complex designs, identification of the appropriate level for tests and full reporting of outcomes |
| ☒ | ☐ | Estimates of effect sizes (e.g. Cohen's $d$, Pearson's $r$), indicating how they were calculated |

*Our web collection on statistics for biologists contains articles on many of the points above.*

## Software and code

Policy information about availability of computer code

| Data collection | Crystallographic and SAXS data collection: software developed in each synchrotron beamline in PAL-5C, PF-NE3, PF-NW12, SP-44XU and PAL-4C, PF-10C, respectively; MALS data collection; ASTRA V; Liposome binding analysis using flow cytometry: BD Accuri C6; SPR: Biacore T100 Chromatography; UNICORN 7.0; Western blot: ImageQuant LAS 4000 mini; Cryo-EM: FEI EPU; Cryo-ET: Tomography 4.0; Mass spectrometry: Q exactive mass spectrometer. |
|---|---|
| Data analysis | Crystallographic data analysis: HKL-2000 v722 and CCP4 suite v8.0; MR and data refine: PHENIX package; Model building: Coot-0.9.8; SAXS data analysis: ATSAS package 3.2.1, ATSAS online and GraphPad Prism 6; Stucture visualization: PyMol 2.5.4 and UCSF ChimeraX 1.5; MALS analysis: ASTRA V and GraphPad Prism 6; Liposome analysis: Dynamics Pro, BD Accuri C6 Plus and GraphPad Prism 6; SPR data analysis: BIAevaluation and GraphPad Prism 6; Chromatography; UNICORN 7.0 and GraphPad Prism 6; Western blot analysis: ImageJ and GraphPad Prism 6; Cryo-ET reconstruction: IMOD 4.11, Dynamo package 1.1.532 and RELION4.0; Multiple Sequence alignment: Bioedit, PROMALS3D and Clustal Omega; Cross-linking analysis: pLink 2 and Xwalk; Mass spectrometry: Proteome Discoverer platform; Gene ontology analysis: DAVID software. |

For manuscripts utilizing custom algorithms or software that are central to the research but not yet described in published literature, software must be made available to editors and reviewers. We strongly encourage code deposition in a community repository (e.g. GitHub). See the Nature Research guidelines for submitting code & software for further information.

## Data

Policy information about availability of data

All manuscripts must include a data availability statement. This statement should provide the following information, where applicable:

- Accession codes, unique identifiers, or web links for publicly available datasets
- A list of figures that have associated raw data
- A description of any restrictions on data availability

The protein data bank (PDB) with the accession codes 7XV2 7XYY, 7XYZ, 7XZ0 7XZ1 and 7XZ2 for crystal structures of TRIM72. The details were described in the extended data table 2. The PDB code 3KB5 was used for structure determination. The PDB codes 2LM3, 7W0T, 2IWG, 7JL0 and 7JL1 were used for structural analysis. The small angle scattering biological data bank (SASDB) with accession codes SASDK86 and SASDK96 for TRIM72 wild-type and ΔRING, respectively. The Electron Microscopy Database (EMDB) with the accession codes EMD-31139, EMD-31151, EMD-33569 and EMD-33582 for structures of the reconstituted proteoliposome of TRIM72 WT or M138R. The Proteomics Identifications Database (PRIDE) with the accession codes PXD024946 and PXD024978 for cross-linking assay and gene ontology analysis, respectively. All available data will be released upon publication.

# Field-specific reporting

Please select the one below that is the best fit for your research. If you are not sure, read the appropriate sections before making your selection.

☒ Life sciences ☐ Behavioural & social sciences ☐ Ecological, evolutionary & environmental sciences

For a reference copy of the document with all sections, see nature.com/documents/nr-reporting-summary-flat.pdf

# Life sciences study design

All studies must disclose on these points even when the disclosure is negative.

| | |
|---|---|
| Sample size | For solving crystal structures, over 1,000 crystal conditions were tested and screened. For cryo-ET, over 100 grids were used for optimization. |
| Data exclusions | No data were excluded from the analysis. |
| Replication | All experiments were performed at least twice (mostly three times) and were replicated successfully. |
| Randomization | For the 3D reconstruction of subtomogram averaging and R-free calculation of the crystal structure, samples were randomly divided into halves. Other experiments are not involved in randomization. |
| Blinding | Blinding is not applicable for this study, which were structural and biochemical studies. |

# Reporting for specific materials, systems and methods

We require information from authors about some types of materials, experimental systems and methods used in many studies. Here, indicate whether each material, system or method listed is relevant to your study. If you are not sure if a list item applies to your research, read the appropriate section before selecting a response.

## Materials & experimental systems

| n/a | Involved in the study |
|---|---|
| ☐ | ☒ Antibodies |
| ☐ | ☒ Eukaryotic cell lines |
| ☒ | ☐ Palaeontology and archaeology |
| ☒ | ☐ Animals and other organisms |
| ☒ | ☐ Human research participants |
| ☒ | ☐ Clinical data |
| ☒ | ☐ Dual use research of concern |

## Methods

| n/a | Involved in the study |
|---|---|
| ☒ | ☐ ChIP-seq |
| ☒ | ☐ Flow cytometry |
| ☒ | ☐ MRI-based neuroimaging |

## Antibodies

| | |
|---|---|
| Antibodies used | The antibody name (with clone name when available), supplier name, catalog number, lot number and dilution factor are as follows; anti-MG53 antibody, Abcam, cat. No. ab154238, Lot No. GR196951-2, 1:5,000. anti-Ubiquitin antibody (P4D1), Santa Cruz, cat. No. sc-8017, Lot No. J0716, 1:1,000. anti-Ubiquitin antibody (VU-1), LifeSensors, cat. No. VU101, Lot No. AB40627.005, 1:1,000. StrepMAB-Classic HRP, IBA, cat. No. 2-1509-001, Lot No. 1509-0075, 1:20,000. anti-Caveolin-1 antibody, CST, cat. No. 3238S, Lot No. 3, 1:1,000. anti-GAPDH antibody (0411), Santa Cruz, cat. No. sc-47724, Lot No. K0615, 1:1,000. anti-GFP antibody (B-2), Santa Cruz, cat. No. sc-9996, Lot No. H1122, 1:5,000. |

anti ATP1A1 (C464.6), Santa Cruz, cat. sc-21712, Lot No. I1820, No, 1:1,000.
anti-beta actin antibody (C4), Santa Cruz, cat. No. sc-47778, Lot. No. L1616, 1:5,000.
anti-rabbit IgG, HRP-linked antibody, CST, cat. No. 7074S, Lot. No. 27, 1:10,000.
mouse IgG kappa BP-HRP, Santa Cruz, cat. No. sc-516102, Lot. No. I1317, 1:10,000.

Validation

The commercial antibodies employed in our study were validated by the manufacturers. The manufacturer's on-line data sheet for each antibody is located at the following web address;

anti-MG53 antibody; rabbit poylclonal IgG; suitable for western blot; reacting with human; https://www.abcam.com/mg53-antibody-ab154238.html

anti-Ubiquitin antibody (P4D1); mouse monoclonal IgG; suitable for broad application including western blot; reacting with human, mouse, rabbit and fruit fly; Citation: Cell (1996) 84:852-62.; https://www.scbt.com/p/ubiquitin-antibody-p4d1

anti-Ubiquitin antibody (VU-1); mouse monoclonal IgG; suitable for western blot and immunohistochemistry; reacting with human and mouse; Citation: Nature (2002) 416:648-53.; https://lifesensors.com/product/vu101-anti-ubiquitin-antibody-mab-clone-vu-1

StrepMAP-Classic HRP; mouse monoclonal IgG; suitable for western blot, reacting with Strep-tag II and Twin-Strep-tag; Citation: Nat. Commun. (2020) 11:2251.; https://www.iba-lifesciences.com/strepmab-classic-hrp-conjugate/2-1509-001

anti-Caveolin-1 antibody; rabbit polyclonal IgG; suitable for western blot; reacting with human, mouse and hamster; Citation: J Biol Chem (1998) 273:5419-22.; https://www.cellsignal.com/products/primary-antibodies/caveolin-1-antibody/3238

anti-GAPDH antibody (0411); mouse monoclonal IgG; suitable for western blot, reacting human; Citation: Nat. Neurosci. (2019) 22:1235-1247.; https://www.scbt.com/p/gapdh-antibody-0411

anti-beta actin antibody (C4); mouse monoclonal IgG; suitable for general application including western blot; reacting with human, mouse, rat and bovine; Citation: Cell (1995) 81:53-62.; https://www.scbt.com/p/beta-actin-antibody-c4

anti-GFP antibody (B-2); mouse monoclonal IgG; suitable for broad application including western blot; reacting with amino acids 1-238 representing full-length GFP and its variants of Aequorea victoria origin; Citation: Science (1994) 263:802-5.; https://www.scbt.com/p/gfp-antibody-b-2

anti ATP1A1 (C464.6); mouse monoclonal IgG; suitable for western blot; reacting with broad range of species including human; Citation: J Biol Chem (2001) 276:20370-8.; https://www.scbt.com/p/na-k-atpase-alpha1-antibody-c464-6

anti-rabbit IgG, HRP-linked antibody; goat anti-rabbit IgG; suitable for western blot; reacting with rabbit polyclonal and monoclonal antibodies; https://www.cellsignal.com/products/secondary-antibodies/anti-rabbit-igg-hrp-linked-antibody/7074

mouse IgG kappa BP-HRP; mouse IgG kappa binding protein; suitable for western blot; reacting with mouse IgG kappa chain; https://www.scbt.com/p/m-igg-kappa-bp-hrp

# Eukaryotic cell lines

Policy information about cell lines

Cell line source(s)

HEK293T cells (CRL-3216, ATCC) and C2C12 cells (CRL-1772, ATCC).

Authentication

Commercial cell lines were authenticated by manufacturers. C2C12 cells were further validated by muscle differentiation.

Mycoplasma contamination

Tested negative for Mycoplasma contamination.

Commonly misidentified lines
(See ICLAC register)

No commonly misidentified cell lines have been used in this study.

