## [Peer Review File · Nature Structural & Molecular Biology]

Peer Review Information

Manuscript Title: Structure and activation of the RING E3 ubiquitin ligase TRIM72 on the membrane

Corresponding author name(s): Hyun Kyu Song

Editorial Notes:

Transferred manuscripts This manuscript has been previously reviewed at another journal that is not operating a transparent peer review scheme. This document only contains reviewer comments, rebuttal and decision letters for versions considered at Nature Structural and Molecular Biology.

Reviewer Comments & Decisions:

Decision Letter, initial version:

Message: 13th Sep 2021

Dear Dr. Song,

Thank you again for submitting your manuscript "Structure and activation of the RING E3 ubiquitin ligase TRIM72 on the membrane". I apologize for the delay in responding, which resulted from the difficulty in obtaining suitable referee reports. Nevertheless, we now have comments (below) from the 3 reviewers who evaluated your paper. In light of those reports, we remain interested in your study and would like to see your response to the comments of the referees, in the form of a revised manuscript.

You will see that while the referees find the study potentially interesting, they all have different concerns about the quality of the data and the support for the main conclusions. To those ends, the reviewers offer detailed guidance for improvement that should help to strengthen the manuscript for potential publication in NSMB. Please be sure to address/respond to all concerns of the referees in full in a point-by-point response and highlight all changes in the revised manuscript text file.

We are committed to providing a fair and constructive peer-review process. Do not

hesitate to contact us if there are specific requests from the reviewers that you believe are technically impossible or unlikely to yield a meaningful outcome.

We appreciate the requested revisions are extensive. We thus expect to see your revised manuscript within 6 months. If you cannot send it within this time, please let us know. We will be happy to consider your revision as long as nothing similar has been accepted for publication at NSMB or published elsewhere. Should your manuscript be substantially delayed without notifying us in advance and your article is eventually published, the received date would be that of the revised, not the original, version.

Reporting Summary:

When submitting the revised version of your manuscript, please pay close attention to our [href="https://www.nature.com/nature-research/editorial-policies/image-integrity">Digital Image Integrity Guidelines.](https://www.nature.com/nature-research/editorial-policies/image-integrity)

We require deposition of coordinates (and, in the case of crystal structures, structure factors) into the Protein Data Bank with the designation of immediate release upon publication (HPUB). Electron microscopy-derived density maps and coordinate data must be deposited in EMDB and released upon publication. Deposition and immediate release of

NMR chemical shift assignments are highly encouraged. Deposition of deep sequencing and microarray data is mandatory, and the datasets must be released prior to or upon publication. To avoid delays in publication, dataset accession numbers must be supplied with the final accepted manuscript and appropriate release dates must be indicated at the galley proof stage. Please find the complete NRG policies on data availability at <http://www.nature.com/authors/policies/availability.html>.

[Redacted]

Kind regards,
Florian

Florian Ullrich, Ph.D.
Associate Editor
Nature Structural & Molecular Biology
ORCID 0000-0002-1153-2040

Referee expertise:

Referee #1: structural biology, ubiquitin ligases

Referee #2: structural biology, ubiquitin ligases

Referee #3: cryo-ET, electron microscopy

Reviewers' Comments:

Reviewer #1:

Remarks to the Author:

This manuscript provides a characterization of the structural features and lipid binding

properties of Mitsugumin 52 (MG53), also known as TRIM72, a protein that belongs to the TRIM family of E3 ligases and has been described a regulator of multiple cellular processes including cardioprotective signalling, membrane repair and degradation of insulin receptor substrate. Details of the molecular mechanism underlying these different functions remain largely unknown, as is the link to any potential ubiquitination activity of TRIM72.

Here, the authors describe crystal structures of multiple constructs of TRIM72 and investigate the ability of its PRYSPRY domain to mediate interaction with membranes and the effects membrane-association has on the self-association of TRIM72 and its catalytic activity. The authors use an extensive range of experimental approaches to investigate these properties including X-ray crystallography, electron microscopy, small angle X-ray scattering SAXS, surface plasmon resonance, protein crosslinking and proteomics.

In principle this is an interesting study and the observation that TRIM family proteins can associate with membranes, which in turn might regulate higher order self-association and catalytic activity could generate a lot of excitement.

However, in its present form the paper is preliminary and very speculative as all the data presented on lipid association of TRIM72 have been generated in an artificial system with overexpressed proteins and it is unclear how the observations described here relate to the physiological function of TRIM72. Furthermore, the link between membrane association and TRIM E3 ligase activity is currently weak, and no consistent molecular explanation is provided how the two might be linked (for details see below). Finally, the multiple diverse experimental approaches applied by the authors and large amount of data presented make the manuscript very difficult to read and comprehend as often no rationale for a given experiment or link between different approaches is provided. Similarly, the manuscript contains a number of statements that are unclear and not always supported by references or data. Finally the authors should reconsider which Figures to put in the main part of the manuscript and which into the Extended Data section. At present a lot of data that are required to understand the manuscript are in Extended Data whereas some of the main Figures are not central to the story and could be moved to Extended data. Specific points are listed below that the authors should address to ensure that the conclusions they draw are appropriately supported by the data provided.

Specific points

- Crystallographic data:

The authors present multiple structures of different constructs that crystallized in different crystal forms, in part at very low resolution (resolution limits of over 5 Å and 7 Å), and some with overestimated resolution ranges (PDB:7DI2, CC1/2 of 0.256 in the highest res shell). This doesn't add much to the story and the authors should rather focus on one or two crystal forms with decent resolution limits and statistics.

Importantly, the authors need to be absolutely specific in the text and figure legends about which structure they are describing at what resolution. This is particularly important for Figure 1. In its current form, the positioning of the RING in Fig 1b,c is rather misleading as no density could be observed for the RING in most crystal forms, and the authors themselves go on to employ SAXS to demonstrate how flexible the position of the RING domain is. This is particularly relevant as the authors describe an inactive state of the RING due to interactions with the CCD (see Ext Data Fig 5a). This is rather confusing and needs to be clarified. If the RING domain is flexible it is unlikely to be inhibited by contacts with the CCD. Panels Fig 1c,d could be moved to Extended Data. Inclusion of the membrane model in Figure 1b is misleading and should be removed.

Furthermore, the authors need to show electron density of multiple regions in the

structures described, given the rather low resolution of many of the structures. Overall, the authors should be careful in the description of the different crystal forms and how much they read into any differences between the structures observed. Part of this might be solely due to crystal contacts.

- The description of the TRIM72 structure contains a long paragraph describing the coiled coil domain (CCD) and multiple associated figures. CCDs of TRIM proteins have been described in detail before – it is not clear what novel insight the authors are providing here. This needs to be made clear, otherwise this section should be condensed.

- SAXS data:

The authors should provide the R_g profiles of the SEC traces of mTRIM72 shown in Ext Data Fig 1c,d. They also need to add the calculated protein molecular weights based on their SAXS data, to the SAXS table. Do they correspond to the theoretical calculated MW of a dimeric molecule or is there any evidence for aggregation? Why is the Porod volume of mTRIM72 WT so much bigger than of the deltaRING construct (Ext Data Table 3)? Why did the authors only take flexibility of the RING domain into account, but not of the ends of the CCD, which move “up and down” when carrying out their modelling of the SAXS data? Would taking flexibility of the CCD into account improve the X₂, which is currently quite poor?

- Role of the PRYSPRY domain in lipid binding:

The observation that the PRYSPRY domain of TRIM72 has lipid binding properties is very exciting and could point to a completely new role for this protein interaction module. To provide a more general context the authors should compare the electrostatic surface potentials of other PRYSPRY domains for which structural information is available. Could this be a more general property of this domain? This is particularly relevant as PRYSPRY domains have also been suggested to bind RNA. How does the surface involved in membrane association relate to the ligand binding sites of PRYSPRY domains identified so far?

This novel property of the PRYSPRY domain is an important section of the manuscript that should be further developed.

- Lipid binding assays:

The total amount of TRIM72 in the membrane fraction between TRIM72 WT and the K2D mutant is very similar, why do the authors think there is no apparent binding of the mutant protein in the SPR experiment?

The SPR-based study showed that mTRIM72 binds to PS-SUVs with very high affinity (9.9 nM). This affinity is entirely reliant on the PRYSPRY domain, which however needs to be dimeric. The authors show this by employing a GST-tagged hTRIM72 PRYSPRY domain, which has been calculated to have an affinity of 21.3nM. However, this is not what the data shown in Ext Data Fig.10g suggest as a much higher concentration of protein is required to reach saturation as with the full-length WT protein (panel 10a). How do the authors explain this? And why do the proteins in panels 10e and f produce such strong signals?

Given the high affinity of TRIM72 for PS-SUVs measured by SPR analysis why is the proportion of TRIM72 comigration with PS-SUVs so small (Extended data Fig 11)?

- Catalytic activity and ubiquitination assays:

Based on the data presented by the authors in Extended Data 11 the linchpin Q57 is responsible for the lack of activity of TRIM72 and its replacement with R completely

negates the need for membrane-induced oligomerisation. However, some RING E3 ligases are active without containing a conventional linchpin residue (e.g. TRIM32, Extended Data Fig 5a). How do the authors explain this?

Furthermore, in light of these data, what do the authors speculate is the molecular mechanism by which membrane binding induces TRIM72 WT activity? They do not provide any insight into this. Many TRIM E3 ligases need to dimerize to be active but this does not appear to be the case here as the authors have shown that fusion of two RING domains is not sufficient to establish catalytic activity, as has been shown for other TRIMs such as TRIM5alpha. What do they suggest is the mechanism of RING activation? If the linchpin residue was indeed the reason for the lack of activity, how can membrane association restore activity?

- I do not understand the experiment in which the dimeric form of TRIM72 can be "activated" by the oligomeric form. How would this activation work? Isn't the observed activity just coming from the "oligomeric form"? Why is there such a significant difference in apparent activity between the 2 dimeric samples in Extended data 16e that have not been incubated with "oligomeric" TRIM72 (neither no heat or heat treated)? Should they not be identical?

- Figure 4: How did the authors check that the hTRIM72 purified from mammalian cells used in this experiment is microsome bound? And given that the protein is purified from mammalian cells – how can the authors exclude that the ubiquitination activity seen does indeed come from TRIM72 and not another protein that has copurified in the microsomes? Similarly, it is not clear if hTRIM72 purified from HEK293T cells is membrane associated or not. There is no detergent in the lysis or purification buffer. Please specify as this is crucial for the interpretation of the results.

It would also be interesting to carry out a mass spectrometry analysis of hTRIM72 purified from HEK293T in order to investigate if the protein may contain post-translational modifications such as phosphorylation that could influence its properties.

- Overall, the authors should provide more details about the experimental methods, especially with respect to the protein preparations that have been used for electron microscopy. It is not clear what preparation has been used for which images, and what precisely the "higher-order oligomer fractions of hWT" (line 628) are. What precisely is the difference between the microsomes versus reconstituted liposomes?

- Electron microscopy

The authors provide some good and interesting negative stain images and cryo-tomograms. However, in their current form they are preliminary, and a number of issues should be addressed to ensure that the conclusions drawn, and models suggested are fully supported by the data provided. Specifically, the authors should provide clear EM evidence that support a specific orientation for membrane interactions with the PRYSPRY domains pointed at the membrane and a specific architecture for their higher-order assembly (the 3D volume rendering shown in Fig 3 does not provide this information). For example the authors could carry out a direct comparison of a calculated density from their crystallographic model (or packing model from within the crystal) with negative stain images looking down on the membrane that shows the right size and shape (Extended Data Figure 13) or correct spacing for the higher-order assembly. Similarly, calculated density from the X-ray model matching either cryo-EM (images) or tomograms showing detailed features at the edge of liposomes. At a minimum, some evidence of dimensions matching or some similarity of density distribution to the crystallographic model should be

provided. A more definitive proof would be obtained by subtomogram averaging (averaging of density volumes within the tomograms). Similarly, for the liposome bridging structure, the overall dimensions may be correct for one of the crystallographic packing models in Extended data 15d. But there is insufficient EM evidence presented to prove this, maybe the proposal could be tested by mutations that disrupting the interface that makes the bridges.

- The authors need to run their SEC-MALS experiments at different protein concentrations to investigate if the protein forms higher order oligomers at higher concentrations. In fact, the molecular weight distribution shown in Ext Data Fig 1b suggests it does.

- In general, members of the TRIM family of E3 ligases have not been described as membrane associated proteins. The authors should provide a short paragraph in the introduction about what precisely is known in the literature about the membrane-binding properties of TRIM72, and how this relates to its described physiological function.

- Experiments presented in this study have been carried out either with human or mouse TRIM72, and it is not always clear why one protein was used over the other. This is particularly problematic as the authors state that the human protein purified from mammalian cells exists in dimeric and oligomeric forms (line 214). How does this influence the experiments?

Minor points:

Abstract: what is "a characteristic pair of PRYSPRY domains"? On its own this statement doesn't make sense.

Line 49: What is meant by "The most recently discovered Ub ligases"?

Line 76: "Ub ligase activity is also controversial due to a lack of biochemical and biophysical information." What is controversial: if it has activity, what types of chains it makes or the substrates? Please be more precise. Neither of the references cited describe the ubiquitination properties of TRIM72 (41, 42). Similarly, please do not describe TRIM72 as a TRIM E3 ligase with auto-ubiquitylation activity (lines 62 and 66), after you stated in the abstract that it is inactive in solution, or at least qualify the statement.

The nomenclature used by the authors for their constructs (lines 91/92) is not very intuitive. Do the authors have data that indicate that the mutant mouse TRIM72 constructs that are being used throughout the manuscript behave as the WT proteins?

Line 122: What point are the authors trying to make when they talk about the "appropriate distance" of the zinc-binding modules to the CCD?

The authors should provide a figure in extended data in which they highlight the loop that has been removed in construct delta272-281. Do they have any evidence to show that this protein is properly folded?

Legend Ext Data Fig1: The authors do not provide "high resolution crystal structures"

Could the authors please elaborate on how "The unique architecture of TRIM72 allows its

promiscuous interaction with any membrane shape if the membrane is enriched in PS.”?

Ext. Data Fig.5d: Why did the authors align the B-box domain with RING domains? What point are they trying to make?

The comparison of the TRIM72 structure with a bird is rather detracting from the description.

Reviewer #2:

Remarks to the Author:

In this manuscript, Park et al showed full-length structures of TRIM72, an E3 ligase with an unusual activity of binding lipid membrane. The structure revealed the overall architecture of TRIM72 and a number of interesting features. While RING is flexible, it is tethered to CC in the autorepressed state that prevents RING from engaging with E2. Membrane binding leads to higher order oligomerization of TRIM72, which was accompanied by significant increase in E3 ligase activity. The authors further argue that a subset of TRIM proteins evolved to adopt Gln, instead of Arg, at the linchpin position of RING to suppress its activity in the absence of higher order oligomerization.

Overall, the structural and biochemical data in this manuscript are novel and interesting, and provide unique insights into the mechanism of a highly unusual E3 ligase that recognizes membrane. However, I feel that the manuscript needs more work in strengthening their models. While the data are expansive as a whole, data supporting each aspect of the mechanisms remain somewhat weak as illustrated below.

1. Ring repression and activation mechanism

It is somewhat confusing what conformation RING is in the dimeric formation. In some area of the manuscript, RING is thought to be flexible. However, in other areas, the authors argue that RING is tightly bound by CC, and thus preventing the interaction with E2. Which one is it? Is a mutation in CC that releases RING leads to spontaneous activation? And how is the higher order oligomerization of TRIM72 helps RING get released from this autorepression? If interdimeric RING:RING interaction is the mechanism of multimerization-dependent activation, mutations in the RING:RING interface (which one can predict based on other RING dimeric structures) should abrogate TRIM72 ubiquitination activity.

2. Higher order oligomerization

This is really an important part of the paper, but I had a hard time understanding the text and related figures. In particular, it is very difficult to understand how the X-linking data in Fig 3d led to the model in Fig 3e. Which arrows indicate which crosslinkings? It is also hard to see the red balls (indicating the reactive Cys). Can the authors separately indicate X-linking between RINGs, B2s and CCs? Perhaps Cys in different domains should be in different colors?

What is the physiological concentration of PS in cellular membrane (organelle vs. plasma membrane)? Could the TRIM72 clustering an artifact of having an excessive concentration of PS in the in vitro liposome assay or microsomes purification/extraction? Membrane-mediated oligomerization: does it simply reflect TRIM72 binding to pre-organized clustering of PS within the liposome or microsomes or does TRIM72 induce clustering of PS?

What domain is required for TRIM72 oligomerization upon membrane binding? Based on X-linking, one would expect that B-box, CC and RING may play an important role?

3. Membrane curvature recognition

The presented data do not appear to provide any information about membrane curvature recognition. In fact the authors also noted in lines 236-237 that TRIM72 can recognize any membrane shapes.. In this case, a better way to phrase this would be that TRIM72 recognizes PS (and perhaps clustered PS) independent of the membrane shape, as it was found to localize on both convex and concave side of the membrane in this study. As is currently written, it suggests that TRIM72 cannot recognize flat membrane structure, for which the authors have no data for. It is also unclear whether this is relevant for the biological function of TRIM72. Wouldn't the local concentrations of PS/PI be a more relevant information for membrane specificity of TRIM72? Is this an important point to make, and do authors have sufficient data to conclude in one way or another? My guess is no.

4. Ub ligase activity

Fig 4 is interesting but I do not believe it was properly performed to allow accurate comparison between dimer vs. multimer (liposome-bound) or between WT vs. mutants. And this is because the comparison was done without any consideration of the TRIM72 quantity. I would strongly suggest that the authors repeat this experiment with the same concentration of TRIM72 for WT vs. mutants, and in free or liposome-bound state and run the Ub conjugation reaction all in the same gel to allow quantitative comparison among the samples. Without such side-by-side comparison, I don't think the authors' conclusions are justified.

Ext. data Fig 16e: I am not sure what the data in this figure actually offer. While the authors argue that the result shows that the dimeric form of TRIM72 can be activated by the oligomeric form, what was added is the catalytically active oligomer of TRIM72 and the liposome, so it is unclear how this data supports the authors argument.

5. Lipid repair and oxidation

Link to oxidative damage (Ext. Data Fig 12g) is not clear since PS induces oligomerization even in the absence of H₂O₂. If the authors want to say anything about the impact of oxidative damage, the experiment has to be a comparison of +/- H₂O₂ in the same gel.

6. PRY-SPRY architecture

While deletion of H3 completely impaired PS liposome binding of TRIM72, despite harboring intact PRY-SPRY, GST-PRY-SPRY displayed relatively strong affinity for PS liposome. The latter seems to suggest that PRY-SPRY does not need to be rigidly held by CC for lipid recognition. Why then the deletion of H3 lead to the loss of lipid binding? If one makes a more sophisticated mutation in the interface between CC and PRY-SPRY (rather than deleting a large portion of the protein), what happens to lipid binding, high-order oligomerization and ubiquitination?

7. Overall, the manuscript has limited description of the experiment performed in either main text or figure legend. More detailed description is necessary (besides the methods section). For example, it is difficult to understand which experiment utilized microsome vs. liposome. For EM images of microsome (in Ext. Data Fig 13 a-b), how do the authors know whether the highlighted structures are from TRIM72 (vs. other cellular proteins)? Shouldn't it require some form of antibody-mediated labeling (e.g. gold labeling)?

8. The manuscript will benefit from more careful editing as I found several areas difficult to understand, some with grammatical errors (e.g. lines 115-116, 183-185).

Reviewer #3:

Remarks to the Author:

In this manuscript the authors present the crystal structure of full-length TRIM72, a RING E3 ubiquitin ligase. TRIM72 is an important protein in protein homeostasis but also function in signal transduction, membrane trafficking and DNA repair. TRIM72 is a multidomain protein.

In this study the authors characterised full-length wild type mouse TRIM72 in solution with SAXS. Determined the crystal structures of several constructs of mouse TRIM72, either truncated or mutated at specific positions as well as a very low resolution ($\sim 7\text{\AA}$) full length. In addition, they characterised membrane binding with other set of mouse TRIM72 constructs as well as constructs of human TRIM72. The membrane binding was assayed with lipid blots, negative stain EM and cryoEM/ET. Based on the data obtained from all these constructs and techniques the authors propose a model for TRIM72 function.

It is a remarkable achievement to get the crystal structure of full-length TRIM72. However, the amount of protein engineering needed and the low resolution obtained raise concerns about the interpretation of the structures. In addition, a major part of the interpretation is based on the cryoET data that is underwhelming. Taken together I find the proposed model highly speculative.

Specific concerns that the authors should address before it can be considered for publication:

1) Regarding the crystal structures:

- a. the resolution of some of the structures is really low $>5\text{\AA}$ and the $R_{\text{work}}/R_{\text{free}}$ are bad questioning the quality of these structures
- b. based on superposition of the different crystal structures the authors suggest 'conformational movement' but since each structure is of a different construct (truncated or mutated) how can the authors rule out that they were not induced by crystal packing or because of the mutations that they introduced?

2) Regarding the structures on membranes:

- a. how did the identity of the proteins on vesicles validated?
 - b. The negative stain data clearly show that hTRIM72 can be organised on membranes, however, this is not obvious from the cryoEM/ET data presented. How does the top views look in cryoET?
 - c. If indeed TRIM72 have a specific high-order arrangement on membranes, then that arrangement should be experimentally determined by sub-tomogram averaging.
- 3) In Fig 3c there is one vesicle with proteins binding to the inside (negative curvature).
- a. How did the protein got to the inside of the liposomes if the protein was added to pre-formed liposomes?
 - b. was assembly of TRIM inside also observed for hTRIM72?
 - c. to assess the significance of this observation the authors need to quantify the number of cases where they observed binding from the inside and compare it to outside. A meaningful size data set should be used for that.

4) In the model presented how the membrane curvature was accounted for? currently the model looks very flat.

Minor points:

- Since structures presented in this manuscript are coming from different techniques, the figure legends should be very clear from which technique each structure is coming from. For example in figure 1 it should be clearly mentioned that these are crystal structures. The legend to Fig 1 should also include the colour coding for each construct and what the author mean by 'overall structure', which construct is this?? The membrane should be removed from this figure as it is not part of the structure determined.
- correct typo in line 222 should be 100 nm not micrometer.

Author Rebuttal to Initial comments

Point-by-point responses to the comments of the reviewers

Thank you again for submitting your manuscript "Structure and activation of the RING E3 ubiquitin ligase TRIM72 on the membrane". I apologize for the delay in responding, which resulted from the difficulty in obtaining suitable referee reports. Nevertheless, we now have comments (below) from the 3 reviewers who evaluated your paper. In light of those reports, we remain interested in your study and would like to see your response to the comments of the referees, in the form of a revised manuscript.

You will see that while the referees find the study potentially interesting, they all have different concerns about the quality of the data and the support for the main conclusions. To those ends, the reviewers offer detailed guidance for improvement that should help to strengthen the manuscript for potential publication in NSMB. Please be sure to address/respond to all concerns of the referees in full in a point-by-point response and highlight all changes in the revised manuscript text file.

We appreciate the requested revisions are extensive. We thus expect to see your revised manuscript within 6 months. If you cannot send it within this time, please let us know. We will be happy to consider your revision as long as nothing similar has been accepted for publication at NSMB or published elsewhere. Should your manuscript be substantially delayed without notifying us in advance and your article is eventually published, the received date would be that of the revised, not the original, version.

Reporting Summary:

Reviewers' Comments:

Responses to Reviewer #1

Reviewer #1:

Remarks to the Author:

<<< This manuscript provides a characterization of the structural features and lipid binding properties of Mitsugumin 52 (MG53), also known as TRIM72, a protein that belongs to the TRIM family of E3 ligases and has been described a regulator of multiple cellular processes including cardioprotective signalling, membrane repair and degradation of insulin receptor substrate. Details of the molecular mechanism underlying these different functions remain largely unknown, as is the link to any potential ubiquitination activity of TRIM72.

Here, the authors describe crystal structures of multiple constructs of TRIM72 and investigate the ability of its PRYSPRY domain to mediate interaction with membranes and the effects

membrane-association has on the self-association of TRIM72 and its catalytic activity. The authors use an extensive range of experimental approaches to investigate these properties including X-ray crystallography, electron microscopy, small angle X-ray scattering SAXS, surface plasmon resonance, protein crosslinking and proteomics.

In principle this is an interesting study and the observation that TRIM family proteins can associate with membranes, which in turn might regulate higher order self-association and catalytic activity could generate a lot of excitement.

However, in its present form the paper is preliminary and very speculative as all the data presented on lipid association of TRIM72 have been generated in an artificial system with overexpressed proteins and it is unclear how the observations described here relate to the physiological function of TRIM72. Furthermore, the link between membrane association and TRIM E3 ligase activity is currently weak, and no consistent molecular explanation is provided how the two might be linked (for details see below).>>>

Thank you for the valuable criticism of our manuscript. The model for the membrane-bound TRIM72 has been validated by cryo-electron tomography and subtomogram averaging with the reconstructed TRIM72-proteoliposome (Fig. 3b, c and Extended Data Figs. 8 & 9 in the revised manuscript).

Based on the high-order TRIM72 assembly structures and additional biochemical data using mutants, we have provided a more convincing model for the activation of TRIM72 E3 ligase on the membrane. While TRIM72 has been shown to degrade insulin receptor substrate-1 (Yi *et al.*, Nat Comms, 2013) and insulin receptor (Song *et al.*, Nature, 2013), its ubiquitination activity and roles in insulin signaling remain controversial (Wang *et al.*, Diabetes, 2020; Philouze *et al.*, PLoS One, 2021). The clear physiological target substrates of TRIM72 have not yet been established, and the connection between E3 ubiquitin ligase

activity on the substrates and membrane binding by TRIM72 is still unclear. Therefore, we believe this issue is beyond the scope of this study.

<<< Finally, the multiple diverse experimental approaches applied by the authors and large amount of data presented make the manuscript very difficult to read and comprehend as often no rationale for a given experiment or link between different approaches is provided. Similarly, the manuscript contains a number of statements that are unclear and not always supported by references or data. >>>

In the revised manuscript, we have tried our best to more clearly communicate our findings to the readers and reviewers. As a very large proportion of the manuscript has been rewritten, it is not easy to highlight the revised text in red. Because the reviewer commented on some grammatical errors in the original manuscript, we have obtained an English editing service from a native English speaker.

<<< Finally the authors should reconsider which Figures to put in the main part of the manuscript and which into the Extended Data section. At present a lot of data that are required to understand the manuscript are in Extended Data whereas some of the main Figures are not central to the story and could be moved to Extended data.>>>

We present new main figures showing the cryo-ET structure (Fig. 3b,c) in the revised manuscript. As the reviewer suggested, several main and extended data figures in the original manuscript have now been replaced, as follows (the revised figure info is on the right):

- **Fig. 1c → Extended Data Fig. 2a**
- **Fig. 1d → Supplementary Fig. 8a**
- **Fig. 2c → Supplementary Fig. 3**

- Fig. 2e → Extended Data Fig. 5a
- Fig. 3a-c → Extended Data Fig. 7c-e.
- Fig. 4a-c,e → Extended Data Fig. 10
- Fig. 4g-h → Removed; instead, “Lips *et al.* (2020) EMBO J.” is cited, and a small simplified inset (upper left corner) is shown in Fig. 6
- Fig. 5 → New Fig. 6

There were too many Extended Data Figures (the maximum limit is 10 figures), so some of the minor data figures have been moved to the Supplementary Figures (8 figures). We hope this rearrangement of figures improves the logical flow of the story.

<<< *Specific points are listed below that the authors should address to ensure that the conclusions they draw are appropriately supported by the data provided.*

Specific points

- Crystallographic data:

The authors present multiple structures of different constructs that crystallized in different crystal forms, in part at very low resolution (resolution limits of over 5 Å and 7 Å), and some with overestimated resolution ranges (PDB:7DI2, CC1/2 of 0.256 in the highest res shell). This doesn't add much to the story and the authors should rather focus on one or two crystal forms with decent resolution limits and statistics.>>

According to the comment, we carefully refined all models again; now, the refinement statistics are slightly better (the PDB IDs have all been updated; please see the new PDB validation files). In the original manuscript, we included all datasets (including the low-resolution data) in the table in a complete list but did not use all of them for structural analysis. Indeed, we mostly used detailed structural comparisons and descriptions with the

2.75 Å resolution dataset. We believe that even at low resolution, the location of the RING domain and global domain movement is undoubtedly assigned. For clarity, the electron density map of the RING domain is included as Supplementary Fig. 1f in the revised manuscript. Again, the low-resolution full-length (FL) structures were only used for the flexibility of the CC domains and the location of the RING domain.

<<< Importantly, the authors need to be absolutely specific in the text and figure legends about which structure they are describing at what resolution. This is particularly important for Figure 1. In its current form, the positioning of the RING in Fig 1b,c is rather misleading as no density could be observed for the RING in most crystal forms, and the authors themselves go on to employ SAXS to demonstrate how flexible the position of the RING domain is. >>>

In the revised manuscript, we have described which structures and resolutions were used (please see the legends of Fig. 1 and Supplementary Table 2). In Fig. 1, the position of the RING domain was essentially based on the full-length structure, and the details are clearly indicated in the revised manuscript.

<<< This is particularly relevant as the authors describe an inactive state of the RING due to interactions with the CCD (see Ext Data Fig 5a). This is rather confusing and needs to be clarified. If the RING domain is flexible it is unlikely to be inhibited by contacts with the CCD. Panels Fig 1c,d could be moved to Extended Data. Inclusion of the membrane model in Figure 1b is misleading and should be removed. >>>

The inactive state of the TRIM72 dimer does not stem from the contact between the RING domain and CCD. Of course, the crystal structure of TRIM72 FL is not an active conformation (most probably), because the RING domain blocks the E2~Ub molecule to form a functional ternary complex. Again, the crystal structures using TRIM72 FL did not

show the electron density of the RING domain except for one FL structure, showing additional evidence for the flexible nature of the RING domain. The flexibility of the RING domain may cause low activity due to a reduced probability of RING dimer formation in solution. This is described in the revised manuscript.

As suggested, Fig. 1c and d have been moved to Extended Data Fig. 2a and Supplementary Fig. 8a, respectively. The membrane model in Fig. 1b has also been removed from the revised manuscript.

<<< Furthermore, the authors need to show electron density of multiple regions in the structures described, given the rather low resolution of many of the structures.

Overall, the authors should be careful in the description of the different crystal forms and how much they read into any differences between the structures observed. Part of this might be solely due to crystal contacts. >>>

Yes, we determined many TRIM72 structures with different crystal forms, though we initially thought this information was somewhat space-consuming. However, as the reviewer requested, we have included the electron density map of multiple regions in the structure as Supplementary Fig. 1 in the revised manuscript.

<<< - The description of the TRIM72 structure contains a long paragraph describing the coiled coil domain (CCD) and multiple associated figures. CCDs of TRIM proteins have been described in detail before – it is not clear what novel insight the authors are providing here. This needs to be made clear, otherwise this section should be condensed. >>>

Thank you very much for your comments. As advised, we have condensed the CCD part significantly in the main text. Most CCD-related figures (Extended Data Figs. 2-4 in the

original manuscript) have been condensed to one supplementary figure, Supplementary Fig. 2a-e, in the revised manuscript.

<<< - SAXS data:

The authors should provide the Rg profiles of the SEC traces of mTRIM72 shown in Ext Data Fig 1c,d. They also need to add the calculated protein molecular weights based on their SAXS dat, to the SAXS table. Do they correspond to the theoretical calculated MW of a dimeric molecule or is there any evidence for aggregation? Why is the Porod volume of mTRIM72 WT so much bigger than of the deltaRING construct (Ext Data Table 3)?

Why did the authors only take flexibility of the RING domain into account, but not of the ends of the CCD, which move “up and down” when carrying out their modelling of the SAXS data?

Would taking flexibility of the CCD into account improve the X2, which is currently quite poor?>>>

As suggested, the Rg profiles of the SEC traces of mouse TRIM72 are included in Extended Data Fig. 1c,d. We have also added the calculated protein molecular weights based on the SAXS data to Supplementary Table 4.

There was no detection of aggregation. We have reanalyzed the SAXS data, and now, the discrepancy between the Porod volume of WT and Δ RING is considerably decreased (though it is still larger than the calculated value, perhaps due to the flexible nature of the exterior region of the molecules (RING domain and the end of the CCD).

We considered the flexibility of the ends of the CCD in addition to the RING domain, and the RING domain is more flexible than the CCDs. The movement of CCD ends was not represented by the SAXS profiles, and thus, it is a difficult problem to solve with SAXS-based computation simulation, including SREFLEX (Panjkovich and Svergun, Phys. Chem.,

Che., Phys., 2016).

In summary, the SAXS data only represent the flexibility of both ends of the TRIM72 molecules, including the RING domain, which is also supported by the lack of an electron density map in several crystal forms of TRIM72 FL.

<<< - *Role of the PRYSPRY domain in lipid binding: The observation that the PRYSPRY domain of TRIM72 has lipid binding properties is very exciting and could point to a completely new role for this protein interaction module. To provide a more general context the authors should compare the electrostatic surface potentials of other PRYSPRY domains for which structural information is available. Could this be a more general property of this domain? This is particularly relevant as PRYSPRY domains have also been suggested to bind RNA. How does the surface involved in membrane association relate to the ligand binding sites of PRYSPRY domains identified so far? This novel property of the PRYSPRY domain is an important section of the manuscript that should be further developed.* >>>

As suggested, we compared the structures of PRYSPRY domains from various TRIM and TRIM-like proteins and have described the results in the revised manuscript with a new figure, Extended Data Fig. 3. The structure-based sequence alignment with functional analysis indicates that the variable loop (VL) regions are critical for substrate recognition, especially VL1, VL3-β6-VL4 and VL6. In contrast to other PRYSPRYs, the TRIM72 PRYSPRYs show a broad distribution of basic residues in VL regions, which contributes positively charged surfaces for membrane binding.

In view of the evolution of PRYSPRY domains, TRIM50, TRIM62, TRIM69, and TRIM72 are located close to each other phylogenetically (D’Cruz et al., Protein Sci., 2013). The cellular localization of TRIM50 is cytoplasmic vesicles, and it is known that the

PRYSPRY domain of TRIM50 recognizes phosphoinositide (Nishi et al., Journal of Biol. Chem., 2012). TRIM69 in sperm flagella interacts with the cation CatSper channel, implying that it plays a role near the membrane (Zhao et al., Nature comm., 2022).

The other TRIM-family proteins have been known to bind diverse biological molecules, as the reviewer commented. In the case of TRIM25, an RNA binding domain (RBD) immediately before the PRYSPRY domain has been identified (Choundhury et al., BMC Biology, 2017, Haubrich et al., BioRxiv., 2021). However, there is no RNA-complex structure, and the PRYSPRY domains of RIPLET and TRIM65, known as RNA-binding TRIM proteins, interact with dsRNA helicase (Kato et al., Mol Cell, 2021). When RIPLELT and TRIM65 interact with RIG-I and MDA5, the surface of the PRYSPRY domain, called surface A (Woo et al., EMBOJ, 2006), is utilized for protein–protein interactions. Interestingly, the surface shares the same membrane binding site as TRIM72; however, the surface properties of the VLs are different. Furthermore, the PRYSPRY domain of TRIM21, an Fc receptor binding TRIM protein, possesses a surface at a similar location to TRIM72 with hydrophobic properties, which is critical for Fc recognition (James et al., PNAS, 2007).

In summary, we conclude that the ligand binding surface of PRYSPRY in TRIM72 is shared with that of other TRIM proteins position-wise; however, the detailed shape and charge properties formed by the compositions of the residues in VLs are different depending on different ligands. This information is described in the Discussion section in the revised manuscript and with a new figure, Extended Data Fig. 3.

<<< - Lipid binding assays:

The total amount of TRIM72 in the membrane fraction between TRIM72 WT and the K2D mutant is very similar, why do the authors think there is no apparent binding of the mutant protein in the SPR experiment?>>

Although the band intensities of the membrane fraction between TRIM72 WT and the K2D (K²D in the revised manuscript) mutant were similar in western blotting, the total expression level (membrane + cytosol) of the K²D mutant was relatively higher than that of the WT (we do not know the exact reason). Therefore, it is not appropriate to directly compare the membrane fraction between TRIM72 WT and K²D, and the key factor is their ratio. As shown in Fig. 2e and Supplementary Fig. 3, TRIM72 WT exists exclusively in the membrane, but the defective mutants localize less than 40% in the membrane and remain mainly in the cytosolic fraction of C2C12 myoblast cells.

<< The SPR-based study showed that mTRIM72 binds to PS-SUVs with very high affinity (9.9 nM). This affinity is entirely reliant on the PRYSPRY domain, which however needs to be dimeric. The authors show this by employing a GST-tagged hTRIM72 PRYSPRY domain, which has been calculated to have an affinity of 21.3nM. However, this is not what the data shown in Ext Data Fig.10g suggest as a much higher concentration of protein is required to reach saturation as with the full-length WT protein (panel 10a). How do the authors explain this? And why do the proteins in panels 10e and f produce such strong signals?

Our GST-tagged hTRIM72 PRYSPRY domain is indeed GST-H₃-PRYSPRY (we apologize for this misleading error, and it has been corrected in the revised Extended Data Fig. 5). The H₃ helices are critical for the unique orientation of the two PRYSPRY domains. In contrast to the full-length TRIM72, which is extremely large due to the CCDs, GST-H₃-PRYSPRY is relatively globular and compact; therefore, the GST-H₃-PRYSPRY protein might be more immobilized in the same area of the sensor chip. The signal (R.U.) is not directly proportional to the binding affinity (many factors are involved, such as the mass on the surface and the number of molecules bound to the surface) because the immobilized proteins are not the same (this case, TRIM72 vs. GST-H₃-PRYSPRY).

Extended Data Fig. 5e and f (original, Extended Data Fig. 4e & f) did not show such a strong signal. The critical aspects are the k_a and k_{off} values, and in both cases, these proteins were immediately released from the PS-liposome chip.

<<< Given the high affinity of TRIM72 for PS-SUVs measured by SPR analysis why is the proportion of TRIM72 comigration with PS-SUVs so small (Extended data Fig 11)? >>>

As explained above, the affinity (apparent affinity - avidity) value obtained from the SPR experiments may be overestimated, but the affinity between TRIM72 and PS-SUVs is comparable with that between Ca^{2+} -dependent annexin-V and PS, ranging from 15 to 0.03 μM (Appelt et al., Cell Death & Diff., 2004). In the revised manuscript, we showed that the B-box mutants crucial for the higher-order TRIM72 assembly abrogate the membrane binding (Fig. 3g). It indicates that the membrane binding affinity measured by the SPR analysis is the sum of the direct interaction with the PS liposomes and the oligomerization of TRIM72 molecules.

When we reconstituted the mTRIM72-proteoliposomes (Extended Data Fig. 11 in the original manuscript; Extended Data Fig. 10d-f), the excess amount of TRIM72 was mixed with PS-liposomes to maximize the proteoliposome species. Therefore, there were large amounts of dimeric TRIM72 without vesicles in the gel filtration.

<<< - Catalytic activity and ubiquitination assays:

Based on the data presented by the authors in Extended Data 11 the linchpin Q57 is responsible for the lack of activity of TRIM72 and its replacement with R completely negates the need for membrane-induced oligomerisation. However, some RING E3 ligases are active without containing a conventional linchpin residue (e.g. TRIM32, Extended Data Fig 5a). How do the authors explain this? >>>

The role of the linchpin residue is to lock the tertiary orientation of the ubiquitin-E2 enzyme-RING ligase for efficient activity. We believe that membrane binding triggers the functional alignment of the TRIM72 molecule on the membrane and reduces the flexibility of the RING domain (because of space restriction) to obtain the right orientation of the above components, which must be a RING-RING intermolecular dimer. As indicated in the new data (Fig. 4e), RING dimerization is also another key factor in the activity of TRIM72. Although the RING domain of dimeric TRIM72 protein in solution exists as a monomer, the mutant with disruption of RING dimerization, L74R, possesses no (or extremely weak) activity, similar to TRIM25 RING (Koliopoulos *et al.*, EMBO J, 2016). In contrast to TRIM25, TRIM72 has a glutamine linchpin, and its ubiquitination activity is extremely low in solution. Therefore, in the case of TRIM72, the RING-RING dimer is invoked by the higher-order assembly on the membrane and obtains functional Ub transfer with E2~Ub (UBE2D in this study).

However, in the case of TRIM32 (with a suboptimal serine linchpin), its RINGs have strong dimerization affinity in solution. Not only the monomeric RING variant of TRIM32 but also the point mutant with disruption of the dimeric interface of RING abrogates the ubiquitination activity with UBE2D or UBE2N (Koliopoulos *et al.*, EMBO J, 2016). Therefore, TRIM32 obtains Ub transfer activity through apparent binding affinity to E2~Ub, which is induced by solid dimerization between RING domains.

<<< Furthermore, in light of these data, what do the authors speculate is the molecular mechanism by which membrane binding induces TRIM72 WT activity? They do not provide any insight into this. Many TRIM E3 ligases need to dimerize to be active but this does not appear to be the case here as the authors have shown that fusion of two RING domains is not sufficient to establish catalytic activity, as has been shown for other TRIMs such as TRIM5alpha. What do

they suggest is the mechanism of RING activation? If the linchpin residue was indeed the reason for the lack of activity, how can membrane association restore activity? >>>

The fusion of two RINGs did not enhance activity, which might have been due to the low activity of the glutamine linchpin and improper topology of the RING dimer. As mentioned, RING dimerization is essential for Ub transfer in almost all TRIM RING E3 ligases (Fiorentini et al., Biochemical Society Transactions, 2020). The N-terminal and C-terminal helices of the RING domain consist of a four-helix bundle involving conserved hydrophobic residues (e.g., Val72 in TRIM25; the equivalent residue in TRIM72 is Leu74). We performed an activity assay using mutants (Q57R and Q57R/L74R) that affect ubiquitination activity and concluded that TRIM72 molecules are oligomerized on the membrane and that the RING domain is dimerized for E3 ligase activity.

In the revised manuscript, we have added the activation model of TRIM72 on the membrane (Fig. 6). The RING-RING dimerization on a particular orientation coupled with reduced flexibility of RING must be critical for the E3 ligation activity, as described above. This spatial preference is achieved by membrane binding and oligomerization, which is presented with the cryo-electron tomography structure. We hope the model shown in Fig. 6 in the revised manuscript provides insight into the activation mechanism of TRIM72 on the membrane.

<<< - I do not understand the experiment in which the dimeric form of TRIM72 can be “activated” by the oligomeric form. How would this activation work? Isn’t the observed activity just coming from the “oligomeric form”? Why is there such a significant difference in apparent activity between the 2 dimeric samples in Extended data 16e that have not been incubated with “oligomeric” TRIM72 (neither no heat or heat treated)? Should they not be identical? >>>

We have provided the activation model of the oligomeric form in the revised Fig. 6. The basic idea is as follows: once small TRIM72 oligomers (like seeds) are formed on the membrane, the remaining TRIM72 dimer in solution might be recruited into the oligomers to make a higher-order assembly.

Extended Data Fig. 10c (Extended Data 16e in the original manuscript) represents the oligomeric form as a seed for the creation of higher-order active oligomers. The hWT fraction (oligomeric peak) without heat treatment can act as a seed to produce more active oligomers by recruiting the dimeric fraction into the membrane. In contrast, the same hWT fraction (oligomeric peak) with heat treatment (90 °C for 10 min) cannot act as a seed because the oligomeric fraction must already be destroyed by the heat. Therefore, more active oligomers cannot be formed by adding dimeric molecules, as shown in Lanes 5-8.

<<< - *Figure 4: How did the authors check that the hTRIM72 purified from mammalian cells used in this experiment is microsome bound? And given that the protein is purified from mammalian cells – how can the authors exclude that the ubiquitination activity seen does indeed come from TRIM72 and not another protein that has copurified in the microsomes? Similarly, it is not clear if hTRIM72 purified from HEK293T cells is membrane associated or not. There is no detergent in the lysis or purification buffer. Please specify as this is crucial for the interpretation of the results.*

First, we must explain how we purified the hTRIM72 protein. The Strep-tagged hTRIM72 was expressed using mammalian HEK293T cells, and then the protein was purified by StrepTactin-conjugated beads as a first step. Next, the partially purified Strep-tagged proteins were applied to the gel filtration column. The elution profile provided the two major peaks of high and low molecular weights, as shown in Fig. 4a (revised manuscript). Both fractions were TRIM72 proteins, and the high-molecular-weight fraction contained a lipid moiety (Fig. 4), as confirmed by transmission electron microscopy (left panel in

Extended Data Fig. 7c). The low-molecular-weight fractions were dimeric forms based on the elution volume (Extended Data Fig. 7b).

Therefore, we did not use any detergent for the lysis or purification buffer, and the high-molecular-weight hTRIM72 fraction comigrated with vesicles from the cells, showing that the hTRIM72 molecules bound to the microvesicles (right panel in Extended Data Fig. 7c). Of course, there is a possibility that TRIM72 interacts with membrane-bound proteins in microvesicles, but we think that hTRIM72 associates with the membrane through PS based on our *in vitro* binding affinity assay. The SDS–PAGE results support the direct interaction because the comigrated fraction contained mainly TRIM72, not the other proteins.

Furthermore, TRIM72 mutants lacking the RING domain or defective in zinc coordination did not comigrate with microvesicles. Thus, the correctly folded oligomeric TRIM72 proteins only interacted with vesicles. The activity assay with the Q57R mutant clearly showed enhanced ubiquitination activity compared to that of the WT, which confirms that the E3 ligase activity originates from TRIM72 (WT or mutants). One more piece of evidence is that when we added inactive dimeric TRIM72 to the tiny proteoliposome fraction, the ubiquitination activity increased in a dose-dependent manner (Extended Data Fig. 10c). In summary, the ubiquitination activity is from TRIM72 bound to microvesicles.

In a previous report (Lee *et al.*, CDD, 2010) as well as in our results, TRIM72 was isolated and identified from the lipid raft fraction, but TRIM72 also exists in microvesicles and the sarcolemma (Cai *et al.*, JBC, 2009; Cai *et al.*, Nat Cell Biol., 2009).

<< *It would also be interesting to carry out a mass spectrometry analysis of hTRIM72 purified from HEK293T in order to investigate if the protein may contain post-translational modifications*

such a phosphorylation that could influence its properties. >>>

Our proteomic analysis of TRIM72 isolated from lipid rafts showed that there was no phosphorylation, and Pro-Q Diamond staining of mammalian TRIM72 showed no evidence of phosphorylation. There is a report of ADP-ribosylation of arginine residues (R207 and R260) of TRIM72 (Ishiwata-Endo *et al.*, JCI Insight, 2018), but the protein expression of ADP ribosyltransferase 1 (ART1) is muscle-specific and absent from HEK293T cells. Therefore, the ubiquitination activity was not controlled by posttranslational modifications in our experimental systems.

<<< - Overall, the authors should provide more details about the experimental methods, especially with respect to the protein preparations that have been used for electron microscopy. It is not clear what preparation has been used for which images, and what precisely the “higher-order oligomer fractions of hWT” (line 628) are.

What precisely is the difference between the microsomes versus reconstituted liposomes? >>>

We have described the experimental methods in more detail in the revised manuscript. The microvesicles were isolated from mammalian cells directly, and the vesicles may have contained other cellular components and different lipid compositions from the reconstituted liposomes. Our reconstituted liposomes were generated with defined components, which are described in the Methods section (Liposome Preparation). In our opinion, the minimal components that interact with TRIM72 molecules are all in the reconstituted liposomes, and thus, the reconstituted liposomes mimic the cellular function of TRIM72 with microvesicles.

<<< - Electron microscopy

The authors provide some good and interesting negative stain images and cryo-tomograms.

However, in their current form they are preliminary, and a number of issues should be addressed to ensure that the conclusions drawn, and models suggested are fully supported by the data provided. Specifically, the authors should provide clear EM evidence that support a specific orientation for membrane interactions with the PRYSPRY domains pointed at the membrane and a specific architecture for their higher-order assembly (the 3D volume rendering shown in Fig 3 does not provide this information). For example the authors could carry out a direct comparison of a calculated density from their crystallographic model (or packing model from within the crystal) with negative stain images looking down on the membrane that shows the right size and shape (Extended Data Figure 13) or correct spacing for the higher-order assembly. Similarly, calculated density from the X-ray model matching either cryo-EM (images) or tomograms showing detailed features at the edge of liposomes. At a minimum, some evidence of dimensions matching or some similarity of density distribution to the crystallographic model should be provided. A more definitive proof would be obtained by subtomogram averaging (averaging of density volumes within the tomograms). Similarly, for the liposome bridging structure, the overall dimensions may be correct for one of the crystallographic packing models in Extended data 15d. But there is insufficient EM evidence presented to prove this, maybe the proposal could be tested by mutations that disrupting the interface that makes the bridges. >>>

Based on the reviewers' (#1 and #3) suggestions, we tried to visualize the membrane-bound structure using cryo-electron tomography and subtomogram averaging. We spent considerable time and effort generating an oligomer structure on the membrane. The subtomogram averaged map had a much higher resolution than the negatively stained and cryo-EM images in the original manuscript; thus, a more convincing model is now presented in the revised manuscript. We have provided new Fig. 3b,c, Extended Data Figs. 8 & 9, and Supplementary Table 3 regarding cryoET data in the revised manuscript.

The membrane-bound TRIM72 oligomer structure obtained by subtomogram averaging matches well with one of the crystal packing models. The crosslinking mass spectrometry (CLMS) results were clearly explained by the higher-order TRIM72 assembly model (Supplementary Figs. 5 & 6). Therefore, given the results of the activity assay using the designed mutants (Q57R/L74R and Q57R/R207E), we were able to propose an activation mechanism of TRIM72 on the membrane (Fig. 6).

<<< - The authors need to run their SEC-MALS experiments at different protein concentrations to investigate if the protein forms higher order oligomers at higher concentrations. In fact, the molecular weight distribution shown in Ext Data Fig 1b suggests it does. >>>

As suggested, we performed SEC-MALS experiments at three different protein concentrations (2.75, 5.6, and 11.2 mg/ml) and showed dimers always in solution (Extended Data Fig. 1b in the revised manuscript). Without PS-vesicles, there was no oligomer fraction in gel filtration.

<<< - In general, members of the TRIM family of E3 ligases have not been described as membrane associated proteins. The authors should provide a short paragraph in the introduction about what precisely is known in the literature about the membrane-binding properties of TRIM72, and how this relates to its described physiological function. >>>

There are several reports on the membrane interactions of TRIM family proteins. TRIM50 (Nishi *et al.*, JBC, 2012) and TRIM72 are membrane-associated, and the C-terminal transmembrane region of TRIM13 has been characterized (Li *et al.*, Sci Adv., 2022). The physiological function of TRIM72 is membrane repair (Cai *et al.*, Nat Cell Biol., 2009), and the data in the literature clearly show the membrane localization of TRIM72 (there are more than 30 papers regarding the membrane repair function of TRIM72). TRIM family

proteins interact with a variety of cellular proteins or RNAs for their specific functions, and we have visualized the higher-order TRIM72 assembly on the membrane. Our study provides insight into the membrane repair function of TRIM72, although the link between E3 ligase activity and membrane repair remains vague. Very recently, TRIM72 was reported to be one of the proteins found in extracellular vesicles (Watanabe *et al.*, PNAS Nexus, 2022).

<<< - *Experiments presented in this study have been carried out either with human or mouse TRIM72, and it is not always clear why one protein was used over the other. This is particularly problematic as the authors state that the human protein purified from mammalian cells exists in dimeric and oligomeric forms (line 214). How does this influence the experiments? >>>*

Human TRIM72 was always prepared with a mammalian expression system, and mouse TRIM72 was prepared with an *E. coli* expression system. Only mTRIM72 was crystallized; thus, mTRIM72 expressed in *E. coli* exists in dimers without liposomes, but hTRIM72 expressed in mammalian cells exists in both dimers and oligomers with vesicles. As shown in Fig. 4a and Extended Data Fig. 7b, the dimer is in solution alone, and the oligomer coexists with the microvesicles. Using gel filtration experiments, we always separated these two states and performed the subsequent experiments.

<<< *Minor points:*

Abstract: what is “a characteristic pair of PRYSPRY domains”? On its own this statement doesn't make sense. >>>

We have removed the word “characteristic” from the Abstract.

<<< Line 49: What is meant by “The most recently discovered Ub ligases”? >>>

We have rewritten the introductory part, and the sentence is no longer in the revised manuscript.

<<< Line 76: “Ub ligase activity is also controversial due to a lack of biochemical and biophysical information.” What is controversial: if it has activity, what types of chains it makes or the substrates? Please be more precise. Neither of the references cited describe the ubiquitination properties of TRIM72 (41, 42). Similarly, please do not describe TRIM72 as a TRIM E3 ligase with auto-ubiquitylation activity (lines 62 and 66), after you stated in the abstract that it is inactive in solution, or at least qualify the statement. >>>

Although no similar sentences are left in the revised manuscript after extensive rewriting, we would like to give some explanations. The controversial issue is that TRIM72 (MG53) has been reported to be an E3 Ub ligase against IRS-1 and the insulin receptor; however, several reports have been published indicating that MG53 is not involved in insulin signaling. Honestly, in our laboratory, TRIM72 did not degrade the IRS-1 protein. We initially thought dimeric TRIM72 in solution possessed no activity and thus no ability to degrade IRS-1. However, IRS-1 was also not degraded by the membrane-bound active TRIM72 oligomer.

TRIM72 has been reported as an E3 Ub ligase, although evident ubiquitination activity has not been detected *in vitro*. However, our paper is the first to show that TRIM72 has ubiquitin transfer activity on the membrane in the form of a higher-order structure but not in solution.

<<< The nomenclature used by the authors for their constructs (lines 91/92) is not very

intuitive. Do the authors have data that indicate that the mutant mouse TRIM72 constructs that are being used throughout the manuscript behave as the WT proteins? >>>

The details of the renamed constructs are described in Supplementary Table 1. To examine the behavior of the mutants, we performed circular dichroism (CD) and thermal shift assays for WT and mutant proteins. They were virtually the same, which is described in the Methods. We determined the full-length crystal structure, solution SAXS envelope, and cryoET model using WT proteins. Three free cysteine residues were mutated to serine residues, which is very minor. Some of the mutations were K279H and A283H at the H₃ helix region, which was designed based on the structure of human H₃-PRYSPRY (Park *et al.*, PROTEINS, 2010). The structure of the H₃ helix of the current mutant is identical to that of the previous domain.

<<< Line 122: What point are the authors trying to make when they talk about the “appropriate distance” of the zinc-binding modules to the CCD? >>>

We agree that this was confusing for readers, so we have removed this sentence from the revised manuscript.

<<< The authors should provide a figure in extended data in which they highlight the loop that has been removed in construct delta272-281. Do they have any evidence to show that this protein is properly folded? >>>

Yes, residues 272-281 form a unique helix, H₃, and we have several lines of evidence that this mutant folds properly. The Kratky plot of SAXS using the mutant (see below) showed that this protein was properly folded. The elution profile of gel filtration chromatography was almost identical. The T_m value of the deletion mutant was slightly higher than that of

the WT but very similar to those of the F274A and W277A mutants. Therefore, we believe that there is no folding defect in the deletion mutant.

<<< Legend Ext Data Fig1: The authors do not provide “high resolution crystal structures” >>>

We have rewritten Extended Data Fig. 1. The phrase “high resolution crystal structures” is no longer used. We simply meant “fit the crystal structure (relatively high resolution) into the SAXS envelope”.

<<< Could the authors please elaborate on how “The unique architecture of TRIM72 allows its promiscuous interaction with any membrane shape if the membrane is enriched in PS.”? >>>

We thought that the unique shape and flexibility of the coiled-coil structure of TRIM72 enabled it to interact with convex, concave, and flat membranes containing high concentrations of PS. Our current data do not directly visualize the promiscuous interaction; thus, we have toned down the wording.

<<< Ext. Data Fig.5d: Why did the authors align the B-box domain with RING domains? What point are they trying to make? >>>

We wanted to show some sequence similarities between the B-box and RING domains.

They turned out to be not critical for the flow of this manuscript; therefore, we have removed that Extended Data Fig. 5d from the revised manuscript.

<<< The comparison of the TRIM72 structure with a bird is rather detracting from the description. >>>

According to the reviewer's comment, we have removed the description of the bird-like structure.

Responses to Reviewer #2

<<< Reviewer #2:

Remarks to the Author:

In this manuscript, Park et al showed full-length structures of TRIM72, an E3 ligase with an unusual activity of binding lipid membrane. The structure revealed the overall architecture of TRIM72 and a number of interesting features. While RING is flexible, it is tethered to CC in the autorepressed state that prevents RING from engaging with E2. Membrane binding leads to higher order oligomerization of TRIM72, which was accompanied by significant increase in E3 ligase activity. The authors further argue that a subset of TRIM proteins evolved to adopt Gln, instead of Arg, at the linchpin position of RING to suppress its activity in the absence of higher order oligomerization.

Overall, the structural and biochemical data in this manuscript are novel and interesting, and provide unique insights into the mechanism of a highly unusual E3 ligase that recognizes membrane. However, I feel that the manuscript needs more work in strengthening their models. While the data are expansive as a whole, data supporting each aspect of the mechanisms remain somewhat weak as illustrated below. >>>

Thank you for being interested in our work, and we have tried our best to address your concerns.

<<< 1. Ring repression and activation mechanism

It is somewhat confusing what conformation RING is in the dimeric formation. In some area of the manuscript, RING is thought to be flexible. However, in other areas, the authors argue that RING is tightly bound by CC, and thus preventing the interaction with E2. Which one is it? Is a mutation in CC that releases RING leads to spontaneous activation? And how is the higher order oligomerization of TRIM72 helps RING get released from this autorepression? If interdimeric RING:RING interaction is the mechanism of multimerization-dependent activation, mutations in the RING:RING interface (which one can predict based on other RING dimeric structures) should abrogate TRIM72 ubiquitination activity. >>>

We appreciate these valuable comments. The experimental data suggested by the reviewer have been added and have markedly strengthened our revised manuscript.

We regret that the description in our original version was quite confusing to the reviewers. Due to the suboptimal linchpin (glutamine at residue 57), the TRIM72 RING domain shows extremely low ubiquitin transfer activity. The low ubiquitination activity is not derived from the steric hinderance between the RING and CCD as shown in the crystal structure because RING domains are highly flexible in solution. Furthermore, as suggested by Reviewer #2, the mutation in the interface between RING and CCD (Q57R vs Q57R/R207E) showed no difference in the ubiquitination activity.

As shown and described in the revised manuscript, the particular RING-RING interaction induced by oligomerization on the membrane showed enhanced enzymatic activity, an even more drastic effect than R-linchpin mutation. Based on the valuable suggestion, we

generated an L74R mutant (with an equivalent hydrophobic residue located at the RING dimer interface of the known RING dimer structure) to disrupt the potential RING-RING interface and performed a self-ubiquitination assay. The mutant showed no activity (even with the Q57R background), as shown in Fig. 4e in the revised manuscript, suggesting that RING-RING dimerization on the membrane platform must be critical for the activation of TRIM72. The oligomeric structures mediated by the intermolecular interaction of two B-boxes form an active intermolecular RING dimer on the membrane. This information has been clearly written in the appropriate parts of the revised manuscript.

<<< 2. Higher order oligomerization

This is really an important part of the paper, but I had a hard time understanding the text and related figures. In particular, it is very difficult to understand how the X-linking data in Fig 3d led to the model in Fig 3e. Which arrows indicate which crosslinkings? It is also hard to see the red balls (indicating the reactive Cys). Can the authors separately indicate X-linking between RINGs, B2s and CCs? Perhaps Cys in different domains should be in different colors? >>>

As suggested, we have rewritten the description of the X-linking data for clarity in the revised manuscript (CLMS results - appropriate parts on Pages 14, 15, and 20; Fig. 3f and Supplementary Fig. 6 and their legends). The red lines on the left (with liposomes) of Fig. 3f are unique X-linked cysteines, which suggests that the cysteines (C53, C55, and C56) in the RING domain participate in the bridge to the cysteines in the neighboring molecules on the liposome. Similarly, cysteines (C105 and C108) in the B-box domain bind to the cysteines in the neighboring TRIM72 molecules. Cysteine 144 in the H₁ region of CC shows the same phenomenon. All these data imply that the RING, B-box, and CC domains in dimeric TRIM72 are more accessible to the interaction with the RING, B-box, and CC domains of neighboring dimeric TRIM72, as illustrated in Supplementary Fig. 6. We absolutely agree

with the suggestion that the cysteine residues in the different domains should be different colors to facilitate understanding the model shown in Supplementary Fig. 6a.

<<< *What is the physiological concentration of PS in cellular membrane (organelle vs. plasma membrane)?>>>*

It is known that the concentrations of PS in the ER and plasma membrane are approximately 2.5~8.5% and 12%, respectively (Leventis and Grinstein, *Annu. Rev. Biophys.*, 2010). Furthermore, the concentrations of PS in the cellular membrane are not uniformly distributed. In particular, it is known that the PS molecules are enriched in the inner leaflet of the plasma membrane and that the local concentration of PS molecules in the region must be higher than average. Furthermore, variable concentrations of PS are quite frequently used for *in vitro* membrane experiments (Lin *et al.*, *Nature*, 2020 [Annexin-V: ~50% DOPS]; Drin *et al.*, *Nat SMB*, 2007 [ArfGAP1: 30% DOPS]; Grushin *et al.*, *Nat Comms*, 2019 [Synaptotagmin: 34 & 80% DOPS], etc.).

<<< *Could the TRIM72 clustering an artifact of having an excessive concentration of PS in the in vitro liposome assay or microsomes purification/extraction? >>>*

We confirmed TRIM72 oligomerization using the PS-liposomes containing 30 mol% PS (Extended Data Fig.6). It indicates that the oligomerization occurs in the physiological concentration of PS *in vitro*. Furthermore, we did not add any PS molecules for purification and extraction of microvesicles. We used a Strep-tagged construct to purify human TRIM72 using mammalian cells. For the first step, biotin beads were used, and the gel filtration results showed high-molecular-weight fractions containing TRIM72 and microvesicles (Fig. 4a in the revised manuscript). We believe this is not an artificial event as we purified TRIM72 proteins under very normal conditions without detergents or

questionable compounds. In addition, the self-assembly of TRIM72 was already reported (Cai *et al.*, Nat Cell Biol, 2009). To show the TRIM72 assembly, cryoET density maps were resolved using subtomogram averaging in the revised manuscript. We also found that B-box mutations at the interface abolished the membrane binding of TRIM72. Therefore, we are confident that the high-order TRIM72 assembly is not an artifact.

<<< Membrane-mediated oligomerization: does it simply reflect TRIM72 binding to pre-organized clustering of PS within the liposome or microsome or does TRIM72 induce clustering of PS?

In our view, it might not be easy to induce the clustering of PS by TRIM72. TRIM72 could not interact with liposomes in the presence of 10 mol% PS, in contrast to 30 mol% PS. If TRIM72 induces the clustering of PS, a high local concentration of PS can be achieved by TRIM72. Therefore, we believe that the TRIM72 molecule interacts with a preorganized cluster of PSs.

<<< What domain is required for TRIM72 oligomerization upon membrane binding? Based on X-linking, one would expect that B-box, CC and RING may play an important role? >>>

As described, all domains of TRIM72, including PRYSPRY, play an important role in oligomerization. The PRYSPRY domain is critical for the interaction with the PS-enriched membrane, together with the correct orientation of H₃ helices. Mutations disrupting the intermolecular interactions of B-boxes result in no membrane binding. When we introduced the M138A mutation, which is important to the intermolecular interaction of CCDs, the mutant interacted with the membrane but could not form the oligomer (Extended Data Fig. 8a). Interestingly, the M138R mutant showed augmented membrane binding and oligomer formation. Finally, the L74R mutant with altered RING dimerization

also showed a lower binding affinity to the membrane and could not form the oligomer. Thus, the RING domain also plays an important role in oligomerization. Therefore, all domains of TRIM72 are directly or indirectly involved in TRIM72 oligomerization on the membrane.

<<< 3. Membrane curvature recognition

The presented data do not appear to provide any information about membrane curvature recognition. In fact the authors also noted in lines 236-237 that TRIM72 can recognize any membrane shapes.. In this case, a better way to phrase this would be that TRIM72 recognizes PS (and perhaps clustered PS) independent of the membrane shape, as it was found to localize on both convex and concave side of the membrane in this study. As is currently written, it suggests that TRIM72 cannot recognize flat membrane structure, for which the authors have no data for. It is also unclear whether this is relevant for the biological function of TRIM72. Wouldn't the local concentrations of PS/PI be a more relevant information for membrane specificity of TRIM72? Is this an important point to make, and do authors have sufficient data to conclude in one way or another? My guess is no. >>>

We appreciate the suggestion. Yes, it is best to remove the wording that is not supported by the data and to rephrase the sentence. Flexibility of CCD in the up-and-down direction is what we observed; thus, we speculated that TRIM72 may interact with any type of membrane curvature, be it convex, concave, or flat. Only this possibility is briefly noted in the Discussion.

<<< 4. Ub ligase activity

Fig 4 is interesting but I do not believe it was properly performed to allow accurate comparison between dimer vs. multimer (liposome-bound) or between WT vs. mutants. And this is because the comparison was done without any consideration of the TRIM72 quantity. I would strongly

suggest that the authors repeat this experiment with the same concentration of TRIM72 for WT vs. mutants, and in free or liposome-bound state and run the Ub conjugation reaction all in the same gel to allow quantitative comparison among the samples. Without such side-by-side comparison, I don't think the authors' conclusions are justified. >>>

We quantitatively reperformed the same experiments, as shown in Fig. 4 in the revised manuscript. In the revised Fig. 4c and e, the same amounts of TRIM72 molecules (WT vs. mutants) were used, and the same gel is shown as a side-by-side comparison.

<<< Ext. data Fig 16e: I am not sure what the data in this figure actually offer. While the authors argue that the result shows that the dimeric form of TRIM72 can be activated by the oligomeric form, what was added is the catalytically active oligomer of TRIM72 and the liposome, so it is unclear how this data supports the authors argument. >>>

What we would like to argue is that the small amount of TRIM72-proteoliposome (TRIM72 vesicle-bound) can be a seed for recruitment of inactive dimeric TRIM72 (TRIM72 free-soluble) to liposomes and then cause it to become an active species (No Heat lanes in Extended Data Fig. 16e in the original manuscript). The activity of TRIM72 increased in a dose-dependent manner (after adding the inactive species). In contrast, the same TRIM72-proteoliposome with heat treatment could not be a seed, for two possible reasons – the liposome was destroyed or membrane-bound TRIM72 was denatured. When we added more inactive dimeric TRIM72 (TRIM72 free-soluble) into the heat-treated fraction, there was no poly-Ub production (Extended Data Fig. 10c in the revised manuscript).

<<< 5. Lipid repair and oxidation

Link to oxidative damage (Ext. Data Fig 12g) is not clear since PS induces oligomerization even

in the absence of H₂O₂. If the authors want to say anything about the impact of oxidative damage, the experiment has to be a comparison of +/- H₂O₂ in the same gel. >>>

Yes, it is not particularly essential for the PS story (in the revised manuscript, the panel has been removed). The reason we performed this experiment is that the oligomerization of TRIM72 has been reported in the previous literature (Cai *et al.*, Nat Cell Biol, 2009); thus, we performed the same experiment with or without H₂O₂. Our data showed that a primary factor for oligomerization is the concentration of PS but not the oxidizing reagent.

<<< 6. PRY-SPRY architecture

While deletion of H3 completely impaired PS liposome binding of TRIM72, despite harboring intact PRY-SPRY, GST-PRY-SPRY displayed relatively strong affinity for PS liposome. The latter seems to suggest that PRY-SPRY does not need to be rigidly held by CC for lipid recognition. Why then the deletion of H3 lead to the loss of lipid binding? If one makes a more sophisticated mutation in the interface between CC and PRY-SPRY (rather than deleting a large portion of the protein), what happens to lipid binding, high-order oligomerization and ubiquitination? >>>

We regret that we could not describe it in more detail. Indeed, the PRY-SPRY and GST-PRY-SPRY constructs contained the H3 helix. The names of the constructs have been corrected to MBP-H₃-PRYSPRY and GST-H₃-PRYSPRY (Extended Data Fig. 5). The H₃ helices from each monomer form a particular 4-helical bundle structure to correctly orient whole PRY-SPRY domains toward PS liposomes, as shown below (an alphafold model of the GST-H₃-PRYSPRY construct). Not only the deletion but also the F274A/W277A mutants did not bind to the PS-liposomes (Extended Data Fig. 9a in the revised manuscript). The T_m values by thermal shift assay were very similar between deletion and point mutants, although they showed slightly lower values than the WT. Therefore, the H₃ region is critical for the proper orientation of the PRYSPRY domain to

the membrane. If TRIM72 cannot interact with the membrane, there is no oligomerization or ubiquitination.

<<< 7. Overall, the manuscript has limited description of the experiment performed in either main text or figure legend. More detailed description is necessary (besides the methods section). For example, it is difficult to understand which experiment utilized microsome vs. liposome. For EM images of microsome (in Ext. Data Fig 13 a-b), how do the authors know whether the highlighted structures are from TRIM72 (vs. other cellular proteins)? Shouldn't it require some form of antibody-mediated labeling (e.g. gold labeling)?>>>

As the reviewer requested, we have provided a more detailed description (main text, Figure Legends, and Methods section) in the revised manuscript. Furthermore, for better understanding, the schematic drawings of the model have been included in the many revised figures (Figs. 2c-e, 3f, 4 & 5b; Extended Figs. 7b & 10d-f). We hope the issue of the identity of the TRIM72 oligomer in the low-resolution EM images is now clarified by the newly added subtomogram averaging images (Fig. 3b,c and Extended Data Figs. 8 & 9) in the revised manuscript.

<<< 8. The manuscript will benefit from more careful editing as I found several areas difficult to understand, some with grammatical errors (e.g. lines 115-116, 183-185).>>>

The manuscript has been carefully edited by a native English editor at a professional English editing company. We hope our manuscript is much easier to follow as a whole, including in Lines 115-116 and 183-185.

Responses to Reviewer #3

<<< Reviewer #3:

Remarks to the Author:

In this manuscript the authors present the crystal structure of full-length TRIM72, a RING E3 ubiquitin ligase. TRIM72 is an important protein in protein homeostasis but also function in signal transduction, membrane trafficking and DNA repair. TRIM72 is a multidomain protein.

In this study the authors characterised full-length wild type mouse TRIM72 in solution with SAXS. Determined the crystal structures of several constructs of mouse TRIM72, either truncated or mutated at specific positions as well as a very low resolution ($\sim 7\text{\AA}$) full length. In addition, they characterised membrane binding with other set of mouse TRIM72 constructs as well as constructs of human TRIM72. The membrane binding was assayed with lipid blots, negative stain EM and cryoEM/ET. Based on the data obtained from all these constructs and techniques the authors propose a model for TRIM72 function.

It is a remarkable achievement to get the crystal structure of full-length TRIM72. However, the amount of protein engineering needed and the low resolution obtained raise concerns about the interpretation of the structures. In addition, a major part of the interpretation is based on the cryoET data that is underwhelming. Taken together I find the proposed model highly speculative.

>>>

We believe the mutations for structure determination resulted in pretty minimal structural perturbation because the engineered residues were located at the surface exposed to solvents. The mutants showed almost identical UV spectra from circular dichroism spectroscopy, and their T_m values were also not much different from those of wild-type TRIM72 (both approximately $48.3\text{ }^\circ\text{C}$). Although we have relatively low-resolution structures of an engineered full-length protein (4.6 \AA) and WT (7.1 \AA), most of the

structural details were described with the highest-resolution (2.75 Å) structure of Δ RING TRIM72.

We endeavored to achieve a more convincing model of the oligomeric state on the membrane via the cryoET experiment suggested by the reviewer. Two cryoEM experts participated in the cryoET data collection and subtomogram averaging; thus, they are included as coauthors in the revised manuscript. We truly appreciate your suggestion because the proposed model (Fig. 6 in the revised manuscript) is now more solid and sound based on the subtomogram averaging data, although it required much time and effort. Please see the other responses for more details.

<<< Specific concerns that the authors should address before it can be considered for publication:

1) Regarding the crystal structures:

a. the resolution of some of the structures is really low $>5\text{\AA}$ and the $R_{\text{work}}/R_{\text{free}}$ are bad questioning the quality of these structures >>>

We have reprocessed the low-resolution data and refined the models carefully. Now, the statistics are slightly improved, as shown in the revised Supplementary Table 2. More importantly, we did not use these low-resolution structures for detailed structural analysis or biochemical studies; we only used them to show the limited directional movement of the CCD.

<<< b. based on superposition of the different crystal structures the authors suggest 'conformational movement' but since each structure is of a different construct (truncated or mutated) how can the authors rule out that they were not induced by crystal packing or because of the mutations that they introduced? >>>

Yes, we determined the structures of TRIM72 using various constructs, and the perpendicular movement that we argued occurred at the end of the CCD. The mutations that we generated were very minor mutations, such as C55S, C144S, and C242S, which involved only a replacement of sulfur with an oxygen atom. The other mutation, K279H/A283H, was originally designed by structure-based engineering based on the crystal structure of human TRIM72 H₃-PRYSPRY. Compared to the solved structure of H₃-PRYSPRY previously reported (Park *et al.*, Proteins, 2010), the K279H/A283H mutation did not affect the overall structure at all. Furthermore, we compared five other CCD structures of the TRIM family protein previously determined (Extended Data Fig. 2b), and the perpendicular movement of each end of the long CCD is a common feature of the TRIM family. Therefore, we do not think this was due to the crystal packing or the introduced mutations. Rather, we believe it was the nature of a straight hendecad-repeat helical dimer, which is a unique structural feature of the TRIM family.

<<< 2) Regarding the structures on membranes:

a. how did the identity of the proteins on vesicles validated? >>>

The vesicles were copurified with Strep-tagged TRIM72. The first step of purification was affinity purification with the StrepTrap column. Even in the next purification step of TRIM72 using gel filtration chromatography, the vesicles comigrated with TRIM72 protein. TRIM72 was detected by SDS-PAGE, Coomassie staining and western blotting with an anti-TRIM72 antibody. To further check the identity of the copurified vesicles, we performed proteomic analysis and analyzed the ontology data provided in the revised Supplementary Data S1. We found that they could be classified as microsomes, but their definition was quite vague; thus, we were not 100% sure. Therefore, the term microsome in the original manuscript has been replaced with microvesicle in the revised manuscript.

<<< *b. The negative stain data clearly show that hTRIM72 can be organised on membranes, however, this is not obvious from the cryoEM/ET data presented. How does the top views look in cryoET? >>>*

The following tomogram slice shows the top view of TRIM72 in the reconstructed proteoliposomes (see below). The top views are also visible in Supplementary Movies S3 & S4, which were additionally created during the revision process. We performed subtomogram averaging and obtained a more convincing cryoET map (Fig. 3b, c and Extended Data Figs. 8 & 9 in the revised manuscript), and the top view is well organized.

<<< *c. If indeed TRIM72 have a specific high-order arrangement on membranes, then that arrangement should be experimentally determined by sub-tomogram averaging. >>>*

According to your valuable suggestion, we experimentally determined the structure of the higher-order TRIM72 assembly on the membrane by subtomogram averaging. The data

and figures (Fig. 3b, c and Extended Data Figs. 8 & 9) are included in the revised manuscript.

<<< 3) In Fig 3c there is one vesicle with proteins binding to the inside (negative curvature).

a. How did the protein got to the inside of the liposomes if the protein was added to pre-formed liposomes?

b. was assembly of TRIM inside also observed for hTRIM72? >>>

We observed the inner liposomes, which might be produced in the following way (see below). We are not 100% sure about this process, but it might be a plausible scenario.

<< Blue represents oligomeric TRIM72 molecules on the membrane. >>

Unfortunately, it was not easy to examine TRIM72-bound microvesicles in cryoET experiments because concentrating the vesicles was very difficult. However, we purified TRIM72-bound microvesicles by using the interaction between the Strep-tagged TRIM72 protein and StrepTactin-conjugated resins through affinity chromatography in the initial stage. Therefore, we are confident that TRIM72 proteins exist on the outer surfaces of the microvesicles.

<<< c. to assess the significance of this observation the authors need to quantify the number of cases where they observed binding from the inside and compare it to outside. A meaningful size data set should be used for that. >>>

We mostly observed TRIM72 molecules inside liposomes and few TRIM72 molecules outside in the cryoET images. Our interpretation is that the TRIM72 molecules located on the inner liposomal membrane are a kind of trap, while the molecules outside must be dynamic. Therefore, we observed more oligomeric species inside the liposomes, and it was possible to perform subtomogram averaging only for concave membrane cases. We also tried to perform the same experiment for the outside membrane surface, but unfortunately, there were not enough vesicles for subtomogram averaging. It must be noted that the membrane is nearly flat, considering the dimension of the protein. The important finding is the higher-order TRIM72 assembly on the membrane.

<<< 4) In the model presented how the membrane curvature was accounted for? currently the model looks very flat. >>>

To characterize the higher-order TRIM72 assembly, we used the simplified flat membrane based on the subtomogram average map with low curvature. We showed that TRIM72 can self-oligomerize on liposomes with relatively broad ranges of curvature from high (radius of curvature $1/50 \text{ nm}^{-1}$) to low (over $1/500 \text{ nm}^{-1}$). Although it is one of the interesting features of TRIM72, we did not address the membrane curvature in the final model because it was not a significant issue in this study, as suggested by Reviewer #2. Furthermore, TRIM72 molecules are much smaller than the membrane; thus, in the narrow range of the TRIM72 assembly, the membrane seems almost flat.

<<< Minor points:

- Since structures presented in this manuscript are coming from different techniques, the figure legends should be very clear from which technique each structure is coming from. For example in figure 1 it should be clearly mentioned that these are crystal structures. The legend to Fig 1 should also include the colour coding for each construct and what the author mean by 'overall structure', which construct is this?? The membrane should be removed from this figure as it is not part of the structure determined. >>>

As suggested, we have rewritten the Figure Legends indicating the techniques and the constructs used. We have also removed the membrane model from Fig. 1b.

<<< - correct typo in line 222 should be 100 nm not micrometer. >>>

Thank you. We have corrected the typo

Decision Letter, first revision:

Message:

Nature Structural & Molecular Biology NSMB-A45296A

13th Jan 2023

Dear Dr. Song,

Thank you for submitting your manuscript, "Structure and activation of the RING E3 ubiquitin ligase TRIM72 on the membrane". The comments of 3 expert referees are below. You will see that reviewer #1 has serious concerns which cast doubt on the strength of the biological conclusions of the manuscript. Specifically, while we appreciate the effort put into the revision, providing further structural data, as pointed out by reviewer #1, these fail to unequivocally prove the ubiquitination model presented. Based on these comments, we cannot offer to publish the study in Nature Structural & Molecular Biology. We hope the referees' comments will be useful to you in revising the manuscript for submission elsewhere.

While we are unable to further pursue publication, I would suggest transfer to our sister journal in the Nature Portfolio – Nature Communications. During submission you chose to opt out of transfer consultations. In circumstances like these, I would be happy to consult my colleagues at Nature Communications on your behalf, as they might offer to consider

the manuscript further. Please do let me know if you would like me to do so.

I am sorry we could not be more positive on this occasion.

Sincerely,

Katarzyna Ciazynska
(she/her)
Associate Editor
Nature Structural & Molecular Biology
<https://orcid.org/0000-0002-9899-2428>

Reviewer #1 (Remarks to the Author):

In response to concerns raised by the reviewers, the authors have made significant changes to the manuscript, and in particular have carried out additional cryo-electron tomography and subtomogram averaging with reconstructed TRIM72-proteoliposomes. They have also significantly restructured the manuscript with the aim of communicating their findings more clearly.

Unfortunately, the manuscript is still very dense and difficult to read with relevant data distributed across the main manuscript, Extended data and Supplementary data. It would have been more helpful if the authors focussed on those experiments that are important to support their conclusions and final model of TRIM72 function, and removed those that do not contribute significantly to the overall story.

Specific points:

I do not understand the response given to my question about the SPR-based analysis of interaction studies between PS-SUVs and FL TRIM72 versus the GST-H3-PRYSRY construct. Based on the description of the SPR experiments in Methods it was the liposomes that were immobilised so liposome density on the sensor chip is expected to be similar between all experiments. In this case, I do not understand why different concentrations of protein (over 10-fold) are required to reach saturation at very similar affinities. I'm not referring to the value of the R.U signal.

Similarly, I am still unconvinced by the model provided for the ubiquitination activity of TRIM72, which the authors indicate is affected by the flexibility of the RING domain at the N-terminus of the protein, RING dimerization and the identity of the linchpin residue (a glutamine residue). The authors state in their rebuttal that fusion of the two RING domains did not enhance activity but I cannot find these data in the manuscript. This is clearly different to what has been shown for other TRIM proteins (such as TRIM5a which also forms higher order structures and may work in a similar manner) and the authors should comment on what they mean by "The RING-RING dimerization on a particular orientation coupled with reduced flexibility of RING must be critical for the E3 ligation activity, as described above". Similarly, I still do not understand the activation or seed model (Ext Data Fig 10c), it would be more convincing if this was repeated either with heat-treated soluble TRIM72 or TRIM72 delta RING as a control.

Furthermore, the authors have not fully addressed my concerns about purification of hTRIM72 overexpressed in HEK293T cells and bound to microvesicles. As shown on the SDS gel shown in Fig. 4A this preparation has a number of impurities and hence it cannot be excluded that a co-migrating E3 ligase may contribute to the observed catalytic activity. Importantly, I do not understand the following statement in the rebuttal: "Furthermore, TRIM72 mutants lacking the RING domain or defective in zinc coordination did not comigrate with microvesicles". Given that lipid binding is mediated by the PRYSPRY domain, and TRIM72 without the RING domain is clearly fully structured (the authors were able to solve the 2.75Å crystal structure of this construct) – why would this construct not bind to lipid vesicles? Even their own model shown in Fig 6 assumes that the RING is not involved in the interaction with the membrane.

What are the differences between the ubiquitination assays shown in Figs 4B and c? Why is the activity of TRIM72 bound to liposomes so different, given that it appears that the TRIM72 concentration used is very similar?

Finally, another structure of TRIM72 has recently been published (PMID: 36053137). The authors should mention this structure and comment on any differences between the different structures observed.

Reviewer #2 (Remarks to the Author):

the authors have appropriately addressed my comments.

Reviewer #3 (Remarks to the Author):

The revised manuscript is much improved and addressing all my initial concerns. Though I have a couple of points for the revised version that still needs addressing:

1. The addition of the subtomogram averaging was absolutely crucial for this manuscript and thus substantially improved the manuscript. To allow the readers to judge the map, a 3D rendering of the map needs to be first presented without the fit of the crystal structure and modelling of the whole lattice. A top view and a side view should be shown.

2. It is not clear from the text if the subtomogram averaging map is from a delta-Ring construct or not. And if it is from a full-length why not the full-length structure fitted in? Even if there is no density for the ring domain it will be interesting to see on the whole lattice where it should be (outside of the density). If the ring domain is causing clashes when modelling the lattice, it means its position as determined by crystallography is not correct? The authors have to discuss this point more clearly and openly.

3. In the meantime a cryoEM structure of dimeric TRIM72 became available in the PDB (7Y4S). The authors should include a comparison between their crystal structure and the new cryoEM structure. Also the authors should remove all claims for providing the 'first' structure of TRIM72.

4. The SAXS envelope is redundant here and distracting, I suggest to remove

5. I am still not convinced that the superposition of the different constructs/structures are a proof of genuine flexibility. A stronger evidence can come from MD simulations.
6. Many times Fig 3c is referenced but should have been Fig 3b (subtomogram averaging map).
7. Also in Fig 3 the red and blue in 'd' and 'e' should be the same as in 'b' (the map with a fit).
8. The manuscript will benefit from another round of English editing

** As a service to authors, Springer Nature Limited provides authors with the ability to transfer a manuscript that one journal cannot offer to publish to another journal, without the author having to upload the manuscript data again. To transfer your manuscript to another NPG journal using this service, please click on [Redacted]

** For Springer Nature Limited general information and news for authors, see <http://npg.nature.com/authors>.

Author Rebuttal, first revision:

<< *Reviewer #1 (Remarks to the Author):*

In response to concerns raised by the reviewers, the authors have made significant changes to the manuscript, and in particular have carried out additional cryo-electron tomography and subtomogram averaging with reconstructed TRIM72-proteoliposomes. They have also significantly restructured the manuscript with the aim of communicating their findings more clearly.

Unfortunately, the manuscript is still very dense and difficult to read with relevant data distributed across the main manuscript, Extended data and Supplementary data. It would have been more helpful if the authors focussed on those experiments that are important to support their conclusions and final model of TRIM72 function, and removed those that do not contribute significantly to the overall story. >>

-

We agree that the manuscript could be streamlined, and we are willing to rewrite the manuscript to present a more structured argument. Therefore, we propose to remove the following sections and experiments.

- Description of coiled-coil domain and its flexibility in detail.

- Extended Data Figure 2a,b

- Supplementary Figure 2

- SAXS envelope models of TRIM72 WT and Δ RING in solution

- Extended Data Figure 1g,h

-Calcium resistance of TRIM72 oligomer bound to the membrane.

- Figure 5

- Extended Data Figure 3

We will further refine the manuscript as commented on by reviewer #3.

<< I do not understand the response given to my question about the SPR-based analysis of interaction studies between PS-SUVs and FL TRIM72 versus the GST-H3-PRYSPRY construct. Based on the description of the SPR experiments in Methods it was the liposomes that were immobilised so liposome density on the sensor chip is expected to be similar between all experiments. In this case, I do not understand why different concentrations of protein (over 10-fold) are required to reach saturation at very similar affinities. I'm not referring to the value of the R.U signal.>>

We are happy to discuss whether to perform the same experiment again using GST-H₃PRYSPRY on the liposome-coated SPR chips with similar concentrations of TRIM72 FL and revise the Extended Data Fig. 4g. Alternatively, we can remove the data altogether, as we consider these findings not critical for understanding the activation mechanism of TRIM72.

<< Similarly, I am still unconvinced by the model provided for the ubiquitination activity of TRIM72, which the authors indicate is affected by the flexibility of the RING domain at the N-terminus of the protein, RING dimerization and the identity of the linchpin residue (a glutamine residue). The authors state in their rebuttal that fusion of the two RING domains did not enhance activity but I cannot find these data in the manuscript. >>

As reviewer #1 commented, the ubiquitination activity is not enhanced by tandem TRIM72 RING domains in contrast to the TRIM5a RING domain (Fletcher et al., 2018). The ubiquitination assay using two RING domains was already included in Supplementary Figure 7a-c of the revised manuscript (please see below, Appeal Figure 1a-c). The ubiquitination assays using tandem TRIM72 RING constructs clearly show that TRIM72 RING is inactive in solution, although two RING domains are concatenated.

Appeal Figure 1. *In vitro* ubiquitination assay with tandem RING domains of TRIM72 a, SDS-PAGE analysis of TRIM72 constructs containing the RING domain. b-c, *In vitro* E2 screening with various TRIM72 constructs: WT and RING+WT (b), RING and

2×RING (c). [The same as Supplementary Fig. 7a-c in the revised manuscript]

<< This is clearly different to what has been shown for other TRIM proteins (such as TRIM5a which also forms higher order structures and may work in a similar manner).>>

The major difference between TRIM72 and TRIM5a in the RING domain is the ionic strength of the linchpin motif. Unlike the TRIM5a RING domain, which is still active in monomeric and dimeric constructs (Fletcher et al., 2018), TRIM72 has a glutamine linchpin motif rather than a canonical arginine. Therefore, we can repeat the experiments with TRIM72 Q57R mutants and demonstrate the enhancement of ubiquitination activity of the tandem two RING domains like TRIM5a.

<< and the authors should comment on what they mean by “The RING-RING dimerization on a particular orientation coupled with reduced flexibility of RING must be critical for the E3 ligation activity, as described above”.>>

We found that the higher-order TRIM72 assembly is critical for activating ubiquitin transfer on phospholipid membranes (Fig. 4b). To dissect the detailed activation mode, we introduced an L74R mutation that eliminates the RING-RING dimer interaction. The mutation abrogated the ubiquitination activity supporting the dimerization model of the RING domain for Ub transfer from the E2~Ub intermediates (Fig. 6). We hence concluded that the TRIM72 RING is activated by dimerization following the higher-order assembly on the phospholipid membrane. The B-box:B-box interactions (critical for the formation of TRIM72 assembly) place two RING domains into close proximity and thereby increase local

concentration and the probability of dimerization. Furthermore, the movement of the dynamic RING domain due to the flexible linker region between the RING and B-box domains (Supplementary Fig. 8a) will likely be restricted, given the space limitations imposed by the oligomer.

<< Similarly, I still do not understand the activation or seed model (Ext Data Fig 10c), it would be more convincing if this was repeated either with heat-treated soluble TRIM72 or TRIM72 delta RING as a control. >>

As reviewer #1 commented, we are happy to perform this experiment using heat-treated soluble TRIM72 or TRIM72 delta-RING as a control.

<< Furthermore, the authors have not fully addressed my concerns about purification of hTRIM72 overexpressed in HEK293T cells and bound to microvesicles. As shown on the SDS gel shown in Fig. 4A this preparation has a number of impurities and hence it cannot be excluded that a co-migrating E3 ligase may contribute to the observed catalytic activity.>>

We will perform the same experiment using hTRIM72 L74R mutants and compare the ubiquitination activity with WT to clarify the ubiquitination activity of TRIM72microvesicle complexes. Alternatively, we have preliminary data demonstrating that the ubiquitination of hTRIM72 is inhibited by a TRIM72 RING domain-specific antibody (Appeal Figure 2). It clearly showed that the ubiquitination is induced by TRIM72, but not an endogenous E3 ligase. Therefore, this figure will be included in a revised figure. More importantly, the Q57R mutant showed an enhanced ubiquitination activity (Fig. 4c in the revised

manuscript). These are the pieces of evidence that TRIM72 participates in the enzymatic reaction, not comigrating E3 ligases.

Appeal Figure 2. Ubiquitination is inhibited by the hTRIM72 RING domain-specific ScFv antibody in the microvesicle-containing fractions. Ubiquitination activity of hTRIM72 WT in the absence (a) or presence (b) of a RING domain-specific ScFv antibody.

<<Importantly, I do not understand the following statement in the rebuttal: “Furthermore, TRIM72 mutants lacking the RING domain or defective in zinc coordination did not comigrate with microvesicles”. Given that lipid binding is mediated by the PRYSPRY domain, and TRIM72 without the RING domain is clearly fully structured (the authors were able to solve the 2.75Å crystal structure of this construct) – why would this construct not bind to lipid vesicles? Even their own model shown in Fig 6 assumes that the RING is not involved in the interaction with the membrane.>>

We will perform the same experiment using the optimized construct of TRIM72 deltaRING and show that the major ubiquitination activity came from TRIM72, not endogenous E3

ligases. As reviewer #1 mentioned, we already expressed the TRIM72 delta-RING based on the crystallized constructs (residues 79-477) in mammalian cells and performed the ubiquitination assay *in vitro*. However, unfortunately, the crystallized delta-RING construct interacted with endogenous chaperon Hsp70 proteins rather than microvesicles unlikely to TRIM72 WT (Appeal Figure 2). It seems to be the Hsp70 proteins recruit to the N-terminal flexible region of TRIM72 delta-RING proteins and interfere the oligomerization of TRIM72, and even more the interaction between TRIM72 and microvesicle. Therefore, we will perform the ubiquitination assay using the intact delta-RING construct to avoid interference from Hsp70.

Appeal Figure 2. Purification of TRIM72 constructs in the mammalian expressions. The SEC profiles of TRIM WT (a) and RING-deletion (b). Chromatograms (top) and SDS-PAGE of each fraction (bottom) were shown. Notes that the 70 kDa proteins are Hsp70 identified by Mass spectrometry in b.

<< What is the differences between the ubiquitination assays shown in Figs 4B and c? Why is the activity of TRIM72 bound to liposomes so different, given that it appears that the TRIM72 concentration used is very similar?>>

The difference is only in exposure time between Fig. 4b and 4c. We used the same amount of sample for the assay in both Fig. 4b and 4c. The ubiquitination activity of TRIM72 Q57R is significantly higher than that of TRIM72 WT (Fig. 4c), so we should reduce the exposure time to avoid the saturation of signals. We attached these western blots with the identical exposure time (Appeal Figure 4).

Appeal Figure 4. Western blot data with the same exposure time. a, The same blot as in Fig. 4b with a shorter exposure time. b, The same Fig. 4c from the revised manuscript for comparison.

<< Finally, another structure of TRIM72 has recently been published (PMID: 36053137). The authors should mention this structure and comment on any differences between the different structures observed. >>

As reviewer #1 (and also reviewer #3) pointed out, the CryoEM structure of TRIM72 was recently published (Niu et al., 2022). However, the published structure did not encompass the full-length TRIM72 structure and only identified the CCD and PRYYPRY domains at 3.5 Å resolution. In addition, due to the flexibility of CCD, side chains or amino acids were missing in the residues 1-148 and 211-241, precluding a detailed comparison with our structures. The overall shape could be compared and the core folding (residue 170-190 and 266-470) was very similar between the crystal structure and the CryoEM model (Appeal Figure 5). We will integrate this comparison into the discussion section.

Appeal Figure 5. Structural comparison between the crystal structure and the CryoEM model of TRIM72. Our crystal structure of TRIM72 FL (green) including RING and Bbox domain (cyan), and CryoEM model of TRIM72 CCD and PRYSRY (magenta). The side view (left) and the front view (right) of TRIM72 FL were shown as ribbon diagrams.

As suggested by reviewers #2 and #3 in the first round of revision, we propose an activation model of TRIM72 on the lipid membranes using structural and biochemical data. Our finding is still important enough to publish the paper because it is the first fulllength structure of the TRIM family, which suggests the functional assembly is critical for the activation of the TRIM E3 ubiquitin ligases. Therefore, we would be grateful if the editorial decision could be reconsidered. We believe that a second revision would allow us to improve

the manuscript further and to highlight our important findings in the best possible way.
Thank you for reviewing my appeal and for your time.

Sincerely,

Hyun Kyu Song

Decision Letter, second revision:

Message: 22nd Feb 2023

Dear Dr. Song,

Thank you again for asking us to reconsider our decision on the manuscript "Structure and activation of the RING E3 ubiquitin ligase TRIM72 on the membrane". I am sorry for the delay in reaching a decision on your work. As explained in my previous email, appeals require careful thought and unfortunately have to take second place to the routine business of running the journal.

After careful consideration and discussion with my colleagues, I am happy to tell you that we have decided that we would consider a revised version of your manuscript that addresses the remaining technical concerns raised by the reviewer #1. We would like to see your response to the comments of the referee in the form of a revised manuscript and a rebuttal letter. We would request that you please do your best to fully address all of the comments of the reviewer in the revision. Should you be able to adequately respond to the reviewer's concerns in full, we would be happy to look at the revised manuscript.

Specifically, we would expect the revision to address why varying protein concentrations were used in SPR experiments and/or the experiments to be repeated with the same concentration of TRIP72 FL. Moreover, we agree that comparing ubiquitination activity of TRIM72 Q57R mutant with TRIM72 and TRIM5a, as well as additional controls using heat treated or delta-RING TRIM72, would strengthen the manuscript. Finally, we would like to stress that the revision must unequivocally show that TRIM72 is responsible for ubiquitination, reasonably excluding the possibility of endogenous E3 ligases contributing to enzymatic activity.

Please be sure to address/respond to all concerns of the referee in a point-by-point response and highlight all changes in the revised manuscript text file. If you have comments that are intended for editors only, please include those in a separate cover

letter.

We expect to see your revised manuscript within 10 weeks. If you cannot send it within this time, please contact us to discuss an extension; we would still consider your revision, provided that no similar work has been accepted for publication at NSMB or published elsewhere.

Reporting Summary:

When submitting the revised version of your manuscript, please pay close attention to our [href="https://www.nature.com/nature-portfolio/editorial-policies/image-integrity">Digital Image Integrity Guidelines. and to the following points below:](https://www.nature.com/nature-portfolio/editorial-policies/image-integrity)

Please note that all key data shown in the main figures as cropped gels or blots should be presented in uncropped form, with molecular weight markers. These data can be aggregated into a single supplementary figure item. While these data can be displayed in a relatively informal style, they must refer back to the relevant figures. These data should be submitted with the final revision, as source data, prior to acceptance, but you may want to start putting it together at this point.

SOURCE DATA: we urge authors to provide, in tabular form, the data underlying the graphical representations used in figures. This is to further increase transparency in data reporting, as detailed in this editorial

(<http://www.nature.com/nsmb/journal/v22/n10/full/nsmb.3110.html>). Spreadsheets can be submitted in excel format. Only one (1) file per figure is permitted; thus, for multi-paneled figures, the source data for each panel should be clearly labeled in the Excel file; alternately the data can be provided as multiple, clearly labeled sheets in an Excel file. When submitting files, the title field should indicate which figure the source data pertains to. We encourage our authors to provide source data at the revision stage, so that they are part of the peer-review process.

Data availability: this journal strongly supports public availability of data. All data used in accepted papers should be available via a public data repository, or alternatively, as Supplementary Information. If data can only be shared on request, please explain why in your Data Availability Statement, and also in the correspondence with your editor. Please note that for some data types, deposition in a public repository is mandatory - more information on our data deposition policies and available repositories can be found below: <https://www.nature.com/nature-research/editorial-policies/reporting-standards#availability-of-data>

[Redacted]

Note: This URL links to your confidential home page and associated

information about manuscripts you may have submitted, or that you are reviewing for us. If you wish to forward this email to co-authors, please delete the link to your homepage.

Please do not hesitate to contact me if you have any questions or would like to discuss the required revisions further. Thank you for the opportunity to review your work.

Sincerely,
Kat

Katarzyna Ciazynska
(she/her)
Associate Editor
Nature Structural & Molecular Biology
<https://orcid.org/0000-0002-9899-2428>

Author Rebuttal, second revision:

Point-by-point responses to the comments of the reviewers

Responses to Reviewer #1

Reviewer #1 (Remarks to the Author):

<< In response to concerns raised by the reviewers, the authors have made significant changes to the manuscript, and in particular have carried out additional cryo-electron tomography and subtomogram averaging with reconstructed TRIM72-proteoliposomes. They have also significantly restructured the manuscript with the aim of communicating their findings more clearly.

Unfortunately, the manuscript is still very dense and difficult to read with relevant data distributed across the main manuscript, Extended data and Supplementary data. It would have been more helpful if the authors focussed on those experiments that are important to support

their conclusions and final model of TRIM72 function, and removed those that do not contribute significantly to the overall story. >>

Following the suggestions of reviewer #1 and in accordance with reviewer #3, we removed the following figures to streamline the manuscript.

- 1) SAXS envelope models of TRIM72 - Extended Data Figure 1g,h (previous ver.)**
- 2) Perpendicular flexibility of the coiled-coil domain - Extended Data Figure 2b (previous ver.)**
- 3) Heat inactivation experiments – Extended Data Figure 10c (previous ver.)**
- 4) Redundant ubiquitination assays with reconstituted liposomes - Extended Data Figure 10d-f/Supplementary Figure 7a (previous ver.)**

We updated the following figure to strengthen our argument in the revised manuscript.

- 1) Time course of ubiquitination activity assay using microvesicle-bound TRIM72 - Figure 4d (new version)**
- 2) Ubiquitination assay for comparison between TRIM72 and TRIM5 α RING constructs - Figure 4e,f (new version)**

The extended and supplementary figures were further rearranged and refined following the comments by reviewers, which we think will help convey the key messages of the paper. It particularly concerns the activation mechanism of E3 ubiquitin ligase TRIM72 via higher-order assembly on the phospholipid membrane (Fig. 6 in the revised manuscript).

<< *Specific points:*

I do not understand the response given to my question about the SPR-based analysis of interaction studies between PS-SUVs and FL TRIM72 versus the GST-H3-PRYSPRY construct. Based on the description of the SPR experiments in Methods it was the liposomes that were immobilised so liposome density on the sensor chip is expected to be similar between all experiments. In this case, I do not understand why different concentrations of protein (over 10-fold) are required to reach saturation at very similar affinities. I'm not referring to the value of the R.U signal. >>

We appreciate this feedback, which has prompted us to revise our liposome-based SPR analysis. Previous experiments were performed with different batches at different dates. We hence prepared all materials freshly and reperformed SPR experiments to obtain results that are more consistent between experiments. We obtained binding constants (K_D) for each mutant protein, including the GST-H3-PRYSPRY construct, using similar protein concentrations and also using a more sensitive SPR instrument (BIAcore T200). As noted, we observed that the GST-H3-PRYSPRY construct bound to the PS-liposomes less than TRIM72 WT, requiring higher protein amounts to reach saturation. The binding affinity of GST-H3-PRYSPRY was ~ 7.7 times lower than that of TRIM72 WT (Revision Figure 1). The new SPR results have been included in Extended Data Fig. 4 in the revised manuscript. We believe that our updated liposome-based SPR data can demonstrate the validity of our membrane binding model of TRIM72, which has also been supported by the cryo-ET structure of higher-order TRIM72 assembly.

Revision Figure 1. SPR analysis with similar concentrations of analytes [the same as the Extended Data Fig. 4 in the revised manuscript].

<< Similarly, I am still unconvinced by the model provided for the ubiquitination activity of TRIM72, which the authors indicate is affected by the flexibility of the RING domain at the N-terminus of the protein, RING dimerization and the identity of the linchpin residue (a glutamine

residue). The authors state in their rebuttal that fusion of the two RING domains did not enhance activity but I cannot find these data in the manuscript. This is clearly different to what has been shown for other TRIM proteins (such as TRIM5 α which also forms higher order structures and may work in a similar manner) and the authors should comment on what they mean by “The RING-RING dimerization on a particular orientation coupled with reduced flexibility of RING must be critical for the E3 ligation activity, as described above”.>>

As reviewer #1 pointed out, we did not observe any enhanced ubiquitination activity of TRIM72 RING by tandem linearization, unlike the TRIM5 α RING domain (Supplementary Fig. 7c in the previous version; Revision Figure 2).

Revision Figure 2. *In vitro* ubiquitination assay with TRIM72 RING constructs. *In vitro* E2 screening with various TRIM72 constructs: RING and 2×RING [the same as Supplementary Fig. 7b (bottom) in the revised manuscript].

We already showed that the suboptimal linchpin motif of TRIM72 RING (Gln57) attenuates the activity of the RING domain. The major difference between TRIM72 and TRIM5 α in the RING domain is the charge of the linchpin, with Gln57 in TRIM72

corresponding to Arg60 in TRIM5 α (Revision Figure 3).

```

TRIM72_RING      1  MSA-APGLLRQELSCPLCLQLFDAPVTAEGGSFCRACLIRVAGEPAA--DGTVACPCCC*APTRPQALSTNLQLSRLVEGLAQ      80
TRIM5a_RING      1  MASGILLNVKKEEVTCPICLELLTEPLSLHCGHSFCQACITANHKKSMLYKEGERSCPVCRISYQPENIQPNRHVANIVEK---      80
Consensus_aa:    Mtt.h...l+pElcCPLCLpLhs.PlohCGSFCpAClh.sh.cshh..-Gp.tCPhCshpPpslpsN.pltpIVE....
Consensus_ss:    hhhhhhhhhhh  hhhhhh  ee      hhhhhhhhhhhhh  hhhhhhhhhhhhhhhhh

```

Revision Figure 3. Sequence alignment of RING domains between *mouse* TRIM72 and *rhesus* TRIM5 α . Linchpin residues are highlighted with a red star.

To improve our RING dimerization activation model, we generated multiple constructs, including TRIM72 RING_Q57R, TRIM72 2×RING_Q57R, TRIM5 α RING, and 2×RING. In the ubiquitination activity assay using TRIM72 RING and 2×RING constructs, we found that the ubiquitination activity of the TRIM72 2×RING_Q57R constructs was enhanced (Revision Figure 4, left). This finding indicates that RING dimerization is critical for ubiquitin transfer in TRIM72 activation.

Next, we compared the ubiquitination activity of TRIM72_Q57R and TRIM5 α based on the RING and 2×RING constructs (Revision Figure 4, right). Although the ubiquitination activity of TRIM72_Q57R 2×RING was weaker than that of TRIM5 α RING, both activities were enhanced by RING duplication, as reported previously (Yudina et al., 2015, Fletcher et al., 2018, Herkules et al., 2022). Additionally, we demonstrated that the defect in RING dimerization results in decreased ubiquitination activity using TRIM72 L74R-proteoliposomes (see Fig. 4c in the revised manuscript). In conclusion, our RING dimerization-mediated activation model of TRIM72 is strongly supported by these ubiquitination assay data. All of the experiments are included in the revised manuscript (Fig. 4d-f). We appreciate the valuable comments provided by the reviewer that improved our TRIM72 activation model.

Revision Figure 4. Ubiquitination activity of TRIM72 and TRIM5 α using RING and duplicated RING (2 \times RING) constructs [the same as main Fig. 4e,f in the revised manuscript].

<< Similarly, I still do not understand the activation or seed model (Ext Data Fig 10c), it would be more convincing if this was repeated either with heat-treated soluble TRIM72 or TRIM72 delta RING as a control. >>

As per the reviewers' feedback, we reperfomed the heat-inactivation experiments with careful control. There is an endogenous E3 ligase activity in the microvesicle fraction (see our response on a comigrating E3 ligase) and thus, the experimental results were not reproducible (not consistent). Therefore, we have removed our hypothesis about the seeding model along with the corresponding figure. However, we believe that our TRIM72 activation model is still presented in a clear and understandable manner for readers, without these data. In addition, this has helped to streamline and simplify the manuscript.

<< Furthermore, the authors have not fully addressed my concerns about purification of hTRIM72 overexpressed in HEK293T cells and bound to microvesicles. As shown on the SDS gel shown in Fig. 4A this preparation has a number of impurities and hence it cannot be excluded that a co-migrating E3 ligase may contribute to the observed catalytic activity. >>

We fully agree with the reviewer's point about the difficulties in observing TRIM72 ubiquitination activity (due to interference from comigrating Ub ligases). As expected from the ubiquitination activity of TRIM72-proteoliposomes, the activity of TRIM72-microvesicles should be low due to the suboptimal linchpin. However, the ubiquitination activity of TRIM72-microvesicles appeared to be higher than that of TRIM72-proteoliposomes, indicating that endogenous ubiquitination activities still remain in the TRIM72-microvesicles, most likely due to comigrating Ub ligases, as pointed out by reviewer #1.

To address the reviewer's concern, we performed a time-course ubiquitination activity assay using hyperactive TRIM72 Q57R and inactive TRIM72 Q57R/L74R with microvesicles and free-soluble fractions, respectively (Revision Figure 5). The ubiquitination activity of TRIM72 Q57R/L74R-microvesicles was clearly decreased compared to that of TRIM72 Q57R-microvesicles, indicating that TRIM72 activates ubiquitin transfer on the microvesicles. We have updated the corresponding experimental results in Fig. 4f in the revised manuscript. Indeed, it is a consistent result of the ubiquitination activity of TRIM72 Q57R and Q57R/L74L in the presence or absence of PS-liposomes (Revision Figure 6). We are convinced that this comparative result resolves your concern.

Revision Figure 5. Ubiquitination activity assay of TRIM72 Q57R and Q57R/L74R with microvesicles and free soluble state [the same as Fig. 4d in the revised manuscript].

Revision Figure 6. Ubiquitination activity assay of TRIM72 Q57R and Q57R/L74R in the presence or absence of PS-liposomes [the same as Fig. 4c in the revised manuscript].

<< *Importantly, I do not understand the following statement in the rebuttal: “Furthermore, TRIM72 mutants lacking the RING domain or defective in zinc coordination did not comigrate with microvesicles”. Given that lipid binding is mediated by the PRYSPRY domain, and TRIM72 without the RING domain is clearly fully structured (the authors were able to solve the 2.75Å crystal structure of this construct) – why would this construct not bind to lipid vesicles? Even their own model shown in Fig 6 assumes that the RING is not involved in the interaction with the membrane. >>*

1) We repeated the comigration experiments for TRIM72 microvesicles, including mutants lacking the RING domain. In the first round of revision, we used the delta-RING construct (TRIM72 Δ RING_59-477, a less deleted version of Δ RING), which interacted with the Hsp70 chaperone rather than microvesicles, unlike WT (Revision Fig. 6a,b). This could be due to the flexible linker 1 region (see main Fig. 1a in the revised manuscript) being trapped by the Hsp70 chaperone and interfering with the interaction with the microvesicle. To address this, we generated and tested the intact version of Δ RING_79-477, which is the same construct we used to solve the crystal structure (also SAXS analysis), and found that it interacted well with microvesicles, similar to TRIM72 WT (Revision Fig. 6a, c). We believe this fully addresses your concern about the comigration of TRIM72 and microvesicles. RING is hence not involved in the interaction with microvesicles, which is consistent with our model in Figure 6.

2)

3)

Revision Figure 6. Comigration of TRIM72 WT and Δ RING constructs with microvesicles. The SEC profiles of TRIM WT (a), Δ RING_59-477 (b), and Δ RING_79-477 (c). Chromatograms (top) and SDS-PAGE of each fraction (bottom) are shown. Note that the 70 kDa proteins are Hsp70 identified by mass spectrometry in (b).

<< What is the differences between the ubiquitination assays shown in Figs 4B and c? Why is the activity of TRIM72 bound to liposomes so different, given that it appears that the TRIM72 concentration used is very similar? >>

The difference is only in the exposure time between Fig. 4b and 4c. The ubiquitination activity of TRIM72 Q57R is significantly higher (Fig. 4c in the previous version: now Fig. 4b in the current revised version) than that of TRIM72 WT, so we reduced the exposure time for detection to prevent the saturation of signals in Fig. 4c. We have attached these western blots with identical exposure times (Revision Figure 7).

Revision Figure 7. Western blot data of Fig. 4b and 4c (former figure number) with the same exposure time. a, The same blot as in Fig. 4b (the previous version) with a shorter exposure time (the same exposure time as in Fig. 4c). b, Fig. 4c from the previously revised manuscript for comparison [now Fig. 4b in the current revised version].

<< Finally, another structure of TRIM72 has recently been published (PMID: 36053137). The authors should mention this structure and comment on any differences between the different structures observed. >>

As pointed out by reviewer #1 (and reviewer #3), the cryo-EM structure of mouse TRIM72 CCD-PRYSRY (Niu et al., 2022, Biochem J), and the crystal structure of human TRIM72 ΔRING were recently published (Ma et al., 2023, Nature Comms). When comparing them with our TRIM72 structures (Revision Figure 8), the overall shapes and core folding (residues 170-190 and 266-470) were similar. Each end of the coiled-coil domain is dynamic, which was already observed in comparison with our crystal structures (Supplementary Fig. 2f). We included this information briefly in the Discussion.

Revision Figure 8. Structural comparison of TRIM72 FL between the crystal structure of TRIM72 Δ RING and the cryo-EM model of TRIM72 CCD-RPYSRY. Our crystal structure of TRIM72 FL (green), including RING (red) and B-box domains (cyan), the cryo-EM model of TRIM72 CCD-PRYSRY (magenta) and the crystal structure of TRIM72 Δ RING (pink). The side view (left panel) and the front view (right panel) are shown as ribbon diagrams.

Responses to Reviewer #2

Reviewer #2 (Remarks to the Author):

<< the authors have appropriately addressed my comments. >>

We are happy to hear that you were satisfied with our revision. Your previous comments were truly valuable for improving the manuscript.

Responses to Reviewer #3

Reviewer #3 (Remarks to the Author):

The revised manuscript is much improved and addressing all my initial concerns. Though I have a couple of points for the revised version that still needs addressing:

<< 1. The addition of the subtomogram averaging was absolutely crucial for this manuscript and thus substantially improved the manuscript. To allow the readers to judge the map, a 3D rendering of the map needs to be first presented without the fit of the crystal structure and modelling of the whole lattice. A top view and a side view should be shown. >>

According to the reviewer's comment, we included the cryo-ET 3D-rendering map without the fitting of the crystal structure and modeling of the whole lattice (Revision Figure 9), as shown in Fig. 3a (top and side view) in the revised manuscript.

Revision Figure 9. Subtomogram average map of the TRIM72 proteoliposome (the same as Fig. 3a in the revised version).

<< 2. It is not clear from the text if the subtomogram averaging map is from a delta-Ring construct or not. And if it is from a full-length why not the full-length structure fitted in? Even if there is no density for the ring domain it will be interesting to see on the whole lattice where it should be (outside of the density). If the ring domain is causing clashes when modelling the

lattice, it means its position as determined by crystallography is not correct? The authors have to discuss this point more clearly and openly. >>

We used full-length TRIM72 proteoliposomes for cryo-ET and subtomogram averaging, so we updated Fig. 3b in the revised manuscript with the TRIM72 full-length model. We highlighted the position of the RING domain (Revision Figure 10), which does not clash with fits of the FL models fit into the lattice.

Revision Figure 10. TRIM72 including the RING domain in the subtomogram average map. The updated model (left panel) and the close-up view of the RING domain (right panel). The RING domain is colored cyan.

<< 3. In the meantime a cryoEM structure of dimeric TRIM72 became available in the PDB (7Y4S). The authors should include a comparison between their crystal structure and the new cryoEM structure. Also the authors should remove all claims for providing the 'first' structure of TRIM72. >>

Indeed, the cryo-EM structure [PMID: 36053137] and, very recently, crystal structure

[PMID: 36944613] were also reported. The cryo-EM model is incomplete due to the CCD flexibility (Revised Figure 8), and there are several missing parts (RING and B-box) and differences in the resolution and atomic accuracy of the models. Therefore, instead of a detailed structural comparison, we described their results regarding CCD flexibility in the Discussion section and referenced them appropriately. We also removed the word ‘first’ in the revised manuscript.

<< 4. The SAXS envelope is redundant here and distracting, I suggest to remove >>

Per the reviewer’s comments, we have omitted the SAXS envelope models of TRIM72.

<< 5. I am still not convinced that the superposition of the different constructs/structures are a proof of genuine flexibility. A stronger evidence can come from MD simulations. >>

Per the reviewer’s comments, we have removed the description of perpendicular CCD movements and corresponding figures. In addition to our SAXS and structural superposition results, two groups independently reported the flexibility of the peripheral region of TRIM72 by cryo-EM and crystallographic methods with partial MD simulations (although not an in-depth MD analysis). To date, all experimental evidence from us and others supports the flexibility of CCD.

<< 6. Many times Fig 3c is referenced but should have been Fig 3b (subtomogram averaging map). >>

We carefully read through the manuscript and corrected these references appropriately.

<< Also in Fig 3 the red and blue in 'd' and 'e' should be the same as in 'b' (the map with a fit).
>>

We have updated the color in the revised Fig. 3d,e, as suggested.

<< *The manuscript will benefit from another round of English editing* >>

We have obtained an English editing service again from a native English speak

Decision Letter, third revision:

Message: Our ref: NSMB-A45296C

5th Jun 2023

Dear Dr. Song,

Thank you for submitting your revised manuscript "Structure and activation of the RING E3 ubiquitin ligase TRIM72 on the membrane" (NSMB-A45296C). It has now been seen by the original referees and their comments are below. The reviewers find that the paper has improved in revision, and therefore we'll be happy in principle to publish it in Nature Structural & Molecular Biology, pending minor revisions to satisfy the referees' final requests and to comply with our editorial and formatting guidelines.

Sincerely,

Katarzyna Ciazynska
(she/her)
Associate Editor
Nature Structural & Molecular Biology
<https://orcid.org/0000-0002-9899-2428>

Reviewer #1 (Remarks to the Author):

To respond to the remaining concerns of reviewers the authors have carried out additional experiments to validate their data and strengthen the conclusions drawn. Furthermore, they have removed or rearranged some of the figures to streamline the manuscript and highlight their key messages. This has made the manuscript more easily accessible to the reader. My remaining concerns have now been addressed and I am happy to recommend publication of this work.

Final Decision Letter:

Message Dear Dr Song,

Please find below a copy of the decision letter for your manuscript "Structure and activation of the RING E3 ubiquitin ligase TRIM72 on the membrane" [NSMB-A45296D], which has just been accepted for publication in Nature Structural & Molecular Biology.

The exact publication date will be communicated to the corresponding author. Please note that until publication, the content of your paper remains under embargo (to determine when the paper can be discussed with the media, please consult our embargo policy at http://www.nature.com/authors/editorial_policies/embargo.html).

If you wish to order reprints of your article or have any questions about reprints please send an email to author-reprints@nature.com.

Please contact the corresponding author directly with any queries you may have related to the content and publication of your paper.

As we prepare the manuscript for publication, we would like to confirm that your address details are correct. Could you please click on the link below to verify your profile and correct it as needed? Your prompt attention to this will help us to avoid delays in publication of your manuscript.

Please verify your address details promptly and correct them as needed by clicking here and following the link to "Login to My Account/Modify My NRG Profile":

<https://mts-nsmb.nature.com/cgi-bin/main.plex?el=A7J4BRy7A2BAwC3J3A9ftdJ9okJu4dZvNZnfKC6eRrQZ>

You can now use a single sign-on for all your accounts, view the status of all your manuscript submissions and reviews, access usage statistics for your published articles and

download a record of your refereeing activity for the Nature journals.

Sincerely,
Katarzyna Ciazynska
(she/her)
Associate Editor
Nature Structural & Molecular Biology
<https://orcid.org/0000-0002-9899-2428>

Subject: Decision on Nature Structural & Molecular Biology submission NSMB-A45296D

16th Aug 2023

Dear Dr. Song,

We are now happy to accept your revised paper "Structure and activation of the RING E3 ubiquitin ligase TRIM72 on the membrane" for publication as an Article in Nature Structural & Molecular Biology.

Your paper will be published online soon after we receive proof corrections and will appear in print in the next available issue. You can find out your date of online publication by contacting the production team shortly after sending your proof corrections. Content is published online weekly on Mondays and Thursdays, and the embargo is set at 16:00 London time (GMT)/11:00 am US Eastern time (EST) on the day of publication. Now is the time to inform your Public Relations or Press Office about your paper, as they might be interested in promoting its publication. This will allow them time to prepare an accurate and satisfactory press release. Include your manuscript tracking number (NSMB-A45296D) and our journal name, which they will need when they contact our press office.

About one week before your paper is published online, we shall be distributing a press release to news organizations worldwide, which may very well include details of your work. We are happy for your institution or funding agency to prepare its own press release, but it must mention the embargo date and Nature Structural & Molecular Biology. If you or your Press Office have any enquiries in the meantime, please contact press@nature.com.

Please note that *Nature Structural & Molecular Biology* is a Transformative Journal (TJ). Authors may publish their research with us through the traditional subscription access route or make their paper immediately open access through payment of an article-processing charge (APC). Authors will not be required to make a final decision about access to their article until it has been accepted. http://www.nature.com/transformative

[href="https://www.springernature.com/gp/open-research/transformative-journals">](https://www.springernature.com/gp/open-research/transformative-journals) Find out more about Transformative Journals

Authors may need to take specific actions to achieve compliance with funder and institutional open access mandates. If your research is supported by a funder that requires immediate open access (e.g. according to Plan S principles) then you should select the gold OA route, and we will direct you to the compliant route where possible. For authors selecting the subscription publication route, the journal's standard licensing terms will need to be accepted, including self-archiving policies. Those licensing terms will supersede any other terms that the author or any third party may assert apply to any version of the manuscript.

Sincerely,

Katarzyna Ciazynska
(she/her)
Associate Editor
Nature Structural & Molecular Biology
<https://orcid.org/0000-0002-9899-2428>
